**Subject Category:**
Biology (whole organism)

palaeontology/biomechanics/evolution

vertebral morphology, hyposphene-hypantrum, Archosauria, body size

**Author for correspondence:**
Candice M. Stefanic
e-mail: cms292@vt.edu

# The evolution and role of the hyposphene-hypantrum articulation in Archosauria: phylogeny, size and/or mechanics?

## Candice M. Stefanic[1,2] and Sterling J. Nesbitt[1]

[1]Department of Geosciences, Virginia Tech, Blacksburg, VA 24061, USA
[2]Department of Anatomical Sciences, Stony Brook University, Stony Brook, NY 11794, USA

CMS, 0000-0002-7196-999X

Living members of Archosauria, the reptile clade containing Crocodylia and Aves, have a wide range of skeletal morphologies, ecologies and body size. The range of body size greatly increases when extinct archosaurs are included, because extinct Archosauria includes the largest members of any terrestrial vertebrate group (e.g. 70-tonne titanosaurs, 20-tonne theropods). Archosaurs evolved various skeletal adaptations for large body size, but these adaptations varied among clades and did not always appear consistently with body size or ecology. Modification of intervertebral articulations, specifically the presence of a hyposphene-hypantrum articulation between trunk vertebrae, occurs in a variety of extinct archosaurs (e.g. non-avian dinosaurs, pseudosuchians). We surveyed the phylogenetic distribution of the hyposphene-hypantrum to test its relationship with body size. We found convergent evolution among large-bodied clades, except when the clade evolved an alternative mechanism for vertebral bracing. For example, some extinct lineages that lack the hyposphene-hypantrum articulation (e.g. ornithischians) have ossified tendons that braced their vertebral column. Ossified tendons are present even in small taxa and in small-bodied juveniles, but large-bodied taxa with ossified tendons reached those body sizes without evolving the hyposphene-hypantrum articulation. The hyposphene-hypantrum was permanently lost in extinct crownward members of both major archosaur lineages (i.e. Crocodylia and Aves) as they underwent phyletic size decrease, changes in vertebral morphology and shifts in ecology.

# 1. Introduction

Living terrestrial vertebrates have a wide array of body size from tiny frogs (e.g. the 7.7 mm long *Paedophryne amanuensis* [1]) to enormous elephants (e.g. 7-tonne *Elephas maximus* [2]), but the fossil record shows that the disparity of body sizes in extinct terrestrial vertebrates has been much greater. Some reptiles (e.g. non-avian dinosaurs [3]) reached over 70 tonnes in estimated body mass, and some mammals (e.g. *Paraceratherium*) probably surpassed 15 tonnes [4]. Many studies have focused on reconstructing the evolution of body size [5–11] through time. However, fewer studies have focused on skeletal features that may have helped support phyletic body size increase across lineages [12,13]. The skeletons of large and small vertebrates are generally different in form and proportions, but it has been challenging to pinpoint differences correlated with increases in body size, but mechanically supported larger bodies.

Several orders of magnitude differentiate between the smallest and largest dinosaurs (e.g. sauropods versus hummingbirds). Size increases in some dinosaur clades were accompanied by substantial changes in limbs through columniation [14], joint articulations [15], vertebral morphology [16] and growth rates [17]. Of these skeletal changes, the evolution of vertebrae and body size has been correlated with increased pneumatization [18], the addition of neural spine projections [16] and the increase of complex neurocentral sutures [19]. Additionally, many large non-avian dinosaurs evolved an accessory intervertebral articulation between the trunk vertebrae—a hyposphene-hypantrum articulation [20,21] that appears absent in smaller dinosaurs (e.g. *Rahonavis*, *Mahakala*, *Microvenator*, *Mononykus*, *Parvicursor*) and entire groups (e.g. ornithischians). Similar structures have recently been found in large pseudosuchian archosaurs [22,23] (e.g. *Desmatosuchus*, *Poposaurus*) but again absent in smaller relatives (e.g. *Parringtonia*, *Effigia*). Finally, hyposphene-hypantrum articulations are not present in any living or extinct members of Crocodylia and Aves, the crown clades of Archosauria, and the articulation appears to have been lost in extinct lineages leading to those clades as they became smaller overall and in some cases changed their ecologies.

To investigate the complex pattern of the presence and absence of the hyposphene-hypantrum articulation and its relationship with body size, we studied the distribution of the articulation across Archosauria. We focused solely on members of Archosauria because no immediate outgroups have been reported to possess these articulations or any structures homologous to the articulation. Here, we hypothesize that the presence of the hyposphene-hypantrum articulation in extinct archosaurs is more correlated with body size than ancestry, although there are exceptions in clades that modified their trunk regions (e.g. ornithischians, titanosaurs). To test this hypothesis, we examined fossil specimens to determine presence or absence of the hyposphene-hypantrum and gathered body size data for those taxa (table 1). Then we analysed within a phylogenetic framework the history of the trait of the articulation and whether the losses and gains can be related to changes in body size. Although we hypothesize that the hyposphene-hypantrum articulation had biomechanical implications for growth to large body size, this is a difficult question to tackle because all of the taxa that possess the articulation are extinct and thus all associated soft tissue structures are not preserved. The goal of this study is to simply report the distribution of the hyposphene-hypantrum in archosaurs and the body sizes at which it appears in the clades that possess the articulation.

## 1.1. Institutional abbreviations

AMNH—American Museum of Natural History, New York, NY, USA; CM—Carnegie Museum of Natural History, Pittsburgh, PA, USA; CPE2—Coleção Municipal, São Pedro do Sul, Brazil; FMNH—Field Museum of Natural History, Chicago, USA; GR—Ruth Hall Museum of Paleontology, Ghost Ranch, NM, USA; IVPP—Institute of Vertebrate Paleontology and Paleoanthropology, Beijing, China; MNA—Museum of Northern Arizona, Flagstaff, AZ, USA; MSM—Arizona Museum of Natural History (formerly Mesa Southwest Museum), Mesa, AZ, USA; NHMUK—Natural History Museum, London, UK; NMNH—National Museum of Natural History, Washington, DC, USA; NMT—National Museum of Tanzania, Dar es Salaam, Tanzania; PEFO—Petrified Forest National Park, Petrified Forest, AZ, USA; PIZ—Paläontologisches Institut und Museum der Universität Zürich, Switzerland; PVL—Instituto Miguel Lillo, Tucumán, Argentina; PVSJ—Division of Paleontology of the Museo de Ciencias Naturales de la Universidad Nacional de San Juan, San Juan, Argentina; SAM—Iziko: South African Museum, Cape Town, South Africa; SM—Sirindhorn Museum, Changwat Kalasin, Thailand; SMNS—Staatliches Museum für Naturkunde, Stuttgart, Germany; TMM—Jackson School of Geosciences Vertebrate Paleontology Laboratory, University of Texas at Austin, Austin, TX, USA; TTU P—Texas Tech University Museum, Lubbock, TX, USA; UA—University of Antananarivo, Madagascar; UCMP—University of California Museum of Paleontology, Berkeley, CA, USA; UFRGS-PV—Instituto de Geociencias, Universidade Federal do Rio Grande do Sul, Porto Allegre, Brazil; USNM—National Museum of Natural History, Washington, DC, USA

**Table 1.** Table including taxon name, femoral length, score for presence (1) or absence (0) of the hyposphene-hypantrum articulation, reference for femoral length, and major archosaurian group to which the taxon is assigned. Taxa omitted from our statistical analysis are indicated in grey.

| taxon | femoral length (mm) | hyposphene-hypantrum (presence/absence) | source | major archosaurian group |
|---|---|---|---|---|
| *Postosuchus kirkpatrickorum* | 528 | 1 | Turner & Nesbitt [11] | pseudosuchian |
| *Postosuchus alisonae* | 558 | 1 | Turner & Nesbitt [11] | pseudosuchian |
| *Fasolasuchus* | 750 | 1 | Turner & Nesbitt [11] | pseudosuchian |
| *Batrachotomus* | 420 | 1 | Turner & Nesbitt [11] | pseudosuchian |
| *Prestosuchus* | 538 | 1 | Turner & Nesbitt [11] | pseudosuchian |
| *Poposaurus langstoni* | 353 | 1 | Turner & Nesbitt [11] | pseudosuchian |
| *Poposaurus gracilis* | 353 | 1 | Turner & Nesbitt [11] | pseudosuchian |
| *Sillosuchus* | 440 | ? | Alcober & Parrish [24] | pseudosuchian |
| *Shuvosaurus* | 255 | 0 | Turner & Nesbitt [11] | pseudosuchian |
| *Effigia* | 301 | 0 | Turner & Nesbitt [11] | pseudosuchian |
| *Arizonasaurus* | 490 | 1 | Turner & Nesbitt [11] | pseudosuchian |
| *Xilousuchus* | 302 | 1 | Turner & Nesbitt [11] | pseudosuchian |
| *Ticinosuchus* | 240 | ? | Turner & Nesbitt [11] | pseudosuchian |
| *Nundasuchus* | 230 | 0 | NMT RB48 | pseudosuchian |
| *Mandasuchus* | 212 | 1 | Butler *et al.* [25] | pseudosuchian |
| *Turfanosuchus* | 136 | 0 | Turner & Nesbitt [11] | pseudosuchian |
| *Gracilisuchus* | 78 | 0 | Turner & Nesbitt [11] | pseudosuchian |
| *Longosuchus* | 337 | ? | Turner & Nesbitt [11] | pseudosuchian (aetosaur) |
| *Desmatosuchus* | 450 | 1 | Parker [22] | pseudosuchian (aetosaur) |
| *Scutarx* | ? | 1 | PEFO 34045 | pseudosuchian (aetosaur) |
| *Typothorax* | 291 | 0 | Heckert *et al.* [26] | pseudosuchian (aetosaur) |
| *Paratypothorax* | ? | 0 | Hunt & Lucas [27] | pseudosuchian (aetosaur) |
| *Aetobarbakinoides* | 120 | 0 | Desojo *et al.* [28] | pseudosuchian (aetosaur) |
| *Coahomasuchus* | ? | 0 | TMM 31100-437 | pseudosuchian (aetosaur) |
| *Parringtonia* | ? | 0 | NMT RB426 | pseudosuchian |
| Ornithosuchidae | 114 | — | ancestral state from Turner & Nesbitt [11] | Ornithosuchidae (node) |
| *Smilosuchus* | 545 | 0 | Turner & Nesbitt [11] | phytosaur |
| *Machaeroprosopus* | 444 | 0 | Turner & Nesbitt [11] | phytosaur |

(*Continued.*)

| taxon | femoral length (mm) | hyposphene-hypantrum (presence/absence) | source | major archosaurian group |
|---|---|---|---|---|
| Phytosauria | 235 | — | ancestral state from Turner & Nesbitt [11] | Phytosauria (node) |
| *Sphenosuchus* | 140 | 0 | Turner & Nesbitt [11] | crocodylomorph |
| *Terrestrisuchus* | 80 | 0 | Turner & Nesbitt [11] | crocodylomorph |
| *Sarcosuchus* | 860 | 0 | estimated from fig. 3 in Sereno *et al.* [29] | crocodylomorph |
| *Deinosuchus* | 530 | 0 | estimated from fig. 5 in Farlow *et al.* [30] | crown crocodylian |
| *Alligator* | 250 | 0 | estimated from fig. 5 in Farlow *et al.* [30] | crown crocodylian |
| *Crocodylus* | 275 | 0 | estimated from fig. 5 in Farlow *et al.* [30] | crown crocodylian |
| Crocodylomorpha | 260 | — | ancestral state from Turner & Nesbitt [11] | Crocodylomorpha (node) |
| *Teleocrater* | 170 | 1 | Nesbitt *et al.* [31] | avemetatarsalian |
| *Asilisaurus* | 177 | 1 | Turner & Nesbitt [11] | avemetatarsalian |
| *Silesaurus* | 200 | 1 | Turner & Nesbitt [11] | avemetatarsalian |
| *Dromomeron* | 95 | 0 | Nesbitt *et al.* [32] | avemetatarsalian |
| *Psittacosaurus* | 160 | 0 | Butler *et al.* [33] | ornithischian |
| *Torosaurus* | 874 | 0 | Hunt & Lehman [34] | ornithischian |
| *Stygimoloch* | 441 | 0 | Carrano [6] | ornithischian |
| *Thescelosaurus* | 355 | 0 | Galton [35] | ornithischian |
| *Stegosaurus* | 978 | 0 | Redelstorff & Sander [36] | ornithischian |
| Ornithischia | 112 | — | ancestral state from Turner & Nesbitt [11] | Ornithischia (node) |
| *Plateosaurus* | 750 | 1 | Turner & Nesbitt [11] | sauropodomorph |
| *Saturnalia* | 157 | 1 | Turner & Nesbitt [11] | sauropodomorph |
| *Sarahsaurus* | 400 | 1 | Marsh [37] | sauropodomorph |
| *Haplocanthosaurus* | 1275 | 1 | Mazzetta *et al.* [3] | sauropodomorph |
| *Barosaurus* | 1440 | 1 | McIntosh [38] | sauropodomorph |
| *Diplodocus* | 1540 | 1 | measured from CM 94 | sauropodomorph |
| *Vulcanodon* | 1100 | 1 | Turner & Nesbitt [11] | sauropodomorph |
| *Alamosaurus* | 1850 | 0 | Fowler & Sullivan [39] | sauropodomorph (titanosaur) |
| *Argentinosaurus* | 2557 | 0 | Mazzetta *et al.* [3] | sauropodomorph (titanosaur) |
| Sauropodomorpha | 162 | — | ancestral state from Turner & Nesbitt [11] | sauropodomorpha (node) |
| *Tawa* | 174 | 1 | Turner & Nesbitt [11] | theropod |
| *Dilophosaurus* | 552 | 1 | Turner & Nesbitt [11] | theropod |

(*Continued.*)

**Table 1.** (Continued.)

| taxon | femoral length (mm) | hyposphene-hypantrum (presence/absence) | source | major archosaurian group |
|---|---|---|---|---|
| *Struthiomimus* | 486 | 1 | Christiansen & Fariña [40] | theropod |
| *Ornitholestes* | 210 | 1 | Christiansen & Fariña [40] | theropod |
| *Saurornitholestes* | 214 | 1 | Christiansen & Fariña [40] | theropod |
| *Tyrannosaurus* | 1273 | 1 | Christiansen & Fariña [40] | theropod |
| *Allosaurus* | 872 | 1 | Bybee et al. [41] | theropod |
| *Velociraptor* | 238 | 1 | Norell & Mackovicky [42] | theropod |
| *Microvenator* | 124 | 0 | Makovicky & Sues [43] | theropod |
| *Deinocheirus* | 1381 | 1 | Lee et al. [44] | theropod |
| *Troodon* | 310 | 1 | Varricchio [45] | theropod |
| *Saurornithoides* | ? | 1 | Norrell et al. [46] | theropod |
| *Mahakala* | 79 | 0 | Turner et al. [47] | theropod |
| *Mononykus* | 150 | 0 | Chiappe et al. [48] | theropod |
| *Parvicursor* | 50 | 0 | Karhu & Rautian [49] | theropod |
| *Patagonykus* | 285 | 1 | Novas [50] | theropod |
| *Haplocheirus* | 214 | 1 | Choiniere et al. [51] | theropod |
| *Avimimus* | 186 | 1 | Vickers-Rich et al. [52] | theropod |
| *Gigantoraptor* | 1100 | 1 | Xu et al. [53] | theropod |
| *Deinonychus* | 344 | 1 | Novas et al. [54] | theropod |
| *Buitreraptor* | 77 | ? | Makovicky et al. [55] | theropod |
| *Unenlagia* | 380 | 1 | Novas & Puerta [56] | theropod |
| *Austroraptor* | 560 | ? | Novas et al. [54] | theropod |
| *Rahonavis* | 86.9 | 0 | Carrano [6] | theropod |
| *Microraptor* | 75 | ? | Hwang et al. [57] | theropod |
| *Shanag* | 55 | ? | Turner et al. [58] | theropod |
| *Achillobator* | 550 | 1 | Perle et al. [59] | theropod |
| *Archaeopteryx* | 52.6 | 0 | Wellnhofer [60] | theropod |
| Theropoda | 187 | — | ancestral state from Turner & Nesbitt [11] | Theropoda (node) |
| *Struthio camelus* | 334 | 0 | FMNH 3392 | crown Aves |

# 2. Material and methods

## 2.1. Recognition of the hyposphene-hypantrum articulation

The hyposphene-hypantrum articulation is defined as a two-part complex consisting of a bony projection, the hyposphene, on the posterior face of the neural arch of the vertebra that fits into a complementary space, the hypantrum, on the anterior face of the neural arch of the subsequent vertebra [23]. The hyposphene is orientated dorsoventrally and is symmetrical across the midline in posterior view. The hyposphene is located ventral to the articular surfaces of the postzygapophyses that face dorsally and are positioned between 0 and 45 degrees dorsal to the horizontal in posterior view. The hyposphene is a continuation of the articular surfaces of the postzygapophyses where they converge and is dorsal to the neural canal (figure 1). The hyposphene and hypantrum articulate

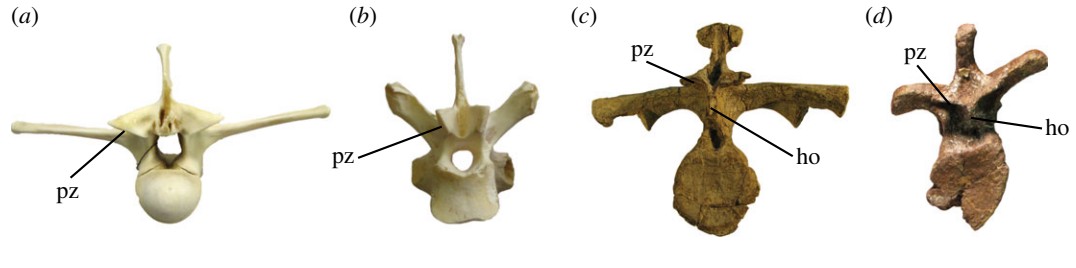

**Figure 1.** Examples of articular surfaces on the posterior aspect of several archosaur trunk vertebrae, including extant species that lack the hyposphene-hypantrum, *Alligator mississippiensis* [TMM M-12606] (*a*) and *Struthio camelus* [NMNH 291160] (*b*), and extinct taxa that possess the hyposphene-hypantrum articulation, *Desmatosuchus spurensis* [MNA V9300] (*c*) and *Plateosaurus* [AMNH 2108] (*d*). pz: postzygapophysis, ho: hyposphene. Scale bars, 1 cm.

precisely; thus the hyposphene is comparable in both size and shape with its corresponding hypantrum. The lateral surfaces of the hyposphene articulate with the medial surfaces of the hypantrum, which is located between and ventral to the prezygapophyses and dorsal to the neural canal [23]. The articular surfaces of the prezygapophyses continue ventrally from their medial surfaces to form the articular surfaces of the hyposphene. In dorsal view, a hypantrum appears as a gap framed by parallel to sub-parallel medial surfaces of the prezygapophyses, which contact the neural arch just dorsal to the neural canal.

Based on these criteria for recognizing the presence of the hyposphene-hypantrum articulation in fossils, we examined specimens and searched for figured material in literature of fossil trunk vertebrae. We surveyed 97 taxa from nearly every major clade within Archosauria (e.g. Theropoda, Sauropodomorpha, Ornithischia, Paracrocodylomorpha), as well as a few proximate outgroups (e.g. *Euparkeria capensis*, *Vancleavea campi*, *Erythrosuchus africanus*). For each taxon we recorded the presence or absence of the hyposphene-hypantrum articulation (table 1).

## 2.2. Body size correlation

For all taxa scored for the presence or absence of the hyposphene-hypantrum articulation, we also collected body size information. We used femoral length (= FL) as a proxy for body size (following [5–7,9–11,30,40]) and we used the largest recorded length in the published literature for each species (table 1).

## 2.3. Phylogenetic survey

In order to place our analysis in a phylogenetic context, we created a phylogenetic tree for each of the two major branches of archosaurs (Pseudosuchia, Avemetatarsalia). We combined previously published phylogenies [6,7,29,31,44,48,54,61–65] by hand. In nearly all cases, the relationships we present are consistent across all published hypotheses; exceptions are noted below. We then optimized, using maximum parsimony, the presence or absence of the hyposphene-hypantrum in Mesquite version 3.40 [66] to pinpoint acquisition and loss of the articulation across the phylogenies.

To understand how body size maps onto the phylogeny, we assigned published ancestral size reconstructions (femoral length in mm) to corresponding nodes of interest on our phylogenies. We obtained these ancestral reconstruction femoral lengths for taxa across Pseudosuchia from [11], whose authors used an extensive phylogenetic analysis from [64], and for taxa across Neotetanurae by converting $Log_{10}$ femoral reconstruction from [44] to measurements in mm. We mapped these reconstructed femoral lengths onto major nodes (e.g. Paracrocodylomorpha, Aetosauria, Sauropodomorpha, Theropoda) of our phylogenies, which target where the hyposphene-hypantrum is either lost or gained. We included the group Squamata on our phylogenies for context and to illustrate that within Reptilia no other taxa outside Archosauria possess the hyposphene-hypantrum articulation. Whereas snakes have an accessory intervertebral articulation, the zygosphene-zygantrum articulation, these structures are not homologous to the hyposphene-hypantrum. Some archosaur taxa are included in our phylogeny solely for completeness because hyposphene-hypantrum presence or absence could not be determined. This is because the preservation of known material of those taxa prevents us from examining the anterior and posterior views of their vertebrae in three dimensions (e.g. *Sillosuchus longicervix*, PVSJ 85; *Ticinosuchus ferox*; PIZ T 2817).

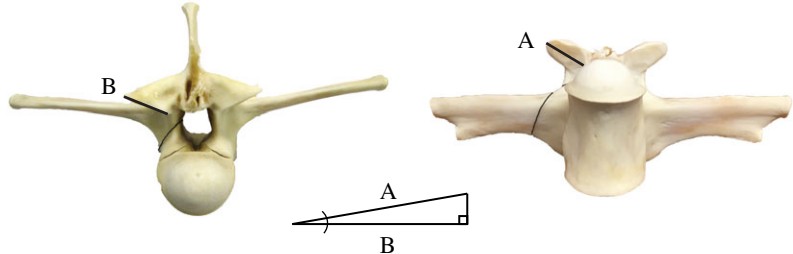

**Figure 2.** Schematic illustrating the in-person and Photoshop measurements of vertebrae. Measurements 'A' and 'B' are shown on an *Alligator mississippiensis* vertebra in posterior and ventral oblique views. Measurement 'A'/ Measurement 'B' = 1.1 (conversion factor).

## 2.4. Measurements

We sampled 17 archosaur taxa from both fossil and extant osteological collections to measure the articular processes (the postzygapophyses in all taxa and hyposphene structures when applicable) on the posterior aspects of trunk vertebrae to track surface length involved in intervertebral articulation. Even though, as noted, living archosaurs lack the hyposphene-hypantrum articulation, we measured their vertebrae to see if there were other consistent differences that varied with size and phylogenetic placement. The postzygapophyses were measured along the maximum length of their articular surface—from the point where they contact the neural arch to their distal-most aspect (measurement 'A' in figure 2). Hyposphenes were measured along their lateral faces. To obtain a unit-less metric independent of body size, we also measured centrum height for each vertebra. By dividing the maximum length (long axis) of one postzygapophysis by the centrum height for each vertebra, we obtained a metric corrected for body size that could be used as a proxy for the relative surface area of articulation between vertebrae.

We measured (in mm) complete presacral vertebral columns in two crocodylian species (*Alligator mississippiensis* [TMM M-12606], *Crocodylus acutus* [USNM 247943]) and six avian species (*Dromaius novaehollandiae* [NMNH 345221], *Apteryx australis* [NMNH 500629], *Rhea americana* [NMNH 20286], *Struthio camelus* [NMNH 291160], *Gallus gallus* [NMNH 560790], *Chauna chavaria* [NMNH 18996]). In each we measured the long axis of the left postzygapophysis (measurement 'A' in figure 2) and centrum height to normalize vertebral size. All measurements of these extant taxa were done in-person and recorded in millimetres (tables 2 and 3).

We measured vertebrae of the pseudosuchian *Parringtonia gracilis* (NMT RB426, [67]), the theropod dinosaur *Dilophosaurus wetherilli* (UCMP 37302, [68]) and the phytosaur '*Machaeroprosopus*' *zunii* (UCMP 27036, [69]). The fossil vertebral columns were not complete, but all specimens preserved between 6 and 13 vertebrae, complete enough to measure and approximate vertebral position, with an estimated error of one position in either direction. We made these estimates based on the location of the para- and diapophyses on the centrum using a complete vertebral column from an extant *Alligator mississippiensis* (TMM M-12606) and a nearly complete vertebral column from the Jurassic theropod *Allosaurus fragilis* [70] as models.

To expand our dataset of extinct archosaurs, we included measurements of an additional pseudosuchian with a fairly complete vertebral column, *Desmatosuchus spurensis* (MNA V9300, [22]), based on our photographs. These capture posterior views of isolated vertebrae as close to perpendicular (between the posterior plane and the camera lens) as possible. We used the ruler tool in Photoshop CC 2015 to take measurements of (i) standardized length of a cm (on the scale bar included in the photograph), (ii) centrum height, and (iii) length of the left prezygapophysis (or the right if the left was not preserved or post-depositionally altered) (measurement 'B' in figure 2). Using the 1 cm (100 mm) measurement from the scale bar, we recorded the zygapophyseal length and centrum height in millimetres.

We analysed whether the postzygapophyseal photograph measurements were slightly out-of-plane from our empirical measurements of specimens by computing a conversion factor between the two methods. To do this, we took both types of measurements on the same vertebra to determine how different those measurements were from each other. In that way, we could multiply one measure by that factor to convert it to a number about equal to the other measure (table 4). Using several in-person and photographic measurements from vertebrae of *Alligator mississippiensis* (TMM M-12606), we determined that the ratio between measurement 'B' (Photoshop) and measurement 'A' (in-person) (figure 2) was about 1.1 for three samples from varying positions along the presacral region of the *Alligator* vertebral column (i.e. the in-person measurement is 1.1 times greater than the measurement

**Table 2.** Table including vertebral measurements of the extant crocodylians used in figures 8 and 9. Measurements are in millimetres.

| taxon: | Crocodylus acutus | | | taxon: | Alligator mississippiensis | | |
|---|---|---|---|---|---|---|---|
| position | max length l. postzyg. | centrum height | LPZ/centrum height | position | max length l. postzyg. | centrum height | LPZ/centrum height |
| 2 | 17.07 | 23.74 | 0.7 | 2 | 6.15 | 12.64 | 0.5 |
| 3 | 19.64 | 24.66 | 0.8 | 3 | 7.59 | 12.75 | 0.6 |
| 4 | 21.05 | 26.28 | 0.8 | 4 | 8.41 | 13.58 | 0.6 |
| 5 | 21.62 | 25.78 | 0.8 | 5 | 8.91 | 13.78 | 0.6 |
| 6 | 24.78 | 25.3 | 1.0 | 6 | 9.48 | 14.45 | 0.7 |
| 7 | 26.07 | 26.25 | 1.0 | 7 | 10.48 | 14.54 | 0.7 |
| 8 | 26.73 | 26.55 | 1.0 | 8 | 10.55 | 14.34 | 0.7 |
| 9 | 23.28 | 26 | 0.9 | 9 | 9.91 | 14.01 | 0.7 |
| 10 | 25.56 | 26.73 | 1.0 | 10 | 9.84 | 14.29 | 0.7 |
| 11 | 26.9 | 27.99 | 1.0 | 11 | 11.93 | 15.34 | 0.8 |
| 12 | 27.6 | 28.63 | 1.0 | 12 | 11.8 | 14.93 | 0.8 |
| 13 | 27.02 | 29.52 | 0.9 | 13 | 12.54 | 15.38 | 0.8 |
| 14 | 27.81 | 30.65 | 0.9 | 14 | 12 | 15.12 | 0.8 |
| 15 | 29.68 | 30.48 | 1.0 | 15 | 11.76 | 14.95 | 0.8 |
| 16 | 28.03 | 29.72 | 0.9 | 16 | 11.41 | 14.94 | 0.8 |
| 17 | 24.91 | 28.91 | 0.9 | 17 | 10.59 | 14.61 | 0.7 |
| 18 | 26.27 | 28 | 0.9 | 18 | 10.53 | 14.67 | 0.7 |
| 19 | 25.63 | 27.08 | 0.9 | 19 | 10.56 | 14.67 | 0.7 |
| 20 | 25.05 | 26.48 | 0.9 | 20 | 10.93 | 15.12 | 0.7 |
| 21 | 25.97 | 25.51 | 1.0 | 21 | 10.8 | 14.72 | 0.7 |
| 22 | 25.76 | 25.55 | 1.0 | 22 | 10.83 | 14.57 | 0.7 |
| 23 | 27.35 | 22.68 | 1.2 | 23 | 10.61 | 14.5 | 0.7 |

taken from a photograph of the same vertebra in Photoshop CC 2015). Our assumption in using this conversion factor for the *Desmatosuchus spurensis* vertebrae is that the angle of the postzygapophyses in relation to the neural arch is similar across all archosaur taxa, and based on personal observation we conclude that this correction does not distort the overall pattern of data. Other factors, such as compression during fossilization, may warp these angles in some extinct taxa.

To understand how the maximum length of articular surface of the zygapophyses varies within the vertebral column of a single animal as well as among taxa, we plotted each measured vertebra as a separate data point. For each point, the *x*-axis value was the position in the vertebral column (estimated for the extinct taxa based on the location of the para- and diapophyses on the centrum), and the *y*-axis value was the maximum length of the left postzygapophysis (measurement 'A' for in-person measured taxa and measurement 'B' × 1.1 for the Photoshop measured taxa) divided by vertebral centrum height. We plotted only trunk vertebrae (presacral position 10 to the last vertebra before the sacrum) because this is the only portion of the vertebral column that has the hyposphene-hypantrum articulations [23]. These data illustrate how the length of the zygapophyses, relative to the vertebral size, changes along the trunk series of an individual and in different extinct and extant clades.

## 2.5. Statistical analyses

Because we recorded our data dichotomously (presence versus absence; table 1), we performed logistic regression analyses in R (see electronic supplementary material for code and outputs) to determine possible statistical significance of the relationships between the presence of the hyposphene-hypantrum articulation and large body size in the branches of Archosauria that gained and lost the

**Table 3.** Table including vertebral measurements of the extant birds used in figures 8 and 9. Measurements are in millimetres.

| taxon: | *Dromaius novaehollandiae* | | | taxon: | *Apteryx australis* | | |
|---|---|---|---|---|---|---|---|
| position | max length l. postzyg. | centrum height | LPZ/centrum height | position | max length l. postzyg. | centrum height | LPZ/centrum height |
| 2 | 8.1 | 5.51 | 1.5 | 2 | 4.81 | 6.18 | 0.8 |
| 3 | 8.73 | 7.15 | 1.2 | 3 | 4.81 | 5.63 | 0.9 |
| 4 | 9.86 | 9.73 | 1.0 | 4 | 5.12 | 6.48 | 0.8 |
| 5 | 9.99 | 10.85 | 0.9 | 5 | 4.32 | 6.50 | 0.7 |
| 6 | 8.86 | 11.01 | 0.8 | 6 | 4.36 | 5.53 | 0.8 |
| 7 | 7.59 | 10.05 | 0.8 | 7 | 5.44 | 6.91 | 0.8 |
| 8 | 7.65 | 10.67 | 0.7 | 8 | 4.53 | 7.46 | 0.6 |
| 9 | 8.22 | 12.61 | 0.7 | 9 | 4.87 | 7.24 | 0.7 |
| 10 | 11.9 | 14.87 | 0.8 | 10 | 5.7 | 7.44 | 0.8 |
| 11 | 12.79 | 15.76 | 0.8 | 11 | 6.08 | 6.13 | 1.0 |
| 12 | 14.51 | 17.47 | 0.8 | 12 | 5.57 | 6.25 | 0.9 |
| 13 | 13 | 17.00 | 0.8 | 13 | 4.71 | 7.27 | 0.6 |
| 14 | 12.36 | 16.84 | 0.7 | 14 | 5.1 | 8.14 | 0.6 |
| 15 | 12.58 | 19.99 | 0.6 | 15 | 4.76 | 7.35 | 0.6 |
| 16 | 14.23 | 22.79 | 0.6 | 16 | 5.47 | 7.93 | 0.7 |
| 17 | 14.8 | 21.10 | 0.7 | 17 | 4.34 | 8.16 | 0.5 |
| 18 | 15.45 | 23.78 | 0.6 | 18 | 4.72 | 7.15 | 0.6 |
| 19 | 14.87 | 18.73 | 0.8 | 19 | 4.79 | 7.75 | 0.6 |
| 20 | 13.81 | 17.43 | 0.8 | 20 | 4.39 | 9.81 | 0.4 |
| 21 | 12.06 | 20.73 | 0.6 | 21 | 5.51 | 8.11 | 0.7 |
| 22 | 10.06 | 20.86 | 0.5 | 22 | 5.65 | 10.87 | 0.5 |
| 23 | 10.78 | 24.71 | 0.4 | 23 | 4.74 | 9.90 | 0.5 |
| 24 | 11.19 | 27.34 | 0.4 | | | | |
| 25 | 12.72 | 27.40 | 0.5 | | | | |
| 26 | 13.23 | 29.15 | 0.5 | | | | |
| taxon: | *Struthio camelus* | | | taxon: | *Chauna chavaria* | | |
| position | max length l. postzyg. | centrum height | LPZ/centrum height | position | max length l. postzyg. | centrum height | LPZ/centrum height |
| 2 | 8.92 | 20.34 | 0.4 | 2 | 4.32 | 3.53 | 1.2 |
| 3 | 10.31 | 10.68 | 1.0 | 3 | 4.24 | 4.96 | 0.9 |
| 4 | 10.8 | 12.73 | 0.8 | 4 | 5.02 | 4.57 | 1.1 |
| 5 | 10.63 | 14.70 | 0.7 | 5 | 4.22 | 5.33 | 0.8 |
| 6 | 10.71 | 17.32 | 0.6 | 6 | 4.77 | 5.43 | 0.9 |
| 7 | 10.9 | 19.03 | 0.6 | 7 | 6.41 | 4.82 | 1.3 |
| 8 | 11.24 | 14.14 | 0.8 | 8 | 5.73 | 4.39 | 1.3 |
| 9 | 12.27 | 16.21 | 0.8 | 9 | 6.22 | 6.34 | 1.0 |
| 10 | 15.49 | 19.26 | 0.8 | 10 | 7.23 | 6.34 | 1.1 |
| 11 | 17.28 | 19.45 | 0.9 | 11 | 6.8 | 5.71 | 1.2 |

(*Continued.*)

| taxon: | *Struthio camelus* | | | taxon: | *Chauna chavaria* | | |
|---|---|---|---|---|---|---|---|
| position | max length l. postzyg. | centrum height | LPZ/centrum height | position | max length l. postzyg. | centrum height | LPZ/centrum height |
| 12 | 17.02 | 23.45 | 0.7 | 12 | 7.5 | 6.04 | 1.2 |
| 13 | 17.74 | 23.55 | 0.8 | 13 | 7.53 | 6.62 | 1.1 |
| 14 | 17.72 | 23.64 | 0.7 | 14 | 7.7 | 6.41 | 1.2 |
| 15 | 18.35 | 24.90 | 0.7 | 15 | 7.23 | 7.33 | 1.0 |
| 16 | 19.01 | 25.64 | 0.7 | 16 | 7.77 | 6.79 | 1.1 |
| 17 | 18.39 | 27.05 | 0.7 | 17 | 7.55 | 6.47 | 1.2 |
| 18 | 19.49 | 28.16 | 0.7 | 18 | 6.69 | 7.58 | 0.9 |
| 19 | 17.95 | 24.30 | 0.7 | 19 | 6.67 | 5.82 | 1.1 |
| 20 | 17.35 | 27.88 | 0.6 | 20 | 6.01 | 7.46 | 0.8 |
| 21 | 18.76 | 21.61 | 0.9 | 21 | 6.04 | 7.34 | 0.8 |
| 22 | 15.4 | 19.92 | 0.8 | 22 | 5.04 | 8.64 | 0.6 |
| 23 | 13.64 | 28.16 | 0.5 | 23 | 6.64 | 8.02 | 0.8 |
| 24 | 12.58 | 32.15 | 0.4 | 24 | 8.18 | 8.58 | 1.0 |
| 25 | 14.99 | 31.51 | 0.5 | | | | |
| 26 | 16.77 | 34.77 | 0.5 | | | | |

articulation throughout their phylogenetic history (i.e. pseudosuchians and avemetarsalians). We analysed the two clades separately because we noticed that species on each side of the tree evolved the hyposphene-hypantrum at different body sizes, and therefore combining them would distort how the presence of the articulation may be related to specific body size. The logistic regression informs how well body size (represented by femoral length) explains whether any given data point plots as 0 or 1 on the $y$-axis. This test does not produce true $R^2$ values; instead, our analyses produce pseudo $R^2$ values [71] that can be interpreted as a measure of how closely the presence or absence of the hyposphene-hypantrum can be explained by body size. The curves in the logistic regression plots represent the probability that any given $x$-value falls in one of the two discrete data categories (presence or absence). We excluded from our statistical analyses several groups of archosaurs that lack the hyposphene-hypantrum (e.g. titanosaurs, ornithischians, pterosaurs); these groups lost the articulation at or before the origins of their clade and then evolved other bracing mechanisms when they secondarily evolved large body (see §4.2). Some of our specimens could not be included in the statistical analysis because no associated femora are known (e.g. *Desmatosuchus spurensis*, MNA V9300). The logistic regression plot yields visual thresholds for body size. In individual taxa with body size above the threshold, the hyposphene-hypantrum would be expected to be present, and in taxa with body size below it, the hyposphene-hypantrum would be expected to be absent.

## 3. Results

In our assessment of the pattern of presence or absence of the hyposphene-hypantrum articulation versus body size, we discovered that large extinct archosaurs have the hyposphene-hypantrum and smaller ones do not (figure 3), although there are a few notable exceptions (ornithischians, titanosaur sauropods, crown crocodylians, crown birds). Across the phylogenetic tree, we discovered that the articulation evolved during certain phyletic body size increases was lost during some phyletic size decreases. However, the thresholds of these relationships differ between the two major branches of Archosauria, and each group's threshold was represented by a range (figure 4). These ranges are simply the values of femoral length of the smallest taxon reported to possess the hyposphene-hypantrum and the largest taxon reported to lack the hyposphene-hypantrum. Within each range, we interpret a transitional body size where the articulation may or may not be seen in taxa of that size.

**Table 4.** Table including vertebral measurements of the extinct archosaurs used in figure 9. Measurements are in millimetres.

| taxon: | *Dilophosaurus wetherilli* | | | | |
| position | max length l. postzyg. | centrum height | LPZ/centrum height | l. hyposphene vertical length | LH + LPZ/centrum height |
| --- | --- | --- | --- | --- | --- |
| 13 | 15 | 59 | 0.25 | 19 | 0.58 |
| 14 | 18 | 55 | 0.33 | 15 | 0.60 |
| 15 | 16 | 59 | 0.27 | 6 | 0.37 |
| 16 | 17 | 58 | 0.29 | 11 | 0.48 |
| 17 | 20 | 56 | 0.36 | 13 | 0.59 |
| 18 | 15 | 56 | 0.27 | 9 | 0.43 |

| taxon: | *Desmatosuchus spurensis* | | | | | | |
| position | max length l. postzyg. | max l. PZ × 1.1 | centrum height | CH × 1.1 | LPZ/centrum height | l. hyposphene vertical length | LHVL × 1.1 | LH + LPZ/centrum height |
| --- | --- | --- | --- | --- | --- | --- | --- | --- |
| 10 | 13.04 | 14.34 | 69.53 | 76.48 | 0.19 | 7.96 | 8.75 | 0.30 |
| 11 | 14.22 | 15.64 | 68.89 | 75.78 | 0.21 | 15.36 | 16.90 | 0.43 |
| 12 | 13.36 | 14.70 | 68.96 | 75.86 | 0.19 | 34.92 | 38.41 | 0.70 |
| 13 | 27.68 | 30.45 | 72.75 | 80.03 | 0.38 | 19.78 | 21.75 | 0.65 |
| 14 | 28.39 | 31.23 | 80.03 | 88.03 | 0.35 | 36.96 | 40.65 | 0.82 |
| 15 | 20.00 | 22.00 | 90.17 | 99.18 | 0.22 | 45.04 | 49.55 | 0.72 |
| 16 | 25.35 | 27.89 | 87.75 | 96.52 | 0.29 | 34.14 | 37.56 | 0.68 |
| 19 | 16.73 | 18.40 | 92.22 | 101.44 | 0.18 | 24.51 | 26.96 | 0.45 |
| 20 | 38.02 | 41.82 | 114.58 | 126.04 | 0.33 | 37.41 | 41.15 | 0.66 |
| 21 | 29.31 | 32.24 | 111.26 | 122.38 | 0.26 | 27.46 | 30.21 | 0.51 |

(*Continued.*)

**Table 4.** (*Continued.*)

| taxon: | *Desmatosuchus spurensis* | | | | | | | | |
|---|---|---|---|---|---|---|---|---|---|
| position | max length l. postzyg. | max l. PZ × 1.1 | centrum height | CH × 1.1 | LPZ/centrum height | l. hyposphene vertical length | LHVL × 1.1 | LH + LPZ/centrum height |
| 22 | 29.97 | 32.97 | 108.74 | 119.61 | 0.28 | 18.15 | 19.97 | 0.44 |
| 23 | 36.66 | 40.33 | 129.55 | 142.51 | 0.28 | 27.31 | 30.04 | 0.49 |

| taxon: | *Parringtonia gracilis* | | |
|---|---|---|---|
| position | max. length of l. postzyg. | centrum height | LPZ/centrum height |
| 7 | 5.09 | 8.68 | 0.59 |
| 8 | 5.95 | 8.54 | 0.70 |
| 9 | 4.11 | 8.94 | 0.46 |
| 10 | 7.2 | 8.64 | 0.83 |
| 13 | 4.3 | 8.57 | 0.50 |
| 16 | 5.98 | 9.08 | 0.66 |
| 19 | 5.26 | 9.51 | 0.55 |
| 21 | 4.44 | 7.47 | 0.59 |
| 22 | 5.83 | 8.46 | 0.69 |

| taxon: | *'Machaeroprosopus' zunii* | | |
|---|---|---|---|
| position | max. length of l. postzyg. | centrum height | LPZ/centrum height |
| 10 | 41 | 77 | 0.53 |
| 11 | 39 | 84 | 0.46 |
| 12 | 34 | 87 | 0.39 |

(*Continued.*)

**Table 4.** (*Continued.*)

| taxon: | *'Machaeroprosopus' zunii* | | |
|---|---|---|---|
| position | max. length of l. postzyg. | centrum height | LPZ/centrum height |
| 13 | 31 | 86 | 0.36 |
| 14 | 27 | 86 | 0.31 |
| 15 | 40 | 83 | 0.48 |
| 16 | 31 | 87 | 0.36 |
| 17 | 22 | 90 | 0.24 |
| 18 | 25 | 90 | 0.28 |
| 19 | 29 | 89 | 0.33 |
| 20 | 30 | 89 | 0.34 |
| 21 | 33 | 83 | 0.40 |
| 22 | 28 | 82 | 0.34 |

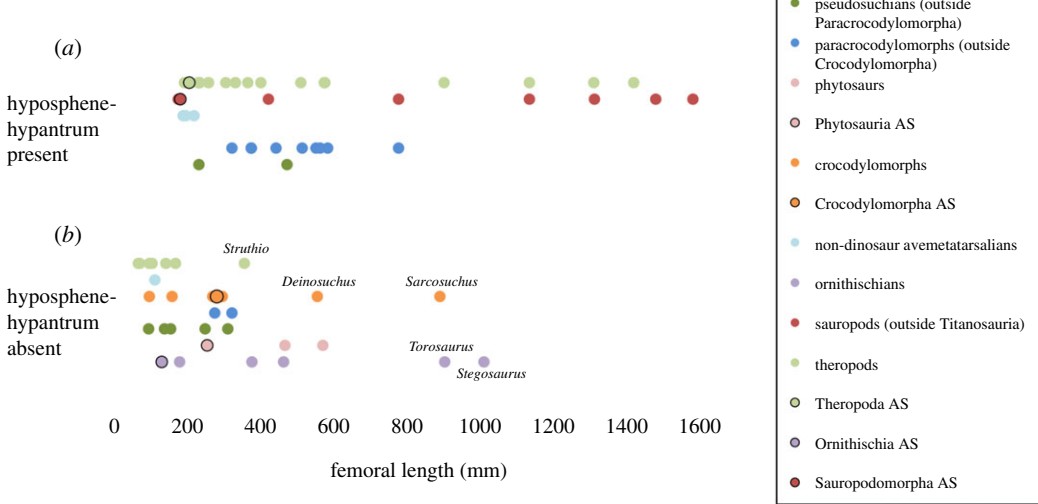

**Figure 3.** Plots showing the relationship between presence/absence of the hyposphene-hypantrum articulation and femoral length in all archosaur taxa included in table 1, including ancestral estimations of femoral length for some clades. Several of the taxa that are exceptions to this trend are labelled. Titanosaurs are not included in this plot. AS = ancestral state.

**Figure 4.** Body size logistic regression plots for (*a*) Pseudosuchia and (*b*) Avemetatarsalia, illustrating the threshold femoral lengths for presence of the hyposphene-hypantrum articulation.

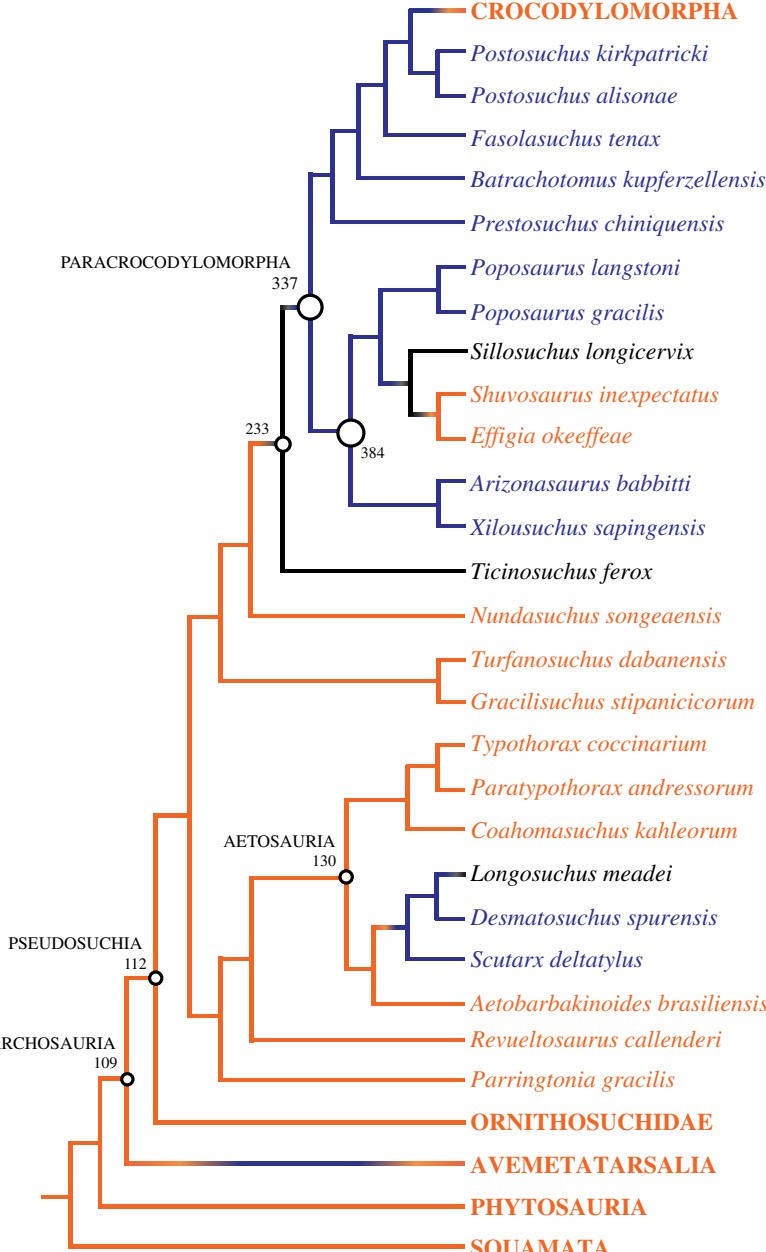

**Figure 5.** Phylogenetic tree of Pseudosuchia (from [25,29,61,64,65,67,72]) with the presence (blue), absence (orange), or ambiguity (black) of the hyposphene-hypantrum articulation mapped on and ancestral state femoral length reconstructions (from [11]) illustrated at several major clades.

There is a close relationship between larger body size and presence of the hyposphene-hypantrum articulation in pseudosuchian archosaurs (Nagelkerke $R^2$: 0.796, McFadden's $R^2$: 0.660), and these data were significant ($p$-value: <0.001) (figure 4$a$). The threshold size at which the hyposphene-hypantrum is present in pseudosuchians is about 212–300 mm femoral length (table 1; figures 4$a$ and 5). Pseudosuchians, with the exceptions of crocodylomorphs and a few others (see below), with a femoral length greater than 300 mm almost always had hyposphene-hypantrum articulations. The pseudosuchian taxon that has a hyposphene-hypantrum with the shortest femoral length is *Mandasuchus tanyauchen* (holotype FL = 212 mm [25]).

Similarly, large body size and the presence of hyposphene-hypantrum articulations are correlated in avemetatarsalians (Nagelkerke $R^2$: 0.789, McFadden's $R^2$: 0.741, significant at $p \leq 0.001$) (figure 4$b$). The range of femoral length at which the hyposphene-hypantrum evolved in avemetatarsalians is restricted to 130–170 mm (table 1; figures 4$b$ and 6). The taxa with the shortest femoral lengths that have the hyposphene-hypantrum are *Asilisaurus kongwe* (femoral length = 177 mm [73,74]) and *Teleocrater*

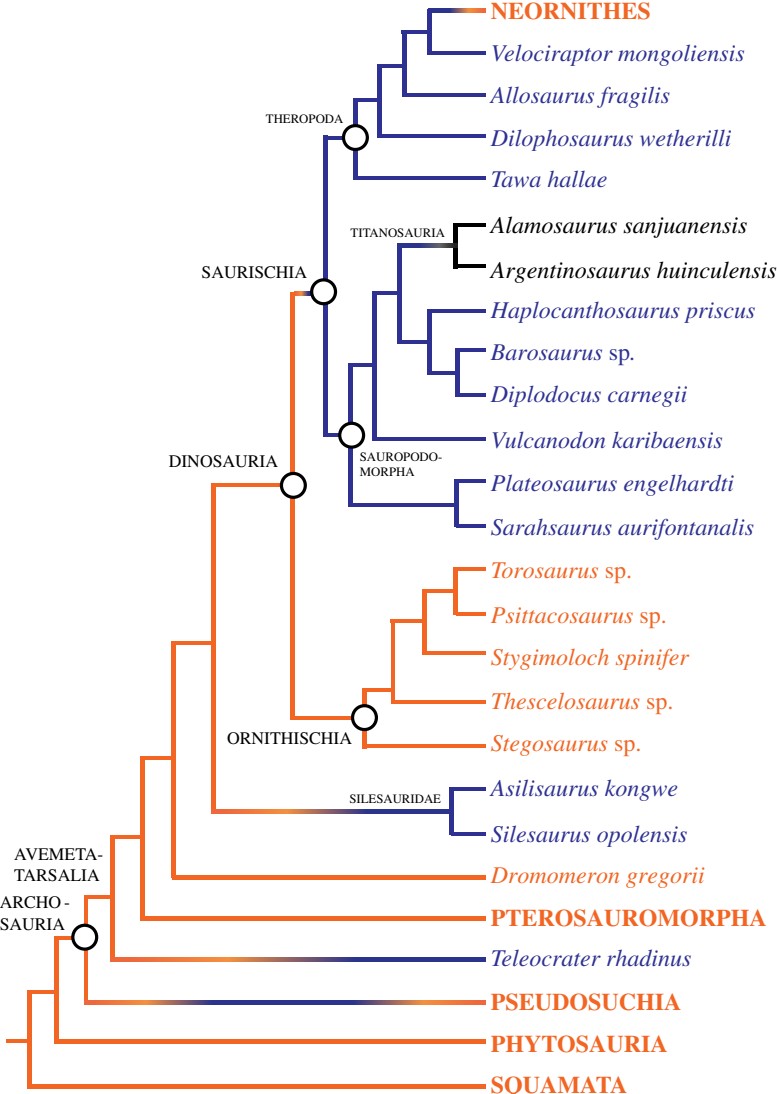

**Figure 6.** Phylogenetic tree of Avemetatarsalia (from [6,7,31,44,48,54,58,62,63]) with the presence (blue) and absence (orange) of the hyposphene-hypantrum articulation mapped on.

*rhadinus* (femoral length = 170 mm [31,75]), and these two taxa are among the earliest diverging avemetatarsalians currently known. Avemetarsalians that have a femoral length greater than about 170 mm almost always had hyposphene-hypantrum articulations in the trunk vertebrae, except ornithischians and some titanosaurs (see below).

The hyposphene-hypantrum is absent in the two archosaurian crown groups (Crocodylia and Aves), and it was lost in several extinct archosaur clades, including some poposauroid pseudosuchians (figure 5) and maniraptoran theropods (figure 7). It was lost in the clade containing *Effigia okeeffeae* and *Shuvosaurus inexpectatus*, which are sister taxa and have femoral lengths of 301 and 255 mm, respectively (figure 5). These body sizes are within the threshold range for presence of the hyposphene-hypantrum in pseudosuchians of 212–300 mm. The presence or absence of the hyposphene-hypantrum in the sister taxon to *Effigia* + *Shuvosaurus*, *Sillosuchus longicervix* (PVSJ 85), is ambiguous because the only known specimen is poorly preserved [24]. The maniraptorans that lack the hyposphene-hypantrum all fall below the lower bound of the threshold range of body size for presence of the hyposphene-hypantrum in avemetatarsalians of 130–170 mm (e.g. *Mononykus olecranus*, *Parvicursor remotus*, *Microvenator celer*, *Mahakala omnogovae*, *Rahonavis ostromi*) (figure 7).

Our analysis that used the maximum length of zygapophyses as a proxy for articular surface area in extant and extinct archosaurs showed that among members of crown Crocodylia and crown Aves, relative articular surface area is roughly the same, when corrected for body size by dividing by centrum height (figure 8, tables 2 and 3). Additionally, in the extinct archosaur taxa we analysed that possess the

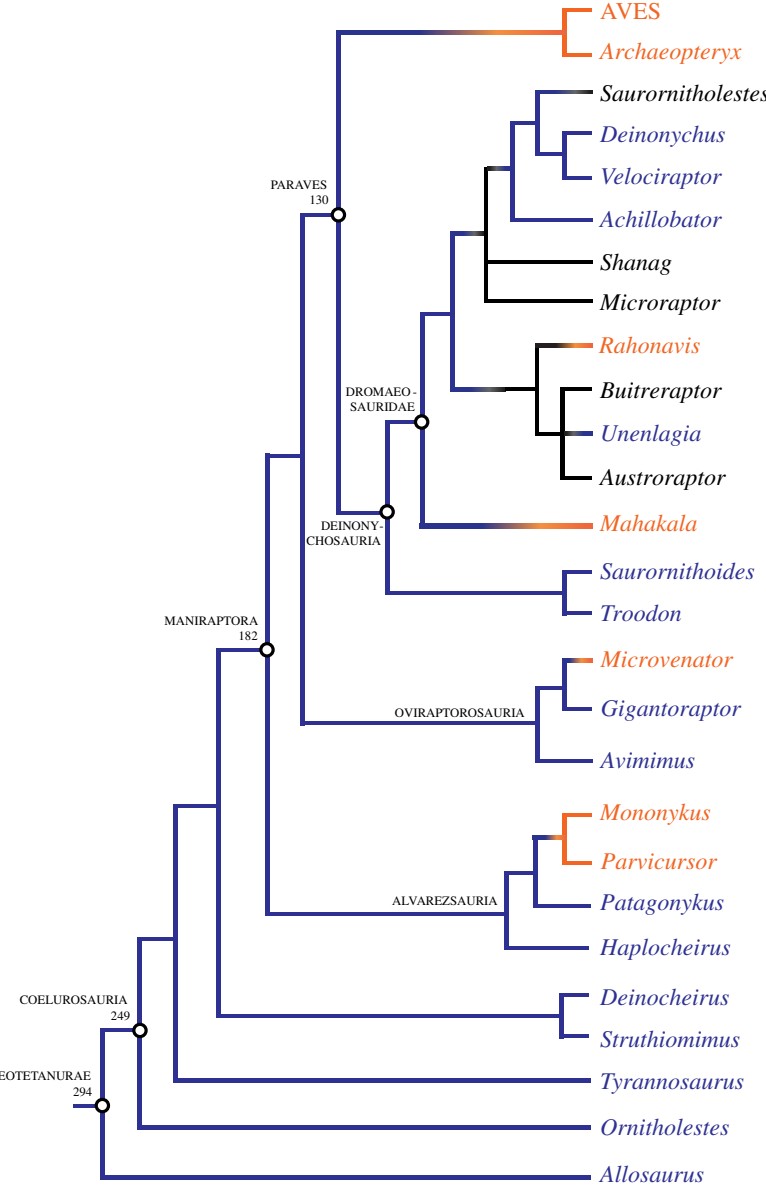

**Figure 7.** Phylogenetic tree showing the loss of the hyposphene-hypantrum in Neotetanurae theropods, with ancestral state femoral length reconstructions (from [44]) illustrated at several major clades.

hyposphene-hypantrum (e.g. *Dilophosaurus*, *Desmatosuchus*) (figure 9, table 4), the total articular surface length (postzygapophyseal maximum length + hyposphene maximum length) divided by centrum height (figure 9, blue data points) plotted more closely with the extant data, whereas the measurement of only the postzygapophyseal maximum length divided by centrum height (figure 9, orange data points) plotted below the extant data. The postzygapophyseal length divided by centrum height of extinct taxa that do not have hyposphene-hypantrum articulations plotted closely with the extant data (figure 9). However, these relationships are merely anecdotal and we do not have enough data to perform a statistical test with enough power to be meaningful. We think this is an interesting pattern worth investigating further, but it is beyond the present study.

## 4. Discussion

### 4.1. The phylogenetic distribution of the hyposphene-hypantrum articulation

The hyposphene-hypantrum articulation appears independently within Archosauria at least four times and is lost independently at least three times (figures 5–7). In all taxa in which the hyposphene-hypantrum is

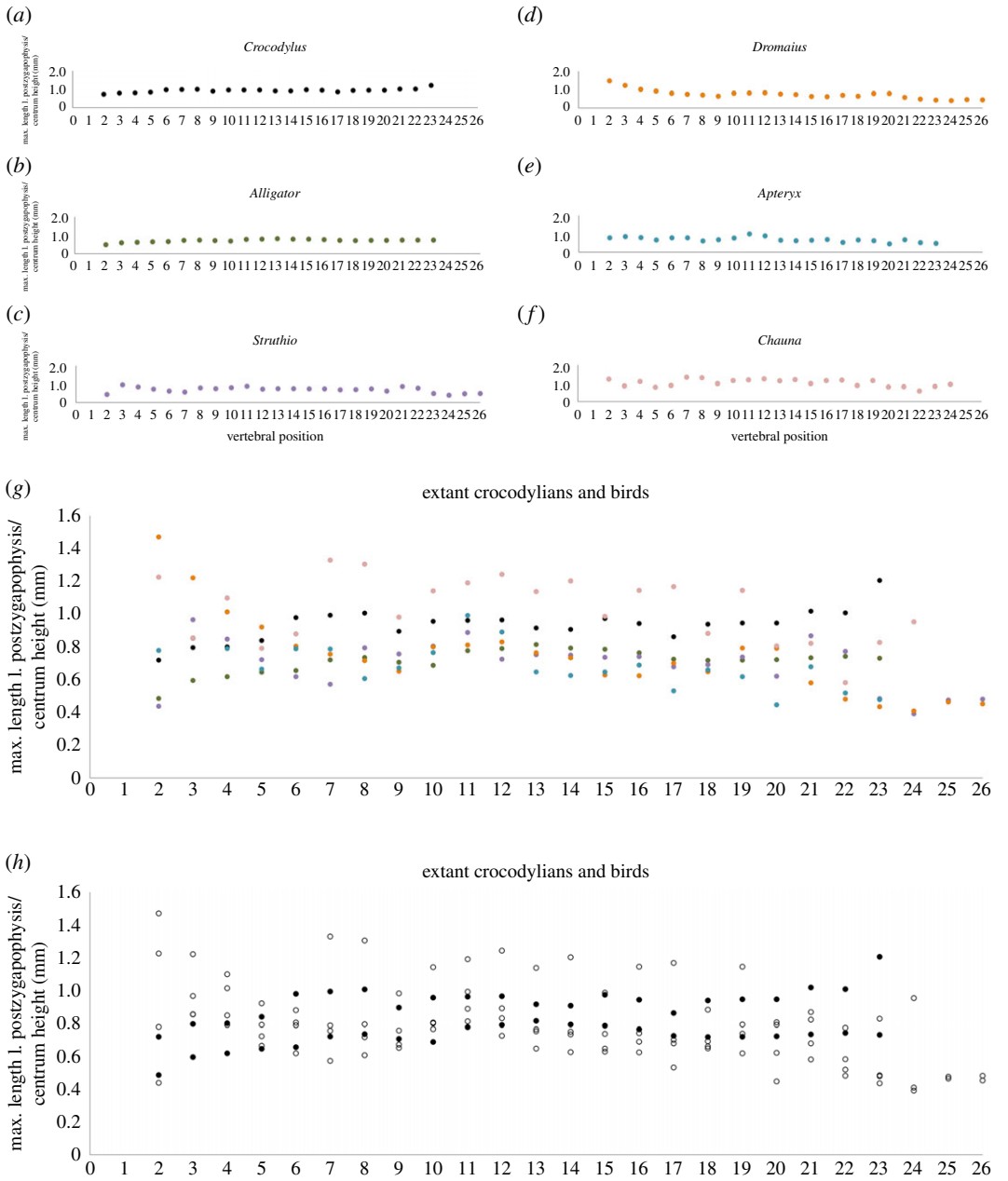

**Figure 8.** Plots showing the maximum length of postzygapophyseal articulation (proxy for surface area articulating between consecutive vertebrae) corrected for body size along the length of the presacral vertebral columns of extant crocodylians, (a) *Alligator mississippiensis* and (b) *Crocodylus acutus*, and extant birds, (c) *Struthio camelus*, (d) *Dromaius novaehollandiae*, (e) *Apteryx australis*, and (f) *Chauna chavaria*. All those data points are plotted together in (g), with the colours matching those in the above data plots. In (h) the crocodylian taxa are illustrated with filled-in data points while the bird taxa data points are outlines.

present, the articulation structures are similar morphologically and appear in the same position in the skeleton. Therefore, we determine that within Archosauria the ability for a taxon to express the hyposphene-hypantrum articulation is a deep (transformational) homology, and where it appears independently in some clades is considered taxically a convergence. The distribution of the hyposphene-hypantrum within Archosauria is especially intriguing because the presence or absence of the articulation is highly correlated with body size, rather than strict phylogenetic legacy (figures 4–7). This relationship between body size and the presence or absence of the hyposphene-hypantrum appears to be unique to Archosauria, because outside Archosauria, the hyposphene-hypantrum articulation is almost always absent, regardless of body size. The only known exception is stem archosaur *Azendohsaurus*

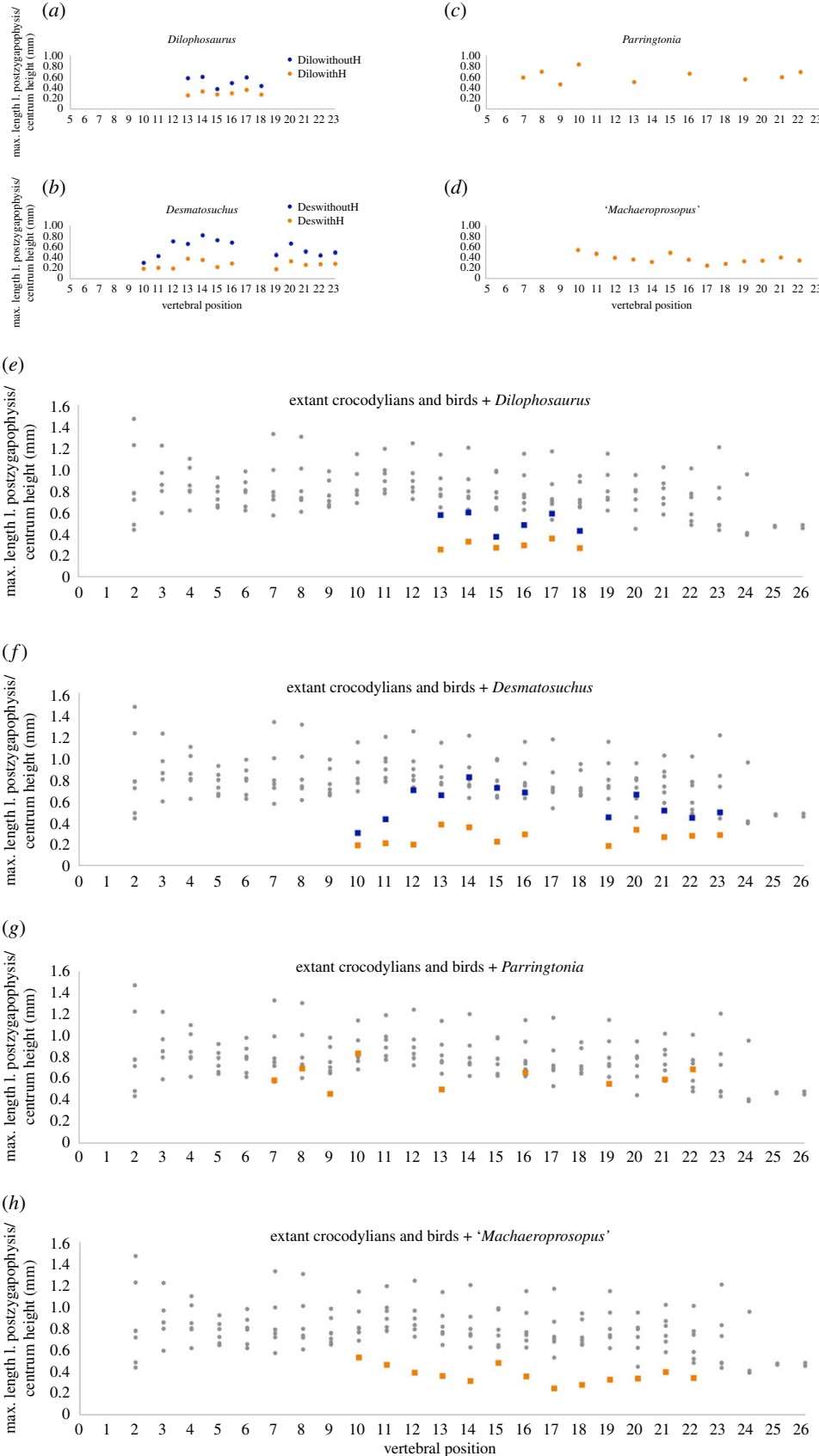

**Figure 9.** (*Caption overleaf.*)

**Figure 9.** (Overleaf.) Plots showing the maximum length of postzygapophyseal articulation (proxy for surface area articulating between consecutive vertebrae) corrected for body size along the length of the presacral vertebral columns of four extinct archosaurs, (a) the theropod dinosaur *Dilophosaurus wetherilli*, (b) the aetosaur *Desmatosuchus spurensis*, (c) the pseudosuchian archosaur *Parringtonia gracilis*, and (d) the phytosaur '*Machaeroprosopus*' *zunii*. For the plots of two taxa that possess the hyposphene-hypantrum articulation (a,b), we plotted both the maximum postzygapophyseal length divided by centrum height as well as maximum postzygapophyseal length plus maximum hyposphene length divided by centrum height to illustrate the increased relative surface area of articulation when taking into account the articular surface the hyposphene-hypantrum provides. In (e), (f), (g) and (h), the fossil taxa from (a), (b), (c) and (d), respectively, are plotted against the extant taxa data from figure 7 (grey). We notice that for taxa that possess the hyposphene-hypantrum, the more inclusive measurement of articular length (blue data points) plot more closely with the extant data, while the less inclusive measurement (orange data points) plot below the extant data. Extinct taxa that do not have hyposphene-hypantrum articulations plot closely with the extant data.

*madagaskarensis* (FL = 205 mm, FMNH PR 2779, [76]), which has a hyposphene-hypantrum articulation in only one vertebra in the anterior trunk [76].

Although many stem archosaurs (outside crown Archosauria) are small and fall below the threshold found in Archosauria (e.g. *Vancleavea campi*: FL = 87 mm, PEFO 2427, GR 138; *Euparkeria capensis*: FL = 65 mm, SAM 5867, SAM 6047B; *Tropidosuchus romeri*: FL = 64 mm, PVL 4601), other stem archosaur taxa that lack the hyposphene-hypantrum are markedly larger; they have femoral lengths above the 130–170 mm threshold range (e.g. *Smilosuchus gregorii*: FL = 545 mm, USNM 18313; *Machaeroprosopus pristinus*: FL = 444 mm, UCMP 27235; *Erythrosuchus africanus*: FL = 466 mm, SAM 905; *Trilophosaurus buettneri*: FL = 205 mm, TMM 31025-140). These body sizes are comparable to some of the largest pseudosuchian archosaurs that have the hyposphene-hypantrum articulation. Recent phylogenetic analyses have placed Phytosauria as the sister group of crown Archosauria [64]; however, if Phytosauria is included in Archosauria, within Pseudosuchia, as suggested by previous work (e.g. [77–80]), that would be an example of an archosaur clade that never evolved the hyposphene-hypantrum. Otherwise, from our observations the hyposphene-hypantrum articulation evolved in most of the trunk vertebrae only within Archosauria. This suggests that the common ancestor of Archosauria may have had the ability to form the articulation, given the wide distribution of the feature in both major branches of the clade. The hyposphene-hypantrum is therefore a deep homology elicited by circumstances of size, ecology and adaptation [81].

Based on femoral length (FL) data, we inferred a minimum body size threshold (FL = 230 mm in pseudosuchians, FL = 130 mm in avemetatarsalians) below which the hyposphene-hypantrum is not present (figure 4) and a maximum threshold (FL = 300 mm in pseudosuchians, FL = 170 mm in avemetatarsalians) above which the hyposphene-hypantrum is present. Using published reconstructed femoral lengths at important nodes within the archosaur phylogeny [11,44], we were able to make further predictions about gain or loss of the hyposphene-hypantrum throughout Archosauria. Our observations show that the hyposphene-hypantrum evolved in both pseudosuchian and avemetatarsalian members of Archosauria at body sizes with femoral lengths above 212 and 130 mm, respectively; however, the reconstructed ancestral state for the femoral length of Archosauria is approximately 109 mm, and this is below the minimum threshold for hyposphene-hypantrum presence for both pseudosuchians and avemetatarsalians. The presence of the hyposphene-hypantrum was ambiguously optimized at the base of Archosauria, but the body size data (based on ancestral femoral length) suggest that the articulation was absent in the common ancestor of archosaurs. Therefore, we interpret that the evolution of the articulation in Pseudosuchia and Avemetatarsalia was convergent.

The hyposphene-hypantrum is present in several members of Poposauroidea, a clade within and near the base of Paracrocodylomorpha (figure 5) (e.g. *Arizonasaurus babbitti*, MSM 4590, [82]; *Xilousuchus sapingensis*, IVPP V6026, [54]; *Poposaurus langstoni*, TMM 31025-257, TMM 31025-1261.1, [23,83,84]; *Poposaurus gracilis*, TTU P-10419, [84]), but its presence is uncertain in *Sillosuchus longicervix* [24] because the articulation structures are not preserved in the only known specimen (PVSJ 85). Another pseudosuchian in which the hyposphene-hypantrum articulation is uncertain is the sister taxon to Paracrocodylomorpha, *Ticinosuchus ferox* (PIZ T 2817, [85,86]). That specimen is preserved on a flattened slab with compressed vertebrae that remain in articulation, so it is difficult to confirm the presence of a hyposphene on the posterior aspects of the vertebrae.

All known Triassic loricatan paracrocodylomorphs possess the hyposphene-hypantrum articulation (e.g. *Fasolasuchus tenax*, PVL 3850, [87]; *Batrachotomus kupferzellensis*, SMNS 80296, [88]; *Prestosuchus chiniquensis*, UFRGS-PV-0156-T, [89]; *Saurosuchus galilei*, PVSJ 32, [90]; *Mandasuchus tanyauchen*, NHMUK PV R6792, [74]). All these taxa, except the smallest known specimen of *Mandasuchus*

*tanyauchen* (holotype FL = 212 mm [74]), have femoral lengths greater than 400 mm; however, among the three known individuals of *Mandasuchus tanyauchen*, the individual from which a complete femur is known (NHMUK PV R6792) is not the largest reported of the taxon [74]. Therefore, it is clear that *Mandasuchus tanyauchen* grew larger than the holotype.

This hints at a relationship between ontogeny and the presence or absence of the hyposphene-hypantrum in that the adult or near maximum body size may be important in determining the presence of absence of the articulation. The *Mandasuchus tanyauchen* specimen we report in this study, the holotype for the species, is our only instance where we knowingly reported a femoral length for a taxon from an individual that is not the maximum size known for the taxon. If the holotype of *Mandasuchus tanyauchen* is a skeletally immature individual, this is evidence for the presence of the hyposphene-hypantrum articulation at a young age (i.e. smaller), before the articulation was needed for biomechanical support for large body size. This further strengthens our hypothesis that the hyposphene-hypantrum was necessary for growth to large body size, because if it is present in small juveniles, it probably could have been present in small taxa. Unfortunately, few pseudosuchians (and Mesozoic avemetatarsalians) are represented by a growth series with associated trunk vertebrae and this cannot be explored further at the present time.

Within the pseudosuchian clade Aetosauria, the hyposphene-hypantrum is present only in taxa with a largest reported femoral length above the threshold range (230–300 mm) for pseudosuchians. The largest aetosaurs (e.g. *Desmatosuchus spurensis*, MNA V9300 [22]; *Scutarx deltatylus*, PEFO 34045 [91]; *Longosuchus meadei*, TMM 31100-448, TMM 31100-452) possess the hyposphene-hypantrum in their trunk vertebrae. The articulation is absent in the closest relatives of aetosaurs (e.g. *Revueltosaurus callenderi* and *Parringtonia gracilis*) and the smallest aetosaur taxa (e.g. *Coahomasuchus kahleorum*, TMM 31100-437, [72]; *Aetobarbakinoides brasiliensis*, CPE2 168, [28]) (figure 5). This supports our hypothesis that the appearance of the hyposphene-hypantrum articulation is generally related to large body size and is present in most pseudosuchian clades (see below) that have body sizes over the 300 mm femoral length threshold.

Within Avemetatarsalia and outside Dinosauria, two taxa have the hyposphene-hypantrum articulation, and they represent the smallest body sizes within Archosauria to possess the articulation. These are the silesaurids *Asilisaurus kongwe* (FL = 177 mm [70,71]) and *Silesaurus opolensis* (FL = 200 mm [92]) (figure 6). In addition, the small avemetatarsalian *Teleocrater rhadinus* (FL = 170 mm [31,75]) has a hyposphene-hypantrum. Consistent with the pattern, the hyposphene-hypantrum articulation is absent in *Dromomeron romeri* (S.J.N. 2018, personal observation), which is smaller (FL = 95 mm [32]) than the aforementioned taxa and has a femoral length below the avemetatarsalian threshold (figure 6).

Among the three major clades of dinosaurs (i.e. Ornithischia, Sauropodomorpha and Theropoda), the hyposphene-hypantrum articulation is only present in Sauropodomorpha and Theropoda; it is absent in all ornithischians, regardless of body size (figure 6; table 1). The ancestral femoral length reconstruction for Saurischia (i.e. Sauropodomorpha + Theropoda) is 156 mm [11], which is above but close to the minimum femoral length at which the hyposphene-hypantrum is seen in other avemetatarsalians (130 mm), so we would predict, in agreement with the previous findings of [77], that the articulation is at least plesiomorphic for Saurischia. Other phylogenetic hypotheses, such as that of Baron *et al*. [93] for early dinosaurs (i.e. ornithischians are more closely related to theropods than either is to sauropodomorphs), do not change this result given the number of state changes (i.e. presence or absence) and femoral length.

## 4.2. Exceptions within Archosauria (extinct lineages)

A confounding scenario in the distribution of the hyposphene-hypantrum in archosaurs is the reported loss of those structures in titanosaurs, taxa well above the body size threshold (figure 6) [21]. These are the largest of all dinosaurs [3], and the hyposphene-hypantrum articulation is lost near the base of the clade (figure 6). The earliest diverging members of Titanosauria, *Andesaurus delgadoi* [94] and *Phuwiangosaurus sirindhornae* (SM K11-0038, [95]), both clearly have the articulation, but Apesteguia [21] demonstrated that later diverging lineages of titanosaurs lack a hyposphene-hypantrum. We have personally observed one titanosaur vertebra, *Alamosaurus sanjuanensis* (TMM 41891-1) and agree that it does not have the hyposphene-hypantrum articulation using the definition stated above. Apesteguia [21] states that the late diverging titanosaurs *Argentinosaurus huinculensis* and *Epachthosaurus sciuttoi* have 'hyposphenal bars', but these are not true hyposphene-hypantrum articulations, based on our definition. However, the original publication of *Argentinosaurus huinculensis* [96] figures a trunk vertebra with a clearly defined and labelled hyposphene that does fit our definition. Furthermore,

although the structure in most titanosaurs does not fit our definition, the condition in some titanosaurs might be a highly derived modification of the hyposphene-hypantrum. For this study, we consider the hyposphene-hypantrum absent in titanosaurs, which is noteworthy because they are so large.

Because of their derived vertebral morphology and lack of a hyposphene-hypantrum articulation in the gigantic titanosaurs, we eliminated these taxa from our body size logistic regression for avemetatarsalians. We also exclude all ornithischian dinosaurs from our analyses because they are a major dinosaur clade that never evolved the hyposphene-hypantrum articulation, and therefore the absence of the articulation in later diverging, large ornithischians may be because their vertebrae have a derived morphology to support large body size in a different way than the hyposphene-hypantrum. For example, ornithischians have ossified tendons in their vertebrae [97–99], which may be a secondary mechanism for vertebral bracing. The distribution of ossified tendons in Ornithischia and its relationship to larger body size should be investigated further; however, we choose not to include the clade in this study. Additionally, the biomechanical implications of having ossified tendons is also an interesting and noteworthy question that is needed to understand if they are an alternative mechanism for the same function (i.e. vertebral column bracing) provided by hyposphene-hypantrum articulations.

We also chose to exclude pterosaurs as a whole for this study because we could not confirm the presence or absence of the hyposphene-hypantrum in any of the specimens of early diverging pterosaurs. This is because most are preserved as flattened slabs and are so small that even in µCT it is virtually impossible to see between the vertebrae to confidently score presence or absence of the hyposphene-hypantrum articulation. Additionally, pterosaurs modified their vertebrae from the plesiomorphic morphology at relatively small sizes and secondarily became large, similarly to ornithischians and titanosaurs. No hyposphene-hypantrum articulation has ever been reported in a large pterodactyloid, and the articulation is undeniably absent in the 'small morph' of *Quetzalcoatlus* [100], although posterior dorsals are missing; however, no vertebrae are preserved in the 'large morph' (K. Padian 2019, personal communication).

## 4.3. Losses of hyposphene-hypantrum in crown Archosauria unequivocally

The absence of the hyposphene-hypantrum articulation in both crown groups of Archosauria (i.e. Crocodylia and Aves) is surprising given the deep history of this feature and that some extinct and extant members have femoral lengths that are higher (e.g. the crocodylian *Deinosuchus riograndensis* and the avian *Dinornis novaezelandiae*) than threshold size for presence of the hyposphene-hypantrum within more early diverging members of Archosauria. The absence of the hyposphene-hypantrum in the Crocodylia and Aves can be traced well outside the crown to members of the stem lineages (figures 5 and 7). The earliest members of Crocodylomorpha that diverged from other pseudosuchians generally reduced their body size in the Late Triassic–Early Jurassic below the threshold range for the presence of a hyposphene-hypantrum (FL = 212–300 mm) (e.g. *Sphenosuchus acutus*: reconstructed FL = 140 mm [11,101]; *Hesperosuchus agilis*: FL = 140 mm [102]; *Terrestrisuchus gracilis*: FL = 80 mm [103]). This smaller body size was retained through the origin of Crocodyliformes [11].

Not only did crocodylomorphs phyletically reduce their body sizes, but as they evolved smaller bodies, clades in this group also shifted their ecologies from fully terrestrial to fully or semi-aquatic freshwater environments [104,105]. The crocodylomorph clade Thalattosuchia even became highly marine-adapted [106,107]. In addition to changing their ecology, crocodylomorphs also modified their vertebral morphology [108] after the loss of the hyposphene-hypantrum, and once the group has reduced average size phyletically. However, it is difficult to tell the order in which changes in ecology and changes in morphology occurred.

One notable shift in vertebral morphology within crocodylomorphs is the transition from the ancestral pseudosuchian condition of amphicoely to procoely. Procoelous vertebrae have a deep anterior cotyle and a well-developed posterior condyle that fits into the complementary cotyle of the subsequent vertebra. Procoely has historically been considered a feature unique to Eusuchia (the clade containing crocodylians and other morphologically 'modern' crocodylomorphs). However, as our sample of non-eusuchian crocodylomorphs have improved, it has become apparent that procoely evolved several times within crocodylomorphs. Amphicoely is the ancestral state for crocodylomorphs, and crocodyliforms and most non-eusuchian neosuchians (e.g. *Goniopholis simus*, [108]) maintain this condition. However, some neosuchians near the origin of Eusuchia have procoelous or weakly procoelous vertebrae (see [109,110]) (e.g. *Isisfordia duncani* [111]; *Shamosuchus djadochtaensis* [108,109]; *Wannchampsus kirpachi* [112]; *Theriosuchus pusillus* [113]; *Pachycheilosuchus trinquel* [114]).

After the transition from terrestrial to aquatic habitat and the evolution of widespread procoely in crocodyliforms, some taxa evolved large body sizes that are higher than the pseudosuchian threshold for presence of the hyposphene-hypantrum (e.g. *Deinosuchus riograndensis*: FL = 530 mm, TMM 43632-1; *Sarcosuchus imperator*: FL = 860 mm [29]) (table 1), but these taxa all lack the articulation. The absence of the hyposphene-hypantrum articulation in these large crown-group crocodylians may be related to either their difference in ecology or procoelous vertebral morphology, or a combination of the two. The evolution of procoely may have allowed vertebral bracing [115] in a novel way that rendered the hyposphene-hypantrum unnecessary for growth to large body sizes.

In avemetatarsalians, the hyposphene-hypantrum was lost within the clade Theropoda (figure 7), at the level of birds and their relatives. However, many large-bodied theropods have the articulation, and this has been extensively reported (*Tyrannosaurus rex*: FL = 1273 [40,116]; *Allosaurus fragilis*: FL = 872 [25,41]; *Deinocheirus mirificus*: FL = 1381 [117]; *Struthiomimus altus*: FL = 486 mm [40]) (table 1). In fact, almost all non-avialan theropods possess the hyposphene-hypantrum even at relatively small body sizes, but those with the articulation still have femoral lengths at or above the minimum threshold for presence of the articulation (130–170 mm) (e.g. *Tawa hallae*: FL = 174 mm [118]; *Ornitholestes hermanni*: FL = 210 mm [119]; *Velociraptor mongoliensis*: FL = 238 mm [42]). Within the theropod clade Oviraptorosauria, the hyposphene-hypantrum is present in the largest members (e.g. *Gigantoraptor erlianensis*: FL = 1100 mm [53]; *Avimimus portentosus*: FL = 186 mm [52]) but absent in smaller-bodied taxa (*Microvenator celer*: FL = 124 mm [43]) (figure 7). The articulation is present in large members of Dromaeosauridae (e.g. *Unenlagia comahuensis*: FL = 380 mm [56]; *Achillobator giganticus*: FL = 550 mm [59]; *Deinonychus antirrhopus*: FL = 440 mm [120]; *Velociraptor mongoliensis*: FL = 238 mm [42]) and notably absent in the smallest known dromaeosaurids, *Mahakala omnogovae* (FL = 79 mm [47]) and *Rahonavis ostromi* (FL = 88 mm [121]) (figure 7). Although *Rahonavis ostromi* was reported to have a hyposphene-hypantrum articulation [121], through direct examination of the holotype (UA 8656) we determined that its morphology does not satisfy the criteria for presence of the hyposphene-hypantrum (*sensu* [23]). Due to poor preservation (e.g. vertebrae are articulated, specimen is preserved as a flattened slab instead of in three dimensions, specimen is weathered and/or broken), the presence of the hyposphene-hypantrum is ambiguous in many dromaeosaurids (e.g. *Buitreraptor gonzalezorum*: FL = 145 mm [55]; *Shanag ashile*: FL = 55 mm [58]; *Austroraptor cabazai*: FL = 560 mm [54]; *Microraptor zhaoianus*: FL = 75 mm [57]; *Saurornitholestes langstoni*: FL = 225 mm [122]). In Alvarezsauria, the larger and early diverging members of the clade (e.g. *Patagonykus puertai*: FL = 285 mm [50]; *Haplocheirus sollers*: FL = 214 [51]) have the hyposphene-hypantrum, and the articulation is absent in the smaller and later diverging members that drop below the 130–170 mm threshold (e.g. *Mononykus olecranus*: FL = 150 mm [48]; *Parvicursor remotus*: FL = 50 mm [49]) (figure 7). Interestingly, these two taxa also have procoelous vertebrae, a condition not seen in their close relatives, *Patagonykus puertai* [50] and *Haplocheirus sollers* [51].

In Aves, or crown group birds, the hyposphene-hypantrum articulation is completely absent in all taxa even though some extinct (e.g. *Dinornis novaezelandiae*) and extant (e.g. *Struthio camelus*) members have femoral lengths above the avemetatarsalian femoral length threshold of 130–170 mm. The articulation is lost just outside Aves at the base of the clade Avialae (e.g. *Archaeopteryx lithographica*: FL = 52.6 mm [60]) (figure 7). Avialae has a reconstructed ancestral femoral length of 83.4 mm [44], which is below the minimum avemetatarsalian threshold femoral length (130–170 mm) for presence of the articulation. Both lack of hyposphene-hypantrum articulations in all known avialans and small ancestral femoral length support our conclusion that the articulation is lost outside or at the base of Avialae.

More inclusively, the losses and gains of the hyposphene-hypantrum articulation in Paraves are more difficult to pinpoint with our taxonomic sampling and confidence in identifying the presence or absence of the articulation in paravians (figure 7). Among the clades included in Paraves, several dromaeosaurids have it (e.g. *Velociraptor mongoliensis* [42]; *Deinonychus antirrhopus* [120]; *Unenlagia comahuensis* [56]; *Achillobator giganticus* [59]), some troodontids have it (e.g. *Troodon formosus*: FL = 310 mm, [45]; *Saurornithoides mongoliensis*: FL = ?, [123]), but all avialans lack it. Additionally, the reconstructed ancestral femoral length of Paraves is 130 mm [44], which is at the absolute minimum of the threshold (130–170 mm) for the presence of the feature. Confounding this further, many of the early diverging members of Dromaeosauridae and Troodontidae are small-bodied but condition of preservation does not allow confirmation of the presence or absence of the hyposphene-hypantrum (discussed above). These factors at the base of Paraves suggest several scenarios for the loss or gain of the hyposphene-hypantrum in this clade: (i) the common ancestor of Paraves had the hyposphene-hypantrum and it was lost in avialans; (ii) it was absent in the common ancestor of Paraves and gained independently in Deinonychosauria; (iii) it was absent in the common ancestor of Paraves and the common ancestor

of Deinonychosauria and gained independently in both dromaeosaurids and troodontids (figure 7), which both undergo a series of body size increases during their evolution [7].

Although it is uncertain where exactly in the paravian phylogeny the hyposphene-hypantrum was lost, it is absent in all known taxa with body sizes below the minimum avemetatarsalian threshold femoral length (130–170 mm) (e.g. *Mahakala omnogovae* [47]; *Rahonavis ostromi* [121]; *Archaeopteryx lithographica* [60]) (figure 7). Immediately following this decrease in body size, avialans evolved powered flight (e.g. *Archaeopteryx lithographica*) [124–127]. In closely related but later diverging non-avian avialans, there are vertebral modifications that differ markedly from the ancestral non-avialan theropod condition, including the evolution of heterocoelous centra articulations. Heterocoely appears just outside the clade Ornithurae in *Patagopteryx deferrariisi* [128]. *Patagopteryx deferrariisi* is a small (FL = 99 mm [128]) avialan and although it was flightless [128], this loss of flight was secondary, and the shift to heterocoelous vertebrae (the condition seen in all living birds) can be placed phylogenetically after the evolution of powered flight. Heterocoely could have provided increased vertebral bracing, similar to the proposed function of procoely in crocodylians [115]. In addition, avialans fuse many of their sacral vertebrae (including elements that were formerly free dorsals), which may also contribute to vertebral bracing. With the addition of these novel morphologies, the hyposphene-hypantrum articulation was probably no longer selected for in large-bodied members of Avialae because their vertebral column was adequately braced by alternative mechanisms.

# 5. Conclusion

In all living Archosauria, the hyposphene-hypantrum articulation is absent. However, the fossil record shows that it was once widespread and closely correlated with two factors: body size and the absence of an alternative vertebral bracing mechanism. The hyposphene-hypantrum appears to have been lost just before the origin of the archosaur crown clades, Crocodylia and Aves, and these losses are correlated with phyletic body size reduction and shifts in ecology. The articulation is independently lost in small-bodied members of earlier diverging, extinct clades across Archosauria. Our results support the hypothesis that the hyposphene-hypantrum provided biomechanical support for large bodies in extinct archosaurs, acting as a bracing mechanism similar to features such as procoely and heterocoely that evolved after phyletic body size reduction in the crown clades. It is likely that the hyposphene-hypantrum is the ancestral bracing system in the vertebral column for Archosauria and that Ornithischia, the major archosaurian clade that lacks the hyposphene-hypantrum but includes taxa with body sizes greater than the thresholds we saw for presence of the structure in Avemetatarsalia and Pseudosuchia, evolved ossified tendons as a secondary bracing mechanism to support large body size [97–99]. These novel intervertebral articulations that evolved in tandem with ecological changes, coupled with other skeletal changes (e.g. osteoderm construction in crocodylians, ankylosing of vertebrae in avialans), may have provided enough vertebral bracing to prevent the reacquisition of the hyposphene-hypantrum in crocodylians and birds, even at body sizes above our predicted thresholds at which the articulation appears necessary.

Data accessibility. All data and codes used in this study are included in either the tables accompanying this manuscript or in the electronic supplementary material submitted with the manuscript.

Authors' contributions. C.M.S. conceived and designed the experiments, performed the experiments, analysed the data, wrote the paper, prepared figures and tables, reviewed drafts of the paper; S.J.N. contributed reagents/materials/analysis tools, reviewed drafts of the paper.

Competing interests. We declare we have no competing interests.

Funding. C.M.S.: Welles Fund (UC Berkeley), Aubrey and Eula Orange Award (Virginia Tech).

Acknowledgements. We thank the following collections managers and scientists at their respective institutions for access to their collections, permission to take photographs for analysis, and discussion: W. Parker at the Petrified Forest National Park, K. Padian and P. Holroyd at The University of California Museum of Paleontology, C. Mehling at the American Museum of Natural History's FARB collection, A. Henrici at the Carnegie Museum, M. Brown and C. Sagebiel at the Vertebrate Paleontology Lab at the University of Texas at Austin, and J. Gillette and D. Gillette at the Museum of Northern Arizona. We thank the University of California Museum of Paleontology's Doris O. and Samuel P. Welles Research Fund for assisting with travel to their collection. We thank M. Stocker and J. Socha for discussion and assistance in editing this manuscript and the rest of the Virginia Tech Paleobiology and Geobiology Research Group (S. Zhao, C. Griffin, C. Colleary, K. Formoso, K. Koeller, M. Riegler) for useful discussion. We also thank A. Turner at Stony Brook University for discussion and assisting in editing this manuscript. Finally, we thank the

exceptional help and encouragement of K. Padian through this project. We thank the Virginia Tech Open Access Subvention Fund for their support in covering the article processing charge for this publication.

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
