## [Reviewer comments · Royal Society Open Science]

Review History

RSOS-180579.R0 (Original submission)

Review form: Reviewer 1 (Paul M. Barrett)

Is the manuscript scientifically sound in its present form?

No

Are the interpretations and conclusions justified by the results?

No

Is the language acceptable?

Yes

Is it clear how to access all supporting data?

Yes

Do you have any ethical concerns with this paper?

No

Have you any concerns about statistical analyses in this paper?

No

Recommendation?

Major revision is needed (please make suggestions in comments)

Comments to the Author(s)

General comments

Hyposphene/hypantrum articulations are commonly found in extinct archosaurs, but are absent from living representatives of the clade. The authors test the hypothesis that the acquisition of these accessory articulations was correlated with increases in body size (and the converse that they were lost during periods of subsequent miniaturisation) by comparing character optimisations of hyposphene/hypantrum presence/absence with ancestral state reconstructions of a body mass proxy (femoral length). They find a strong correlation between the two. However, although I am generally sympathetic to the author's case, as the presence of these articulations in larger-bodied taxa makes intuitive biomechanical sense, I think that the MS requires some additional work before it can be accepted.

The analyses presented are incomplete and would benefit from the inclusion of additional taxa (large ornithischians and pterosaurs would be obvious additions). In addition, it's unclear how ancestral body femoral lengths are derived and more clarity on the evenness of taxon sampling with respect to body size would make some of the assumptions implicit in the dataset clearer. In some ways, the taxon sample selected for the analysis looks biased in a way that would provide their preferred result and although the addition of other taxa might not overturn it, they might reduce its statistical significance or explanatory power. In addition, although the authors prefer to infer functional explanations to phylogenetic ones when examining the observed pattern of hyposphene/hypantrum presence/absence, it would be difficult to disentangle these effects as the functional changes optimise at key nodes within archosaur phylogeny (and there is obvious phylogenetic signal in the autapomorphic loss of these features, and their continued absence, in even large bodied ornithischians).

Some functional explanations for the losses of these articulations are not explored, but are likely to be important when considering the evolution of this feature in archosaurs as a whole. As the authors note, there are complex patterns of gain and loss, but by effectively ignoring ornithischians, pterosaurs and titanosaurs, they miss an opportunity to come up with a more holistic explanation. As noted below, the evolution of other vertebral bracing structures (e.g. ossified tendons in ornithischians) might have been key in explaining some of these patterns, but taxa that are not congruent with their main hypothesis are overlooked, whereas these could actually provide some circumstantial support for their thesis.

Detailed comments

p. 1, lines 26–28. All body masses provided should be in metric tonnes, not tons. Please correct throughout the MS.

p. 2, lines 16–17. Indian elephants generally don't get above 5 tonnes, whereas bull African elephants do get up to 7 tonnes, though this is still relatively rare. I'd suggest changing the species here as it's generally recognised that African bush (savannah) elephants (*Loxodonta*

africana) are the largest living terrestrial vertebrates. If you do decide to stick with Indian elephants, the correct spelling is *Elephas maximus*.

p. 2, lines 26–28. There's been work on the evolution of large body size in sauropods in particular, with Martin Sander's group producing a number of papers on links between sauropod biology and body size, as summarised in Sander et al. (2011) in *Biol. Rev.* and also in Klein et al. (2011) *Biology of the Sauropod Dinosaurs*.

p. 2, lines 29–31. You're right, but it would be useful to provide some examples of these skeletal changes with body size (e.g., relative robustness, etc.), otherwise this is a straw man. You list some of these changes below, but you could integrate these parts of the intro more closely.

p. 2, line 37. I'd suggest 'model system' rather than just 'model'.

p. 2, line 45. Also in terms of overall robustness, patterns of scaling, limb bone eccentricity, etc. Some of these things change ontogenetically with body size within species too.

p. 3, line 5. Give a couple of examples of smaller dinosaurs that lack these features.

p. 3, lines 9–10. It depends how you define 'small' and 'large'. *Teleocrater* and *Effigia* possess hyposphene/hypantra, but are both relatively small by archosaurian standards (the modal mass of a non-avian dinosaur is around 1 tonne and these taxa get nowhere near this size). They're both certainly a lot smaller than many ornithischians that lack these features. Also, zygosphenes and zygantara (which function to more closely integrate vertebrae, just in a slightly different way morphologically) are found in a numerous small-bodied taxa (snakes, lizards).

p. 3, line 26. To some extent, the absence of hyposphene/hypantrum articulations in large-bodied ornithischians already falsifies this hypothesis, at least in part, without needing to test it quantitatively.

p. 4, lines 40–47 and p. 5, lines 2–13. In addition to surveying taxa that are spread across the tree, thus sampling across phylogeny, did you also survey representative taxa from different size classes? Although the taxa and their femoral lengths are noted in a table, it might be good to include some kind of summary diagram that makes it clearer how even your sampling is across size classes and the tree to show your data are not biased towards particular clades or body sizes. For example, visual inspection of your list indicates that very small bodied taxa (e.g., small ornithischians, pterosaurs: femoral lengths <100 mm) are not as extensively sampled as larger-bodied taxa. I'm also surprised that pterosaurs were excluded: maybe the rationale for this should be mentioned? Finally, within the large-bodied taxon sample there are distinct phylogenetic skews: large-bodied ornithischian taxa are severely undersampled (two taxa), whereas there are 10 large-bodied sauropodomorphs (but no small ones, such as *Saturnalia*). In reality there's much more overlap in the sizes of non-avian dinosaurs that lack hyposphenes/hypantra with those that possess this feature than summarised in Figure 1B.

p. 5, lines 33–35. It's not clear how these ancestral sizes were calculated. Have all of the ancestral values you used been lifted from the literature or have you calculated these ancestral values yourselves? In either case, how were they calculated? Squared-change parsimony or something else? Also, if the ancestral sizes were taken from the literature these references mention only some theropods and pseudosuchians – how were the ancestral sizes calculated for members of the other clades you include?

p. 6, lines 6–18. You are not quantifying the surface area of the articulation here, but the length of the articulation (depending on the shape of the articular surface there could be wide variation in

the actual amount of contact for the same length). Would be better to change your terminology throughout the MS to talk about this measure of postzygapophyseal length as a proxy for articular area and to note the pros and cons of using this measurement rather than surface area per se.

p. 6, lines 6–18. You mention that you also gathered data on the hyposphenes, but don't mention how you got these measurements or whether they were also normalised in some way. Or do you include the measurement of the hyposphene within your overall measurement of postzygapophysis length? It's not clear how any the additional articular length from the hypotheses is being captured in your analyses.

p. 9, line 36. You could refer to Figure 1 here.

p. 10, lines 22–26. It would be interesting to see what happened if you added in data from a few more large-bodied ornithischians here. I suspect the R2 would go down and there would definitely be a much, much larger 'transitional' range in Fig. 1B for avemetarsalians.

p. 11, lines 15–54. The derived lithostrotian titanosaur *Opisthocoelicaudia* has rather nice dorsals and no trace of hyposphene/hypantrum articulations. I think it's disingenuous to regard their absence as 'ambiguous' in this clade when all of the available evidence suggests that they were genuinely absent. It would be better to regard this a real anomaly on the basis of current data rather than trying to imply that they might have been there, simply as their absence isn't congruent with the rest of your thesis on the relationship between body size and hyposphene/hypantrum presence/absence. Also, large ornithischians (e.g., multi-tonned hadrosaurs and ceratopsians, many of which are larger than some sauropods and all but the very largest theropods) lack this feature, but simply aren't included in your analysis. As a result, body size might be part of the problem, but it can't be the only factor, nor can it be that critical if the very largest archosaurs live without them.

p. 12, lines 3–22. It would be interesting to include titanosaurs in these regressions to see what effect they have on the overall results and to compare the regressions with and without them. At the moment, the analyses are being stacked to make body size look more important by removing large taxa that lack these features (titanosaurs, ornithischians). This isn't a thorough test of the hypothesis. You are making an a priori judgement about the role of phylogeny here also, rather than testing it, although elsewhere in the paper you say phylogeny doesn't have a strong effect. Quite a bit of circular reasoning.

p. 12, lines 43 onward. Again, I'm not sure how you quantified hyposphene size here.

p. 14, lines 53 onward. Figures 2 and 3 suggest that the presence of hyposphene/hypantra is clearly convergent on the basis of character optimisations, without the need to invoke body size as another factor, and that it was not present in the archosaur common ancestor, contrary to the statement in the text.

p. 16, lines 31–45. Early ornithischians are comparable in size to *Tawa* and *Asilisaurus* in femoral length, but lack hyposphenes/hypantra. Interesting that other dinosaurs gain these features at small body sizes, but ornithischians don't. This has to be phylogenetic signal, as you suggest in passing elsewhere, but is worthy of more comment.

p. 17, lines 3–8. How would this optimisation change if you used an alternative dinosaur phylogeny that included either Ornithoscelida or Phytodinosauria (e.g. Baron et al., 2017; Langer et al. 2017; Parry et al. 2017)? If Ornithoscelida a real entity this suggests that these features are primitive for Dinosauria (and then lost in Ornithischia), not just a saurischian feature.

pp. 20–22. A number of the same measurements are repeated several times in successive paragraphs, which doesn't seem necessary.

pp. 22–24. It's interesting that you don't mention two other obvious bracing systems of archosaurs: osteoderms and ossified tendons. It's been suggested that the osteoderms of crown crocodylians act as a bracing system (and they lack hyposphenes) and also that the ossified tendons present in all ornithischians also provided this function (and they lack hyposphenes). The lack of these features in these taxa could reflect the acquisition of these other bracing systems also. It seems like something that would be important to consider in ornithischians especially, but it's not mentioned here at all. Miniturisation might not have been necessary in the loss of this system at all – it might simply have been replaced by other features at larger body sizes.

Figs 2–5. You include Squamata and Phytosauria in your trees, but there are no data on body size or the presence/absence of hyposphenes in these animals in your data tables. These data should be included. Also, I'd argue that squamates have something very similar to these features, even though the name differs. If you think that the zygantra/zygosphenes of squamates are not homologous with hyposphene/hypantra I'd be tempted to exclude them from your analysis here.

Figs 2–5. I'm unclear why some of the lineages (e.g. Avemetatarsalia in Fig. 2 or Pseudosuchia in Fig. 3) have different colours at different points along their lengths. If using maximum parsimony surely there should simply be an extrapolation of the ancestral state for that clade along the entire length of the branch from the tip to the node. I understand that this might be to capture some information on the presence/absence of the feature within each clade each time this is done, but as you're really only interested in the condition at the base of each clade as an outgroup in each case, adding this extra nuance in just makes it look confusing (and the nuances within each clade are captured in the more expanded trees you have for Pseudosuchia/Avemetatarsalia in successive figures anyhow).

No list of institutional abbreviations is provided for the various specimens listed in the MS and the tables.

Paul M. Barrett

Review form: Reviewer 2

Is the manuscript scientifically sound in its present form?

Yes

Are the interpretations and conclusions justified by the results?

Yes

Is the language acceptable?

Yes

Is it clear how to access all supporting data?

Yes

Do you have any ethical concerns with this paper?

No

Have you any concerns about statistical analyses in this paper?

No

Recommendation?

Accept with minor revision (please list in comments)

Comments to the Author(s)

This is a well-written, clean manuscript that looks at the the evolution of the hyposphene-hypantrum articulations in archosaurs, an important vertebral accessory articulation that was once thought to be restricted to dinosaurs. The authors argue that the presence of this structure is convergent between groups, and mostly tied to large body size; however, its lack in large bodied groups such as the ornithischian dinosaurs, as well as crown group archosaurs suggest that there is some phylogenetic significance as well. This conflict is addressed by the hypothesis that large animals require vertebral bracing and these articulations served this purpose, but that in groups lacking them they accomplished this bracing in other ways. I feel this argument is a little weak, for reasons found in the following comments, but overall the presence in larger individuals does appear to be for the most part supported.

Other comments:

Page 3, line 8: The possibly first recognition of these structures in a pseudosuchian archosaur was by Parker (2003; 2008) for a large specimen of the aetosaur *Desmatosuchus*.

Page 9, Lines 38-39: Yes, there appears to be a relationship between size and presence of the structure, but only in lineages where it has been determined to be present. I don't recall mention of a species level taxon where the structure is present in larger individuals but not smaller ones. At Petrified Forest there is a newly collected giant *Typhothorax* that clearly has the complex; however, it is also present in a much smaller specimen in the same collections. These specimens should be added to the database and the structures marked as present in *Typhothorax*. To be fair the very large specimen was not available at the time of the study, and the smaller specimen was confusing until the large specimen made it clear what is present.

Page 9, line 45: This sentence reads awkwardly.

Page 9, line 44-47: the association in femur length is hard to test in some specimens, such as MNA V9300, where the femur is unknown.

Page 12, line 15: missing a word in: may be a phylogenetic constraint.

Page 16, line 4: The structure is also present in *Typhothorax*. *Aetobarbakinoides* has a very similar structure to what is seen in *Scutarx deltatylus* and described and discussed by Parker (2016). So either what is in those taxa is not a hyposphene-hypantrum, or the complex is in some smaller sized animals.

Page 22, line 34: Some pseudosuchians (*Desmatosuchus*, shuvosaurids) and early diverging theropods also ankylose additional vertebrae. How does this affect your hypothesis?

Page 24, line 42: awkward phrasing.

Page 25, line 25: some large aetosaurs (*Desmatosuchus*, *Longosuchus*) fuse their cervical and anterior trunk armor, paramedian and lateral.

Page 41: line 8: aetosaur phylogeny from Parker (2016).

Figure 2: structure present in Typothorax and Aetobarbakinoides. If what is in Aetobarbakinoides is not a hyposphene/hypantrum, then it is also not present in Scutarx.

Figure 3: How would the hypothesis that silesaurids are early ornischischians affect your hypothesis? This should be mentioned.

Figure 4: How was femur length determined for aetosaurus, most of which lack femora? *Desmatosuchus smalli* has a femur as does *Longosuchus* and *Typothorax* (AMNH).

Figure 7: Please better explain your method for determining position of the trunk vertebrae based on measurements of the diapophyses and parapophyses. That would be very useful and also needs to be testable.

Table 1: *Postosuchus kirkpatrickorum* is the correct name. It's been wrong for decades. How was the femur length for *Desmatosuchus* determined? Did you use *D. smalli*? Desojo et al, 2012 describe the structure as present in *Aetobarbakinoides*. *Paratypothorax* also has a similar structure (TTUP-09416). However, only *Typothorax*, *Desmatosuchus*, and *Longosuchus* have a split hypantrum that is not connected along its entire anteroposterior length by a bony bar.

Review form: Reviewer 3

Is the manuscript scientifically sound in its present form?

No

Are the interpretations and conclusions justified by the results?

Yes

Is the language acceptable?

Yes

Is it clear how to access all supporting data?

Yes

Do you have any ethical concerns with this paper?

No

Have you any concerns about statistical analyses in this paper?

Yes

Recommendation?

Major revision is needed (please make suggestions in comments)

Comments to the Author(s)

This paper examines the evolution of vertebral accessory joints in Archosaurs, and specifically tests the hypothesis that development of the hyposphene-hypantrum may relate to biomechanical stabilization as a consequence of large size. While I like the concept of this paper, and the hypothesis is interesting, I think there could be significant improvements in terms of both its communication and analyses implemented.

Firstly, the methods need to be better explained and could be improved in several regards. Key issues are a. the exclusion of certain groups without sufficient justification, b. the need for phylogenetic correction of the logistic regression analysis, and c. statistical tests for the joint size analysis (see more below).

Another major issue with this paper is that the methods, results and discussion are not presented in a logical sequence, making it quite difficult for the reader to follow. For example, the results figures are cited extensively in the methods, before the results have been stated. Also, several methods justifications are included within the results. Finally, detailed descriptions of the main results data of the paper (articulation presence) doesn't come until the discussion. I suggest the text be reorganized.

Specific comments:

Methods

- Body size correlation – Figure 1 is cited here in the methods. This is clearly a results figures and so shouldn't be cited here.
- You analyzed pseudosuchia and avemetatarsalia separately. In the intro or here it would be good to define these groups and state why you separate them instead of looking across archosaurs (or if you did so, put these results in supp).
- Phylogenetic survey – again, don't cite results here
- Measurements – It is stated that articular surface area is measured, however based on the description in the text it is actually articular surface width? Please clarify this
- The standardization procedure also seems unconventional. If I understand correctly, the centrum height is divided by the articular width. Usually, the measurement of interest is divided by the standardizing measure? Please explain
- Centrum height is used for standardization to “obtain a unit-less metric independent of body size”. However, this assumes that centrum height scales isometrically to body size. Is there any evidence supporting this? Why was centrum height chosen over other measures? Please include discussion of these effects
- The method of ‘correcting’ the photographic measurements is also confusing. I think it would help to state your motivation at the beginning of the paragraph. I assume this correction is to take into account out-of-plane distances? How significant do you think this effect is? Added justification is required. The labelling on Figure 6 is also confusing as the length on the 2D image is labelled as A but is described as B in the text.
- Statistical analyses – The study rightfully emphasizes placing the data into their phylogenetic context, but this is not reflected in the statistical analyses. Phylogenetic logistic regression is available through the r package `phylolm`, and would be appropriate here.

Results:

- Transition ranges and thresholds are raised here for the first time. Please define how these are calculated in the methods.
- There is some conflation of results and discussion here. For example, the exploration of potential size variation in *Mandasuchus* seems like a discussion point to me. Similarly, the detailed descriptions of absence and presence in different taxa from the discussion belongs in the results I think. These are the primary data collected by the study, so I think it would be clearer if they were summarized in the results section. Similarly, the discussion of titanosaurs seems out of place here.
- Along the same lines, the fact that taxa (titanosaurs and ornithischians) were excluded from the statistical analyses is raised for the first time in the results. This should be discussed and fully justified in the methods, as they currently seem very ad hoc. The authors argue that ornithischians do not develop joints due to phylogenetic constraints, yet one of the main findings of the study is that size overcomes phylogenetic constraints. I think this requires more discussion and justification.
- The authors compare body size to articular surface area/width, suggesting that taxa with accessory articulations have smaller joints, but do not provide statistical tests. This could be

tested with an ANCOVA/PGLS, so it is not clear why qualitative comparisons are relied upon. Figure 7 could also be made clearer (see below).

Discussion:

- As mentioned above, there is a lot of results-type content in the discussion which distracts from the major points being argued. I suggest condensing the discussion to focus on interpretation of the patterns already described in the results.
- Biomechanics – There is some literature on mammals which may be useful to draw on here, with regards adaptations/joints of the vertebral column to increasing size (e.g., Jones, 2015; Chen et al., 2005; Halpert and Jenkins, 1987). Also Jones and Holbrook (2016) test a similar hypothesis in horses.

Figures:

- I think there should be a figure showing the anatomy of the hyposphene/hypantrum in relation to other vertebral joints for readers unfamiliar with this anatomy
- The order of the figures seems off (as with the text). Figure 6 relates to methods, and should come before the results figures.
- Seems like there is some repetition in the figures. Can Fig 2/4 and Fig 3/5 not be combined?
- Figure 7 – mistake in the caption, a doesn't seem to show how measurements were taken
- Color mistake in legend "outside Archosauria"
- Figure 7 – These graphs are really difficult to read. Maybe stem and leaf plots by size group would be easier?
- Figure 7a – this seems like a methods figure as it is comparing the different data collection techniques. I think this should be moved earlier or put into the supplement.

Tables:

- Only 3 table captions listed but

Decision letter (RSOS-180579.R0)

06-Jul-2018

Dear Ms Stefanic:

Manuscript ID RSOS-180579 entitled "The Evolution of the Hyposphene-hypantrum Articulation in Archosauria and Its Role in the Evolution of Large Body Size" which you submitted to Royal Society Open Science, has been reviewed. The comments from reviewers are included at the bottom of this letter.

In view of the criticisms of the reviewers, the manuscript has been rejected in its current form. However, a new manuscript may be submitted which takes into consideration these comments.

Please note that resubmitting your manuscript does not guarantee eventual acceptance, and that your resubmission will be subject to peer review before a decision is made.

Once you have revised your manuscript, go to <https://mc.manuscriptcentral.com/rsos> and login to your Author Center. Click on "Manuscripts with Decisions," and then click on "Create a

Resubmission" located next to the manuscript number. Then, follow the steps for resubmitting your manuscript.

Your resubmitted manuscript should be submitted by 03-Jan-2019. If you are unable to submit by this date please contact the Editorial Office.

Please note that Royal Society Open Science will introduce article processing charges for all new submissions received from 1 January 2018. Charges will also apply to papers transferred to Royal Society Open Science from other Royal Society Publishing journals, as well as papers submitted as part of our collaboration with the Royal Society of Chemistry (<http://rsos.royalsocietypublishing.org/chemistry>). If your manuscript is submitted and accepted for publication after 1 Jan 2018, you will be asked to pay the article processing charge, unless you request a waiver and this is approved by Royal Society Publishing. You can find out more about the charges at <http://rsos.royalsocietypublishing.org/page/charges>. Should you have any queries, please contact openscience@royalsociety.org.

Kind regards,
Andrew Dunn
Senior Publishing Editor
Royal Society Open Science
openscience@royalsociety.org

on behalf of Dr Julia Brenda Desojo (Associate Editor) and Kevin Padian (Subject Editor)
openscience@royalsociety.org

Associate Editor Comments to Author (Dr Julia Brenda Desojo):

Comments to the Author:

Dear authors, this is an interesting contribution about the evolution of accessory articulations in archosaurs, but the manuscript should be improve because the conclusions are not justified by the results in the present form. Particularly, the presence of Hyposphene/Hypantrum in larger-bodied taxa contradict the author hypothesis proposed that the acquisition of these accessory articulations was correlated with increases in body size. In order to resolve it, additional taxa, such as large ornithischians, pterosaurs and titanosaurs should be incorporate to the analysis. After that, a reanalysis of the functional explanations for the loss of these articulation should be reevaluate them. The statistical analyses need to be expanded and the clarity and layout of the text need to be significantly improved.

Reviewers' Comments to Author:

Reviewer: 1

Comments to the Author(s)

General comments

Hyposphene/hypantrum articulations are commonly found in extinct archosaurs, but are absent from living representatives of the clade. The authors test the hypothesis that the acquisition of these accessory articulations was correlated with increases in body size (and the converse that they were lost during periods of subsequent miniaturisation) by comparing character optimisations of hyposphene/hypantrum presence/absence with ancestral state reconstructions of a body mass proxy (femoral length). They find a strong correlation between the two. However, although I am generally sympathetic to the author's case, as the presence of these articulations in

larger-bodied taxa makes intuitive biomechanical sense, I think that the MS requires some additional work before it can be accepted.

The analyses presented are incomplete and would benefit from the inclusion of additional taxa (large ornithischians and pterosaurs would be obvious additions). In addition, it's unclear how ancestral body femoral lengths are derived and more clarity on the evenness of taxon sampling with respect to body size would make some of the assumptions implicit in the dataset clearer. In some ways, the taxon sample selected for the analysis looks biased in a way that would provide their preferred result and although the addition of other taxa might not overturn it, they might reduce its statistical significance or explanatory power. In addition, although the authors prefer to infer functional explanations to phylogenetic ones when examining the observed pattern of hypospine/hypantrum presence/absence, it would be difficult to disentangle these effects as the functional changes optimise at key nodes within archosaur phylogeny (and there is obvious phylogenetic signal in the autapomorphic loss of these features, and their continued absence, in even large bodied ornithischians).

Some functional explanations for the losses of these articulations are not explored, but are likely to be important when considering the evolution of this feature in archosaurs as a whole. As the authors note, there are complex patterns of gain and loss, but by effectively ignoring ornithischians, pterosaurs and titanosaurs, they miss an opportunity to come up with a more holistic explanation. As noted below, the evolution of other vertebral bracing structures (e.g. ossified tendons in ornithischians) might have been key in explaining some of these patterns, but taxa that are not congruent with their main hypothesis are overlooked, whereas these could actually provide some circumstantial support for their thesis.

Detailed comments

p. 1, lines 26–28. All body masses provided should be in metric tonnes, not tons. Please correct throughout the MS.

p. 2, lines 16–17. Indian elephants generally don't get above 5 tonnes, whereas bull African elephants do get up to 7 tonnes, though this is still relatively rare. I'd suggest changing the species here as it's generally recognised that African bush (savannah) elephants (*Loxodonta africana*) are the largest living terrestrial vertebrates. If you do decide to stick with Indian elephants, the correct spelling is *Elephas maximus*.

p. 2, lines 26–28. There's been work on the evolution of large body size in sauropods in particular, with Martin Sander's group producing a number of papers on links between sauropod biology and body size, as summarised in Sander et al. (2011) in *Biol. Rev.* and also in Klein et al. (2011) *Biology of the Sauropod Dinosaurs*.

p. 2, lines 29–31. You're right, but it would be useful to provide some examples of these skeletal changes with body size (e.g., relative robustness, etc.), otherwise this is a straw man. You list some of these changes below, but you could integrate these parts of the intro more closely.

p. 2, line 37. I'd suggest 'model system' rather than just 'model'.

p. 2, line 45. Also in terms of overall robustness, patterns of scaling, limb bone eccentricity, etc. Some of these things change ontogenetically with body size within species too.

p. 3, line 5. Give a couple of examples of smaller dinosaurs that lack these features.

p. 3, lines 9–10. It depends how you define ‘small’ and ‘large’. Teleocrater and Effigia possess hyposphene/hypantra, but are both relatively small by archosaurian standards (the modal mass of a non-avian dinosaur is around 1 tonne and these taxa get nowhere near this size). They’re both certainly a lot smaller than many ornithischians that lack these features. Also, zygosphenes and zygantra (which function to more closely integrate vertebrae, just in a slightly different way morphologically) are found in a numerous small-bodied taxa (snakes, lizards).

p. 3, line 26. To some extent, the absence of hyposphene/hypantrum articulations in large-bodied ornithischians already falsifies this hypothesis, at least in part, without needing to test it quantitatively.

p. 4, lines 40–47 and p. 5, lines 2–13. In addition to surveying taxa that are spread across the tree, thus sampling across phylogeny, did you also survey representative taxa from different size classes? Although the taxa and their femoral lengths are noted in a table, it might be good to include some kind of summary diagram that makes it clearer how even your sampling is across size classes and the tree to show your data are not biased towards particular clades or body sizes. For example, visual inspection of your list indicates that very small bodied taxa (e.g., small ornithischians, pterosaurs: femoral lengths <100 mm) are not as extensively sampled as larger-bodied taxa. I’m also surprised that pterosaurs were excluded: maybe the rationale for this should be mentioned? Finally, within the large-bodied taxon sample there are distinct phylogenetic skews: large-bodied ornithischian taxa are severely undersampled (two taxa), whereas there are 10 large-bodied sauropodomorphs (but no small ones, such as Saturnalia). In reality there’s much more overlap in the sizes of non-avian dinosaurs that lack hyposphenes/hypantra with those that possess this feature than summarised in Figure 1B.

p. 5, lines 33–35. It’s not clear how these ancestral sizes were calculated. Have all of the ancestral values you used been lifted from the literature or have you calculated these ancestral values yourselves? In either case, how were they calculated? Squared-change parsimony or something else? Also, if the ancestral sizes were taken from the literature these references mention only some theropods and pseudosuchians – how were the ancestral sizes calculated for members of the other clades you include?

p. 6, lines 6–18. You are not quantifying the surface area of the articulation here, but the length of the articulation (depending on the shape of the articular surface there could be wide variation in the actual amount of contact for the same length). Would be better to change your terminology throughout the MS to talk about this measure of postzygapophyseal length as a proxy for articular area and to note the pros and cons of using this measurement rather than surface area per se.

p. 6, lines 6–18. You mention that you also gathered data on the hyposphenes, but don’t mention how you got these measurements or whether they were also normalised in some way. Or do you include the measurement of the hyposphene within your overall measurement of postzygapophysis length? It’s not clear how any the additional articular length from the hypotheses is being captured in your analyses.

p. 9, line 36. You could refer to Figure 1 here.

p. 10, lines 22–26. It would be interesting to see what happened if you added in data from a few more large-bodied ornithischians here. I suspect the R2 would go down and there would definitely be a much, much larger ‘transitional’ range in Fig. 1B for avemetarsalians.

p. 11, lines 15–54. The derived lithostrotian titanosaur Opisthocoelicaudia has rather nice dorsals and no trace of hyposphene/hypantrum articulations. I think it’s disingenuous to regard their

absence as ‘ambiguous’ in this clade when all of the available evidence suggests that they were genuinely absent. It would be better to regard this a real anomaly on the basis of current data rather than trying to imply that they might have been there, simply as their absence isn’t congruent with the rest of your thesis on the relationship between body size and hyposphene/hypantrum presence/absence. Also, large ornithischians (e.g., multi-tonned hadrosaurs and ceratopsians, many of which are larger than some sauropods and all but the very largest theropods) lack this feature, but simply aren’t included in your analysis. As a result, body size might be part of the problem, but it can’t be the only factor, nor can it be that critical if the very largest archosaurs live without them.

p. 12, lines 3–22. It would be interesting to include titanosaurs in these regressions to see what effect they have on the overall results and to compare the regressions with and without them. At the moment, the analyses are being stacked to make body size look more important by removing large taxa that lack these features (titanosaurs, ornithischians). This isn’t a thorough test of the hypothesis. You are making an a priori judgement about the role of phylogeny here also, rather than testing it, although elsewhere in the paper you say phylogeny doesn’t have a strong effect. Quite a bit of circular reasoning.

p. 12, lines 43 onward. Again, I’m not sure how you quantified hyposphene size here.

p. 14, lines 53 onward. Figures 2 and 3 suggest that the presence of hyposphene/hypantra is clearly convergent on the basis of character optimisations, without the need to invoke body size as another factor, and that it was not present in the archosaur common ancestor, contrary to the statement in the text.

p. 16, lines 31–45. Early ornithischians are comparable in size to Tawa and *Asilisaurus* in femoral length, but lack hyposphenes/hypantra. Interesting that other dinosaurs gain these features at small body sizes, but ornithischians don’t. This has to be phylogenetic signal, as you suggest in passing elsewhere, but is worthy of more comment.

p. 17, lines 3–8. How would this optimisation change if you used an alternative dinosaur phylogeny that included either Ornithoscelida or Phytodinosauria (e.g. Baron et al., 2017; Langer et al. 2017; Parry et al. 2017)? If Ornithoscelida a real entity this suggests that these features are primitive for Dinosauria (and then lost in Ornithischia), not just a saurischian feature.

pp. 20–22. A number of the same measurements are repeated several times in successive paragraphs, which doesn’t seem necessary.

pp. 22–24. It’s interesting that you don’t mention two other obvious bracing systems of archosaurs: osteoderms and ossified tendons. It’s been suggested that the osteoderms of crown crocodylians act as a bracing system (and they lack hyposphenes) and also that the ossified tendons present in all ornithischians also provided this function (and they lack hyposphenes). The lack of these features in these taxa could reflect the acquisition of these other bracing systems also. It seems like something that would be important to consider in ornithischians especially, but it’s not mentioned here at all. Miniturisation might not have been necessary in the loss of this system at all – it might simply have been replaced by other features at larger body sizes.

Figs 2–5. You include Squamata and Phytosauria in your trees, but there are no data on body size or the presence/absence of hyposphenes in these animals in your data tables. These data should be included. Also, I’d argue that squamates have something very similar to these features, even though the name differs. If you think that the zygantara/zygosphenes of squamates are not homologous with hyposphene/hypantra I’d be tempted to exclude them from your analysis here.

Figs 2-5. I'm unclear why some of the lineages (e.g. Avemetatarsalia in Fig. 2 or Pseudosuchia in Fig. 3) have different colours at different points along their lengths. If using maximum parsimony surely there should simply be an extrapolation of the ancestral state for that clade along the entire length of the branch from the tip to the node. I understand that this might be to capture some information on the presence/absence of the feature within each clade each time this is done, but as you're really only interested in the condition at the base of each clade as an outgroup in each case, adding this extra nuance in just makes it look confusing (and the nuances within each clade are captured in the more expanded trees you have for Pseudosuchia/Avemetatarsalia in successive figures anyhow).

No list of institutional abbreviations is provided for the various specimens listed in the MS and the tables.

Paul M. Barrett

Reviewer: 2

Comments to the Author(s)

This is a well-written, clean manuscript that looks at the the evolution of the hyposphene-hypantrum articulations in archosaurs, an important vertebral accessory articulation that was once thought to be restricted to dinosaurs. The authors argue that the presence of this structure is convergent between groups, and mostly tied to large body size; however, its lack in large bodied groups such as the ornithischian dinosaurs, as well as crown group archosaurs suggest that there is some phylogenetic significance as well. This conflict is addressed by the hypothesis that large animals require vertebral bracing and these articulations served this purpose, but that in groups lacking them they accomplished this bracing in other ways. I feel this argument is a little weak, for reasons found in the following comments, but overall the presence in larger individuals does appear to be for the most part supported.

Other comments:

Page 3, line 8: The possibly first recognition of these structures in a pseudosuchian archosaur was by Parker (2003; 2008) for a large specimen of the aetosaur *Desmotosuchus*.

Page 9, Lines 38-39: Yes, there appears to be a relationship between size and presence of the structure, but only in lineages where it has been determined to be present. I don't recall mention of a species level taxon where the structure is present in larger individuals but not smaller ones. At Petrified Forest there is a newly collected giant *Tyothorax* that clearly has the complex; however, it is also present in a much smaller specimen in the same collections. These specimens should be added to the database and the structures marked as present in *Tyothorax*. To be fair the very large specimen was not available at the time of the study, and the smaller specimen was confusing until the large specimen made it clear what is present.

Page 9, line 45: This sentence reads awkwardly.

Page 9, line 44-47: the association in femur length is hard to test in some specimens, such as MNA V9300, where the femur is unknown.

Page 12, line 15: missing a word in: may be a phylogenetic constraint.

Page 16, line 4: The structure is also present in *Tyothorax*. *Aetobarbakinoides* has a very similar structure to what is seen in *Scutarx deltatylus* and described and discussed by Parker (2016). So either what is in those taxa is not a hyposphene-hypantrum, or the complex is in some smaller sized animals.

Page 22, line 34: Some pseudosuchians (*Desmatosuchus*, shuvosaurids) and early diverging theropods also ankylose additional vertebrae. How does this affect your hypothesis?

Page 24, line 42: awkward phrasing.

Page 25, line 25: some large aetosaurs (*Desmatosuchus*, *Longosuchus*) fuse their cervical and anterior trunk armor, paramedian and lateral.

Page 41: line 8: aetosaur phylogeny from Parker (2016).

Figure 2: structure present in *Typothorax* and *Aetobarbakinoides*. If what is in *Aetobarbakinoides* is not a hyposphene/hypantrum, then it is also not present in *Scutarx*.

Figure 3: How would the hypothesis that silesaurids are early ornithischians affect your hypothesis? This should be mentioned.

Figure 4: How was femur length determined for aetosaurs, most of which lack femora? *Desmatosuchus smalli* has a femur as does *Longosuchus* and *Typothorax* (AMNH).

Figure 7: Please better explain your method for determining position of the trunk vertebrae based on measurements of the diapophyses and parapophyses. That would be very useful and also needs to be testable.

Table 1: *Postosuchus kirkpatrickorum* is the correct name. It's been wrong for decades. How was the femur length for *Desmatosuchus* determined? Did you use *D. smalli*? Desojo et al, 2012 describe the structure as present in *Aetobarbakinoides*. *Paratypothorax* also has a similar structure (TTUP-09416). However, only *Typothorax*, *Desmatosuchus*, and *Longosuchus* have a split hypantrum that is not connected along its entire anteroposterior length by a bony bar.

Reviewer: 3

Comments to the Author(s)

This paper examines the evolution of vertebral accessory joints in Archosaurs, and specifically tests the hypothesis that development of the hyposphene-hypantrum may relate to biomechanical stabilization as a consequence of large size. While I like the concept of this paper, and the hypothesis is interesting, I think there could be significant improvements in terms of both its communication and analyses implemented.

Firstly, the methods need to be better explained and could be improved in several regards. Key issues are a. the exclusion of certain groups without sufficient justification, b. the need for phylogenetic correction of the logistic regression analysis, and c. statistical tests for the joint size analysis (see more below).

Another major issue with this paper is that the methods, results and discussion are not presented in a logical sequence, making it quite difficult for the reader to follow. For example, the results figures are cited extensively in the methods, before the results have been stated. Also, several methods justifications are included within the results. Finally, detailed descriptions of the main results data of the paper (articulation presence) doesn't come until the discussion. I suggest the text be reorganized.

Specific comments:

Methods

- Body size correlation – Figure 1 is cited here in the methods. This is clearly a results figures and so shouldn't be cited here.
- You analyzed pseudosuchia and avemetatarsalia separately. In the intro or here it would be good to define these groups and state why you separate them instead of looking across archosaurs (or if you did so, put these results in supp).
- Phylogenetic survey – again, don't cite results here
- Measurements – It is stated that articular surface area is measured, however based on the description in the text it is actually articular surface width? Please clarify this
- The standardization procedure also seems unconventional. If I understand correctly, the centrum height is divided by the articular width. Usually, the measurement of interest is divided by the standardizing measure? Please explain
- Centrum height is used for standardization to “obtain a unit-less metric independent of body size”. However, this assumes that centrum height scales isometrically to body size. Is there any evidence supporting this? Why was centrum height chosen over other measures? Please include discussion of these effects
- The method of ‘correcting’ the photographic measurements is also confusing. I think it would help to state your motivation at the beginning of the paragraph. I assume this correction is to take into account out-of-plane distances? How significant do you think this effect is? Added justification is required. The labelling on Figure 6 is also confusing as the length on the 2D image is labelled as A but is described as B in the text.
- Statistical analyses – The study rightfully emphasizes placing the data into their phylogenetic context, but this is not reflected in the statistical analyses. Phylogenetic logistic regression is available through the r package phylolm, and would be appropriate here.

Results:

- Transition ranges and thresholds are raised here for the first time. Please define how these are calculated in the methods.
- There is some conflation of results and discussion here. For example, the exploration of potential size variation in Mandasuchus seems like a discussion point to me. Similarly, the detailed descriptions of absence and presence in different taxa from the discussion belongs in the results I think. These are the primary data collected by the study, so I think it would be clearer if they were summarized in the results section. Similarly, the discussion of titanosaurs seems out of place here.
- Along the same lines, the fact that taxa (titanosaurs and ornithischians) were excluded from the statistical analyses is raised for the first time in the results. This should be discussed and fully justified in the methods, as they currently seem very ad hoc. The authors argue that ornithischians do not develop joints due to phylogenetic constraints, yet one of the main findings of the study is that size overcomes phylogenetic constraints. I think this requires more discussion and justification.
- The authors compare body size to articular surface area/width, suggesting that taxa with accessory articulations have smaller joints, but do not provide statistical tests. This could be tested with an ANCOVA/PGLS, so it is not clear why qualitative comparisons are relied upon. Figure 7 could also be made clearer (see below).

Discussion:

- As mentioned above, there is a lot of results-type content in the discussion which distracts from the major points being argued. I suggest condensing the discussion to focus on interpretation of the patterns already described in the results.
- Biomechanics – There is some literature on mammals which may be useful to draw on here, with regards adaptations/joints of the vertebral column to increasing size (e.g., Jones, 2015; Chen et al., 2005; Halpert and Jenkins, 1987). Also Jones and Holbrook (2016) test a similar hypothesis in horses.

Figures:

- I think there should be a figure showing the anatomy of the hyposphene/hypantrum in relation to other vertebral joints for readers unfamiliar with this anatomy
- The order of the figures seems off (as with the text). Figure 6 relates to methods, and should come before the results figures.
- Seems like there is some repetition in the figures. Can Fig 2/4 and Fig 3/5 not be combined?
- Figure 7 – mistake in the caption, a doesn't seem to show how measurements were taken
- Color mistake in legend "outside Archosauria"
- Figure 7 – These graphs are really difficult to read. Maybe stem and leaf plots by size group would be easier?
- Figure 7a – this seems like a methods figure as it is comparing the different data collection techniques. I think this should be moved earlier or put into the supplement.

Tables:

- Only 3 table captions listed but

Author's Response to Decision Letter for (RSOS-180579.R0)

See Appendix A.

RSOS-190258.R0

Review form: Reviewer 2

Is the manuscript scientifically sound in its present form?

Yes

Are the interpretations and conclusions justified by the results?

Yes

Is the language acceptable?

Yes

Is it clear how to access all supporting data?

Yes

Do you have any ethical concerns with this paper?

No

Have you any concerns about statistical analyses in this paper?

No

Recommendation?

Accept as is

Comments to the Author(s)

The authors have addressed my concerns from the initial draft. The definition of what constitutes a hyosphene-hypantrum articulation will continue to be debated and I don't entirely agree with the findings of the authors as to what taxa possess this feature; however, their proposed definition has been clearly stated and their identifications are consistent with that definition so I am fine with publication as is. Future studies will support or contest these conclusions.

Review form: Reviewer 3

Is the manuscript scientifically sound in its present form?

No

Are the interpretations and conclusions justified by the results?

No

Is the language acceptable?

Yes

Is it clear how to access all supporting data?

Yes

Do you have any ethical concerns with this paper?

No

Have you any concerns about statistical analyses in this paper?

Yes

Recommendation?

Major revision is needed (please make suggestions in comments)

Comments to the Author(s)

While the structure of the manuscript is much improved, I still feel that it requires significant work to improve clarity and justify its conclusions. Further, there are several points from the original review which remain unaddressed. To my mind the most significant issues are: a. statistical analyses, b. the way the excluded groups are dealt with, and c. several issues with presentation of data. Before publication the authors should implement appropriate phylogenetic statistical analyses, represent and analyze the complete dataset prior to excluding the outlier groups and ensure that all figures are legible, correct and clearly presented.

See my responses to select author replies below:

- You analyzed pseudosuchia and avemetatarsalia separately. In the intro or here it would be good to define these groups and state why you separate them instead of looking across archosaurs (or if you did so, put these results in supp). We only analyzed them separately when we discovered that they have different body size thresholds for presence of the hyosphene-hypantrum articulation. We added a sentence to the third paragraph of the Introduction, "We focused solely on members of Archosauria because no other vertebrate outside of the group has been reported to possess these articulations or any structures homologous to the articulation."

This is not what I was driving at here. Why are the groups separated? If they do have different thresholds, how do you interpret that? Suggests interaction between phylogeny and biomechanics.

- It is stated that articular surface area is measured, however based on the description in the text it is actually articular surface width? Please clarify this. We modified this section to be clearer about the measurements we took: "... measure the articular processes (the postzygapophyses in all taxa and hyposphene structures when applicable) on the posterior aspects of trunk vertebrae to track surface length involved in intervertebral articulation. Postzygapophyses were measured from the point where they contact the neural arch to their distal-most aspect. Hyposphenes were measured along the entire aspect of their lateral faces. To obtain a unit-less metric independent of body size, we also measured centrum height for each vertebra. By dividing centrum height by the length (long-axis) of one postzygapophysis for each vertebra observed, we obtained a measurement that could be used as a proxy for relative surface area of articulation between vertebrae." While I appreciate the enhanced clarity here, it could still be improved. "Distal-most" (distal to what?) would be clearer as "dorsal-most". Also referring to the "long-axis" of the zygapophysis is confusing as long-axis is generally cranio-caudal in the axial skeleton. I would suggest "maximum length of the articular surface". Same on P9, line 17.

Also, this does not justify use of relative surface area throughout the text - this is misleading as it is not what was measured. Alter wording to use 'maximum length' or something similar throughout the paper.

The method of 'correcting' the photographic measurements is also confusing. I think it would help to state your motivation at the beginning of the paragraph. I assume this correction is to take into account out-of-plane distances? How significant do you think this effect is? Added justification is required. The labelling on Figure 6 is also confusing as the length on the 2D image is labelled as A but is described as B in the text.

Reworded the section explaining the conversion factor: "To make sure that the photographic measurements were comparable with our in-person measurements, because the measurements in-person were in a slightly different three dimensional plane than those taken from a photograph in Photoshop, we took both measurements "A" (in-person) and "B" (in Photoshop) (Figure 2) on the same vertebrae to determine a standardized conversion factor of how different those measurements were from each other. That way, we could multiply one measure by that factor to convert it to a number about equal to what the other measure would be."

We fixed the labeling on Figure 6, we did have A and B accidentally swapped in the figure.

This is better, but there is still a problem with Figure 2. There seems to be a panel missing (oblique view is described in caption but does not appear in figure), so measurement A is not illustrated anywhere in the figure, making the triangle with A and B very confusing. Both measurements A and B need to be illustrated or this figure is uninterpretable.

P10, line 10: Make a statement about how variable zygapophysis angle is in archosaurs and if you expect that this could lead to any systematic bias (or not).

- Statistical analyses - The study rightfully emphasizes placing the data into their phylogenetic context, but this is not reflected in the statistical analyses. Phylogenetic logistic regression is available through the *r* package *phylolm*, and would be appropriate here.

The test illustrated in Figure 4 is specifically meant to exclude phylogenetic context once within either Pseudosuchia or Avemetatarsalia and show that within these groups, how well does body size reflect whether or not a species will exhibit the hyposphene-hypantrum articulation.

It is not possible to "exclude phylogenetic context" here. For example, the patterns shown in the logistic regression in Figure 4 could be achieved by a single evolutionary transition with multiple species sampled on either side - in which case the true sample size would be $n=1$, and there

would be no way to determine statistical significance based on such a single occurrence. I'm not saying this is the case here, simply making an extreme example to illustrate my point. Most likely given the observed patterns the effect would still be significant, but it needs to be shown. An uncorrected analysis would be appropriate if the trait were 100% labile with no phylogenetic signal, but this is clearly not the case here.

The authors should therefore use a phylogenetically informed analysis for determining significance. The options are either a phylogenetic logistic regression, or a PGLS. Given that both variables are observed with error, and there isn't a clear controlled variable here, it's not important which is dependent vs independent. PGLS would allow you to ask if taxa with joints are larger than those without, and is simple to implement in R. I suggest authors run the analysis both with and without the ornithischians and titanosaurs to assess the effect.

The size variable should be logged both in Figure 4 and in the analysis as its distribution is skewed by the very large taxa. This will also aid comparisons between the data subsets.

Results:

- Transition ranges and thresholds are raised here for the first time. Please define how these are calculated in the methods.

Added to the end of the methods under (e) Statistical Analyses section: "Our results will yield threshold ranges for body size that can be interpreted as above the range you expect the hyposphene-hypantrum to be present and below the range you expect it to be absent. The range itself represents a transitional body size where the articulation may or may not be seen in taxa of that size."

This does not explain how these ranges are calculated. Based on Figure 4 it seems that the range is simply the the smallest specimen with a joint to the largest specimen without a joint. Is that correct? Or is it derived from the logistic regression somehow? Clarify.

Along the same lines, the fact that taxa (titanosaurs and ornithischians) were excluded from the statistical analyses is raised for the first time in the results. This should be discussed and fully justified in the methods, as they currently seem very ad hoc. The authors argue that ornithischians do not develop joints due to phylogenetic constraints, yet one of the main findings of the study is that size overcomes phylogenetic constraints. I think this requires more discussion and justification.

Added a sentence justifying these exclusions to Methods section (e) Statistical analyses: "We excluded several groups of archosaurs that lack the hyposphene-hypantrum from our statistical analyses (e.g., titanosaurs, ornithischians, pterosaurs) that we consider exceptions because they lose the articulation at or before the base of their clade and then have secondarily derived vertebral morphology before taxa grow to large body sizes (see Discussion section: (b) Exceptions within Archosauria (extinct lineages))."

The exclusion of these groups still seems ad hoc. It would make more sense to plot all the data and allow the data to identify those groups as outliers, at which point you can discuss why they might be outliers and exclude them and look at patterns without them. The primary data presented by this study are the taxonomic survey data of presence of hh joints (Table 1), so a figure displaying this data in its entirety would be helpful. For example, a scatterplot of presence vs size by phylogenetic subgroup.

- The authors compare body size to articular surface area/width, suggesting that taxa with accessory articulations have smaller joints, but do not provide statistical tests. This could be

tested with an ANCOVA/PGLS, so it is not clear why qualitative comparisons are relied upon. Figure 7 could also be made clearer (see below).

For these data shown in Figure 8a and 8b (in the previous submission they were numbered 7b and 7c) we are showing that specimens that have a hyposphene-hypantrum plot distinctly from specimens that do not have a hyposphene-hypantrum in terms of relative surface area of postzygapophyseal articulation. We show this visually using position along the vertebral column as our x-axis, because only datapoints from the same vertebral position can be compared to each other. We did not run a statistical analysis on these data because the subset of specimens/ taxa that had vertebrae from a majority of the trunk regions of the vertebral column was very low (and of course skewed towards the extant). To run an ANCOVA or PGLS we would need many more taxa than we sampled in this study and would need to run each statistical test on only data points of the same vertebral position. Any test we would have ran with this data would have had too low statistical power to be informative, so ultimately we chose to show these data as a visual relationship only. We will however continue to think about how these data could be tested using statistical methods.

Please state the sampling for this analysis in the methods. Also please clarify how the data in figure 8 relate – is each point a single specimen or a mean (see below)? Is it a mix of specimens? Given there seems to be no along-column trend in joint length, one possibility is to calculate mean length for each specimen by averaging across vertebrae. This may give you enough sampling to statistically test between those with and without the joints, both including and excluding the extra articulation surface.

Figures:

- Figure 7 – mistake in the caption, a doesn't seem to show how measurements were taken Fixed.

As stated above this figure still doesn't show how measurement A was taken. I think a panel must be missing as it doesn't make sense as is.

- Color mistake in legend “outside Archosauria” We found no mistake here.

I'm afraid there is a mistake in the key. I cannot see any points in the plot corresponding to the cream crosses “outside Archosauria <300mm” category in the key, whereas I can see various green crosses.

Also, Figure 3 & 8 – What do these data represent? It is unclear if these data are individual specimens or group means. If individual, which species are represented (list in caption)? If they are group means, please display some measure of spread. Why is there (a) and no (b) in 3. There is not enough information to interpret these figures.

- Figure 7 – These graphs are really difficult to read. Maybe stem and leaf plots by size group would be easier? We think that these graphs are adequate and we are leaving them as they are because no other reviewer commented negatively on them.

While these plots may be 'adequate', they are far from compelling. There are too many groupings to convey a clear message. I suggest the authors simplify/clarify the figures if they want readers to come away with a clear take-home.

Additional points:

- Do you have any hypotheses you are testing with this measurement data, as these results aren't currently placed into any specific framework? For example, do you hypothesize that taxa with the joints will have longer articular surfaces, relative to size, than those without? Why might you expect that? This analysis is difficult to interpret because the context is not laid out and questions are not clear.

- P14, line 5 onwards: A tentative link between size, hh and joint length is raised in the results but not tested nor its implications discussed (see setting up framework above). Either give this result context and discuss it (even if the conclusion is we don't have enough information to accept or reject the hypothesis right now) or remove it.
- The authors conclude that the hh joint is "closely constrained by body size in extinct archosaurs". However, it seems it is both related to body size and phylogeny - there are certain major phylogenetic groups for which the joint never evolves, for biomechanical or other reasons. Therefore, the evolution of this trait is more accurately described as a mosaic of straight allometry and phylogenetic inertia.

Decision letter (RSOS-190258.R0)

12-Mar-2019

Dear Ms Stefanic,

The Subject Editor assigned to your paper ("The Evolution of the Hyposphene-hypantrum Articulation in Archosauria and Its Role in the Evolution of Large Body Size") has now received comments from reviewers. We would like you to revise your paper in accordance with the referee and Associate Editor suggestions which can be found below (not including confidential reports to the Editor). Please note this decision does not guarantee eventual acceptance.

Please submit a copy of your revised paper before 04-Apr-2019. Please note that the revision deadline will expire at 00.00am on this date. If we do not hear from you within this time then it will be assumed that the paper has been withdrawn. In exceptional circumstances, extensions may be possible if agreed with the Editorial Office in advance. We do not allow multiple rounds of revision so we urge you to make every effort to fully address all of the comments at this stage. If deemed necessary by the Editors, your manuscript will be sent back to one or more of the original reviewers for assessment. If the original reviewers are not available we may invite new reviewers.

When submitting your revised manuscript, you must respond to the comments made by the referees and upload a file "Response to Referees" in "Section 6 - File Upload". Please use this to document how you have responded to each of the comments, and the adjustments you have made. In order to expedite the processing of the revised manuscript, please be as specific as possible in your response.

- Ethics statement

If your study uses humans or animals please include details of the ethical approval received, including the name of the committee that granted approval. For human studies please also detail

whether informed consent was obtained. For field studies on animals please include details of all permissions, licences and/or approvals granted to carry out the fieldwork.

- Data accessibility

If you wish to submit your supporting data or code to Dryad (<http://datadryad.org/>), or modify your current submission to dryad, please use the following link:
<http://datadryad.org/submit?journalID=RSOS&manu=RSOS-190258>

- Competing interests

- Authors' contributions

- Acknowledgements

- Funding statement

Kind regards,
Andrew Dunn
Royal Society Open Science Editorial Office

on behalf of Dr Julia Brenda Desojo (Associate Editor) and Kevin Padian (Subject Editor)
 openscience@royalsociety.org

Reviewer comments to Author:

Reviewer: 2

Comments to the Author(s)

The authors have addressed my concerns from the initial draft. The definition of what constitutes a hyposphene-hypantrum articulation will continue to be debated and I don't entirely agree with the findings of the authors as to what taxa possess this feature; however, their proposed definition has been clearly stated and their identifications are consistent with that definition so I am fine with publication as is. Future studies will support or contest these conclusions.

Reviewer: 3

Comments to the Author(s)

While the structure of the manuscript is much improved, I still feel that it requires significant work to improve clarity and justify its conclusions. Further, there are several points from the original review which remain unaddressed. To my mind the most significant issues are: a. statistical analyses, b. the way the excluded groups are dealt with, and c. several issues with presentation of data. Before publication the authors should implement appropriate phylogenetic statistical analyses, represent and analyze the complete dataset prior to excluding the outlier groups and ensure that all figures are legible, correct and clearly presented.

See my responses to select author replies below:

- You analyzed pseudosuchia and avemetatarsalia separately. In the intro or here it would be good to define these groups and state why you separate them instead of looking across archosaurs (or if you did so, put these results in supp). We only analyzed them separately when we discovered that they have different body size thresholds for presence of the hyposphene-hypantrum articulation. We added a sentence to the third paragraph of the Introduction, "We focused solely on members of Archosauria because no other vertebrate outside of the group has been reported to possess these articulations or any structures homologous to the articulation." This is not what I was driving at here. Why are the groups separated? If they do have different thresholds, how do you interpret that? Suggests interaction between phylogeny and biomechanics.

- It is stated that articular surface area is measured, however based on the description in the text it is actually articular surface width? Please clarify this.

We modified this section to be clearer about the measurements we took: "... measure the articular processes (the postzygapophyses in all taxa and hyposphene structures when applicable) on the posterior aspects of trunk vertebrae to track surface length involved in intervertebral articulation. Postzygapophyses were measured from the point where they contact the neural arch to their distal-most aspect. Hyposphenes were measured along the entire aspect of their lateral faces. To obtain a unit-less metric independent of body size, we also measured centrum height for each vertebra. By dividing centrum height by the length (long-axis) of one postzygapophysis for each vertebra observed, we obtained a measurement that could be used as a proxy for relative surface area of articulation between vertebrae."

While I appreciate the enhanced clarity here, it could still be improved. "Distal-most" (distal to what?) would be clearer as "dorsal-most". Also referring to the "long-axis" of the zygapophysis

is confusing as long-axis is generally cranio-caudal in the axial skeleton. I would suggest “maximum length of the articular surface”. Same on P9, line 17.

Also, this does not justify use of relative surface area throughout the text – this is misleading as it is not what was measured. Alter wording to use ‘maximum length’ or something similar throughout the paper.

The method of ‘correcting’ the photographic measurements is also confusing. I think it would help to state your motivation at the beginning of the paragraph. I assume this correction is to take into account out-of-plane distances? How significant do you think this effect is? Added justification is required. The labelling on Figure 6 is also confusing as the length on the 2D image is labelled as A but is described as B in the text.

Reworded the section explaining the conversion factor: “To make sure that the photographic measurements were comparable with our in-person measurements, because the measurements in-person were in a slightly different three dimensional plane than those taken from a photograph in Photoshop, we took both measurements “A” (in-person) and “B” (in Photoshop) (Figure 2) on the same vertebrae to determine a standardized conversion factor of how different those measurements were from each other. That way, we could multiply one measure by that factor to convert it to a number about equal to what the other measure would be.”

We fixed the labeling on Figure 6, we did have A and B accidentally swapped in the figure.

This is better, but there is still a problem with Figure 2. There seems to be a panel missing (oblique view is described in caption but does not appear in figure), so measurement A is not illustrated anywhere in the figure, making the triangle with A and B very confusing. Both measurements A and B need to be illustrated or this figure is uninterpretable.

P10, line 10: Make a statement about how variable zygapophysis angle is in archosaurs and if you expect that this could lead to any systematic bias (or not).

- Statistical analyses – The study rightfully emphasizes placing the data into their phylogenetic context, but

this is not reflected in the statistical analyses. Phylogenetic logistic regression is available through the r

package `phylolm`, and would be appropriate here.

The test illustrated in Figure 4 is specifically meant to exclude phylogenetic context once within either

Pseudosuchia or Avemetatarsalia and show that within these groups, how well does body size reflect whether

or not a species will exhibit the hyosphene-hypantrum articulation.

It is not possible to “exclude phylogenetic context” here. For example, the patterns shown in the logistic regression in Figure 4 could be achieved by a single evolutionary transition with multiple species sampled on either side – in which case the true sample size would be $n=1$, and there would be no way to determine statistical significance based on such a single occurrence. I’m not saying this is the case here, simply making an extreme example to illustrate my point. Most likely given the observed patterns the effect would still be significant, but it needs to be shown. An uncorrected analysis would be appropriate if the trait were 100% labile with no phylogenetic signal, but this is clearly not the case here.

The authors should therefore use a phylogenetically informed analysis for determining significance. The options are either a phylogenetic logistic regression, or a PGLS. Given that both variables are observed with error, and there isn’t a clear controlled variable here, it’s not important which is dependent vs independent. PGLS would allow you to ask if taxa with joints

are larger than those without, and is simple to implement in R. I suggest authors run the analysis both with and without the ornithischians and titanosaurs to assess the effect.

The size variable should be logged both in Figure 4 and in the analysis as its distribution is skewed by the very large taxa. This will also aid comparisons between the data subsets.

Results:

- Transition ranges and thresholds are raised here for the first time. Please define how these are calculated in the methods.

Added to the end of the methods under (e) Statistical Analyses section: "Our results will yield threshold ranges for body size that can be interpreted as above the range you expect the hyposphene-hypantrum to be present and below the range you expect it to be absent. The range itself represents a transitional body size where the articulation may or may not be seen in taxa of that size."

This does not explain how these ranges are calculated. Based on Figure 4 it seems that the range is simply the the smallest specimen with a joint to the largest specimen without a joint. Is that correct? Or is it derived from the logistic regression somehow? Clarify.

Along the same lines, the fact that taxa (titanosaurs and ornithischians) were excluded from the statistical analyses is raised for the first time in the results. This should be discussed and fully justified in the methods, as they currently seem very ad hoc. The authors argue that ornithischians do not develop joints due to phylogenetic constraints, yet one of the main findings of the study is that size overcomes phylogenetic constraints. I think this requires more discussion and justification.

Added a sentence justifying these exclusions to Methods section (e) Statistical analyses: "We excluded several groups of archosaurs that lack the hyposphene-hypantrum from our statistical analyses (e.g., titanosaurs, ornithischians, pterosaurs) that we consider exceptions because they lose the articulation at or before the base of their clade and then have secondarily derived vertebral morphology before taxa grow to large body sizes (see Discussion section: (b) Exceptions within Archosauria (extinct lineages))."

The exclusion of these groups still seems ad hoc. It would make more sense to plot all the data and allow the data to identify those groups as outliers, at which point you can discuss why they might be outliers and exclude them and look at patterns without them. The primary data presented by this study are the taxonomic survey data of presence of hh joints (Table 1), so a figure displaying this data in its entirety would be helpful. For example, a scatterplot of presence vs size by phylogenetic subgroup.

- The authors compare body size to articular surface area/width, suggesting that taxa with accessory articulations have smaller joints, but do not provide statistical tests. This could be tested with an

ANCOVA/PGLS, so it is not clear why qualitative comparisons are relied upon. Figure 7 could also be made clearer (see below).

For these data shown in Figure 8a and 8b (in the previous submission they were numbered 7b and 7c) we are showing that specimens that have a hyposphene-hypantrum plot distinctly from specimens that do not have a hyposphene-hypantrum in terms of relative surface area of postzygapophyseal articulation. We show this visually using position along the vertebral column as our x-axis, because only datapoints from the same vertebral position can be compared to each other. We did not run a statistical analysis on these data because the subset of specimens/ taxa that had vertebrae from a majority of the trunk regions of the vertebral column was very low (and of course skewed towards the extant). To run an ANCOVA or PGLS we would need many

more taxa than we sampled in this study and would need to run each statistical test on only data points of the same vertebral position. Any test we would have run with this data would have had too low statistical power to be informative, so ultimately we chose to show these data as a visual relationship only. We will however continue to think about how these data could be tested using statistical methods.

Please state the sampling for this analysis in the methods. Also please clarify how the data in figure 8 relate – is each point a single specimen or a mean (see below)? Is it a mix of specimens? Given there seems to be no along-column trend in joint length, one possibility is to calculate mean length for each specimen by averaging across vertebrae. This may give you enough sampling to statistically test between those with and without the joints, both including and excluding the extra articulation surface.

Figures:

- Figure 7 – mistake in the caption, a doesn't seem to show how measurements were taken Fixed.

As stated above this figure still doesn't show how measurement A was taken. I think a panel must be missing as it doesn't make sense as is.

- Color mistake in legend “outside Archosauria” We found no mistake here.

I'm afraid there is a mistake in the key. I cannot see any points in the plot corresponding to the cream crosses “outside Archosauria <300mm” category in the key, whereas I can see various green crosses.

Also, Figure 3 & 8 – What do these data represent? It is unclear if these data are individual specimens or group means. If individual, which species are represented (list in caption)? If they are group means, please display some measure of spread. Why is there (a) and no (b) in 3. There is not enough information to interpret these figures.

- Figure 7 – These graphs are really difficult to read. Maybe stem and leaf plots by size group would be easier? We think that these graphs are adequate and we are leaving them as they are because no other reviewer commented negatively on them.

While these plots may be ‘adequate’, they are far from compelling. There are too many groupings to convey a clear message. I suggest the authors simplify/clarify the figures if they want readers to come away with a clear take-home.

Additional points:

- Do you have any hypotheses you are testing with this measurement data, as these results aren't currently placed into any specific framework? For example, do you hypothesize that taxa with the joints will have longer articular surfaces, relative to size, than those without? Why might you expect that? This analysis is difficult to interpret because the context is not laid out and questions are not clear.
- P14, line 5 onwards: A tentative link between size, hh and joint length is raised in the results but not tested nor its implications discussed (see setting up framework above). Either give this result context and discuss it (even if the conclusion is we don't have enough information to accept or reject the hypothesis right now) or remove it.
- The authors conclude that the hh joint is “closely constrained by body size in extinct archosaurs”. However, it seems it is both related to body size and phylogeny – there are certain major phylogenetic groups for which the joint never evolves, for biomechanical or other reasons. Therefore, the evolution of this trait is more accurately described as a mosaic of straight allometry and phylogenetic inertia.

Author's Response to Decision Letter for (RSOS-190258.R0)

See Appendix B.

RSOS-190258.R1 (Revision)

Review form: Reviewer 2

Is the manuscript scientifically sound in its present form?

Yes

Are the interpretations and conclusions justified by the results?

Yes

Is the language acceptable?

Yes

Is it clear how to access all supporting data?

Yes

Do you have any ethical concerns with this paper?

No

Have you any concerns about statistical analyses in this paper?

No

Recommendation?

Accept as is

Comments to the Author(s)

All of my concerns have been addressed.

Review form: Reviewer 4

Is the manuscript scientifically sound in its present form?

No

Are the interpretations and conclusions justified by the results?

No

Is the language acceptable?

No

Is it clear how to access all supporting data?

Not Applicable

Do you have any ethical concerns with this paper?

No

Have you any concerns about statistical analyses in this paper?

Yes

Recommendation?

Reject

Comments to the Author(s)

Hyposphene-Hypantrum ms (RSOS-190258.R1)

General observations

What is the general nature of this 'hypothesis-driven' article? It is narrowly focused on their tightly constrained definition of one anatomical configuration (the hyposphene-hypantrum [hypo-hypa]) and its distribution within sub-clades of archosaurian reptiles.

This promotes the idea that the hypo-hypa is a solely size-related construct (i.e. they are analogues randomly distributed across, and restricted to, large-sized archosaurians). The thrust of their argument is that there is no underlying phylogenetic component to their appearance/distribution in individual lineages/clades. Some do, some don't - and if they don't they have alternative mechanisms to solve the 'problems' of rigging and bracing large skeletons.

This 'notion' is backed up by statistical analyses and graphical presentations that are purporting to support this general thesis.

Unfortunately, this general thesis is undermined by their observation that basal (=ancestral) archosaurs inherited a common tendency to develop these structures (page 16:10), which are then expressed, or not expressed, later in their phylogeny. This is not an inspiring juxtaposition of concepts. If hypo-hypa are indeed part of a common inheritance then these features are arguably homologous (as the intricate nature of their similarity across taxa implies) rather than analogous, as the tenor of this article seems to wish to imply.

Figures. Annotation abbreviations in the text would be appropriate if the reader wants to cross-reference anatomy that is being described textually with what has been CLEARLY(?) Illustrated. But, the illustrations are inadequate and confusingly annotated.

As a trivial point of fact, I have a dorsal vertebra of a quite large *Crocodylus niloticus* on my desk (as I type this), which has a structure not unlike a hyposphene between its postzygapophyses! However, it is more of a functional/structural analogue than a feature that fits into their narrow definition of what a hypo is, of course.

First impression: this article is not at all well written and would need to be carefully re-written throughout. I thought it was just the abstract/intro, but this problem is pervasive. The language is occasionally inaccurate, syntactically challenged and quite often confused/confusing to the reader. A non-specialist reading this article would (I fear) be left bemused by this level of literacy. The Introduction is particularly poor in this respect - rather summed up by the last three sentences therein.

In-text commentary

1. Minor point. Abstr and page 2:11. "20-70 tonne" dinosaurs. These are, quite frankly, ridiculously OTT estimates of mass/weight because limb bone strength values will not support such weights under static loading on land, let alone under dynamic loading – the latter would prompt immediate failure! Such 'calculations' seriously over-estimate the density of these large animals, but are used to add 'drama' to the story (I suppose).
2. The focus is on hypo-hypa, but there is no comment about interspinous ligaments in theropods and sauropods; these are very prominent and leave complex and rugose bony scars on the leading and trailing edges of dorsal spines. These ligament bands would have acted as ossified tendon equivalents in stabilizing and strengthening the dorsal vertebral column - if we are comparing, as they do, all three dinosaur clades.
3. Page 3:17 "Many orders of magnitude" – "Many"? ... "smallest and largest" (what dimensions are the authors talking about?).
4. Persistent problem. The use of English is not consistent, or particularly good. A few trivial examples: Page 3:9 - the disparity of body sizes - you mean body size! "sizes" seems to be used ubiquitously and incorrectly. Page 4: 19 - you are not testing a "question" you are exploring the implications of an hypothesis; Page 6:2 "posterior neural arch" (anatomically that makes no literal sense) and gives the impression that there is an anterior and a posterior neural arch (which is absurd).
5. Page 6:6. Posterior zygapophyses DO NOT face dorsally – this is fundamental anatomy. Page 6:8 – reference to Figure 1. The figure is woeful, it needs to be a diagram which shows clearly the features in question and has clear annotations – and there should be a distinction between the hypo and the hypa as well as the zygapophyses (which are equally distinct) in such images – the use of the same annotation simply adds to the general 'air of confusion' in this ms.
6. Page 8:12-15. Measurements - why should length of the postzygapophysis vs centrum height create a proxy that has anything to do with size-corrected surface area of contact between vertebrae? I simply fail to understand the reasoning behind this. The logical choice would have been to use the surface area of the centrum articular surface scaled against femoral length – that does not require a, quite frankly, bizarre 'proxy'. The impression given is that 'dimensional data' are to some extent contrived.
7. Page 8: 16+ Did the same measurement collection for named crocs (2) and several birds 'normalize' for vertebral 'size'? How does that correlate with what was measured in big dinosaurs? I really don't understand the why? and what? – is going on here.
8. Page 9: first para. Wording is clumsy. And, why was such precision needed in partial vertebral columns? Again the wording and phraseology on this page are simply clumsy and imprecise. Trivia – but ... Page 9:19 "millimetres" (spelling? US/UK) – a millimeter in a UK journal would be a reference to an instrument that measures "millis"!
9. Page 10:8. What does "... angle of the postzygapophyses in relation to the neural arch" actually mean anatomically and why is it important? This is totally unclear and yet it becomes a "fair assumption". Do we need a simplified diagram/figure where all these measurements or estimations are indicated and annotated appropriately?
10. Page 10:19. "We plotted only trunk vertebrae (presacral position 10 to the last presacral before the sacrum)" - Q. Is it not clear that this is pedantic repetition? Or, worse still, a matter of literacy?

11. And on the same line ... "position 10" ... again what does that mean? - is it Dorsal 10 ... counted (from where?) established on what basis? ... or is this 'guesstimation' based upon comparison drawn from alligator and allosaur vertebral columns? A further question that arises from this is: why should it be that hyposphene-hypantra only exist in one restricted region of the dorsal column? The biomechanical implications actually seem far more interesting per se than what is being tortured in this article.

12. Page 11 - Statistical analyses - this section (again) lacks literacy. I am at a loss to explain how such garbled English can be generated in a re-read, adjusted, pored over and carefully revised article, such as this?

13. Results - Page 12 onwards. Firstly, the opening paragraph is simply a mangled version of English. Later on, the statements are inconsistent and partly contradictory (again this is made more difficult to follow and understand than it should be because of the extremely poor English usage).

I extract the last paragraph of this section: Pages 14/15:

"Our analysis of maximum length of zygapophyses as a proxy for articular surface area in extant and extinct archosaurs showed that among members of crown Crocodylia and crown Aves, relative articular surface area is roughly the same, when corrected for body size. (Figure 8, Table 2, Table 3) Additionally, we notice that for the extinct archosaur taxa we analyzed that possess the hyposphene-hypantrum (e.g., *Dilophosaurus*, *Desmotosuchus*) (Figure 9, Table 4), the more inclusive measurement of articular length (postzygapophyseal maximum length + hyposphene maximum length / centrum height) plot more closely with the extant data, whereas the less inclusive measurement (orange data points) plot below the extant data. Extinct taxa that do not have hyposphene-hypantrum articulations plot closely with the extant data. However these relationships are merely visual and we do not have enough data to run a statistic test with enough power to be meaningful. We do however believe that this is an interesting pattern worth investigating further."

Just ignoring the lazy punctuation and typos, the entire section has to be read and re-read as I try to extract meaning and sense. This isn't Kantian philosophy!

15. Discussion. Page 15 onwards. Starts off reasonably, but .. "Outside Archosauria, the hyposphene-hypantrum articulation is almost always absent, regardless of body size. The one clear exception is the presence of the accessory articulation structures in the stem archosaur *Azendohsaurus madagaskarensis* (FL = 205 mm, FMNH PR 2779, [46]). This condition differs from all archosaurs; however, because the hyposphene hypantrum articulation is only present in one vertebra in the anterior trunk and not present throughout the trunk [46].

Q. Anything wrong with this section? One, it is clumsily phrased and ungrammatical. Two, this information could be written far more succinctly. Three, the critical "almost" rather undermines the general thesis - the example described shows the feature that they define so rigidly so ... it is there! This is a significant observation because of the logical implications that flow from it. Facts that don't fit are admitted and then dismissed as irrelevant ...

16. I am sorry, but the poor grammar and syntactical errors just get more and more irritating in the Discussion section. Why was such poor English usage not weeded out a long time ago? It is intolerable in a manuscript that has evidently been through the review process for a prestigious journal ... TWICE! As an independent reviewer of what appears to be a problematic article

(judged by previous peer review comments), I find that I keep losing patience and become increasingly (no doubt unfairly) critical.

17. Page 20. The authors' 'discussion' of the structure of intervertebral articulation in titanosaurs shows no intuitive or logical consistency. It simply confuses the reader and leads to a feeling of exasperation. This is simply poor expression, and intellectually inadequate.

18. 20:17. "For example, ornithischians have been cited as possessing ossified tendons in their vertebrae" - this is anatomically incorrect and very misleading. No doubt it is yet another example of inadequate English usage. But it also sows seeds of doubt about the anatomical competence of the authors.

19. Pages 20/21: Pterosaurs - some are actually known to be HUGE! So I am not sure that these authors are being entirely truthful about their reasons for excluding this group. Equally, I do not know whether some of the HUGE pterosaurs have well-preserved dorsal vertebrae that might reveal critical features of their anatomy. But, rather like the example of birds, the stresses encountered by the vertebral column (or indeed the way they are braced) in such animals may be quite atypical and such factors are not really being considered in a biomechanical sense by these authors.

20. Logical issues. 16:10 Authors state that the common ancestor or Archosauria may have had the ability to form "the structure" (hypo-hypa). Followed by a very confused consideration of phytosaurs (if, as seems to be implied, phytosaurs are archosaurians per se).

Then Page 16:23 the presence of this trait is stated as being "ambiguously optimized at the base of Archosauria ..."

Then Page 16 the gain in this articulation is acquired independently (=convergently) in the pseudosuchian and avemetatarsalian lineages of archosaurs.

Pseudos: it is regarded as an ontogenetic phenomenon in 'paracrocodylomorphs' (because of its distribution in known specimens of Mandasuchus). Among aetosaurs it seems to be size-related (effectively the same phenomenon).

Avemetatarsalians: small examples with hypo-hypa include silesaurids and Teleocrater and small dinos: Tawa) but absent in the very small Dromomeron.

Dino clades Therops & Saurops (yes) but Orns (no).

Confounders: Titanosaurs ... no (why?) ... there is some discussion of the presence of possible hypo-hypa in early titanosaurs, and a decision is then made to consider that they don't have the fine details of hypo-hypa morphology as defined by the authors.

21. Page 22. Crocodylian procoely in dorsal vertebrae. Merited lengthy consideration of its role in stabilising intervertebral articulation. I mention this here because it highlights a deficiency in this article. Titanosaurian dinosaurs exhibit opithocoelic dorsal vertebrae - the reverse pattern of what is seen in crocodiles and yet surely a structural conformation that has the same potential benefits? Are the authors not aware of this morphology in titanosaurs?

22. Pages 24/25. Birds. There are large birds that do not have hypo-hypa. But why is that? There is no consideration of the unique thoracic construction seen in birds generally - or the ligament support of the neural spines (which resembles the ossified tendon trellises of ornithischians).

Heterocoely and intervertebral bracing in more derived birds is perfectly reasonable (and translates across to the titanosaur observation).

23. Living archosaurs lack hyposphene-hypantral articulations. But both are HIGHLY DERIVED - flight or semi-aquatic (both media subject the vertebral column to entirely different forces, and these forces are resisted in their own unique ways. These are NOT strictly comparable to the more generalised TERRESTRIAL archosaurs of the Mesozoic Era. The authors are comparing apples and oranges (to use a metaphor rather badly). And there is that rather interesting large Nile croc dorsal vertebra on my desk.

24. Figures - a simple DIAGRAMMATIC vertebra (or exemplar vertebrae) should be illustrated rather than these unclear photographic images. The diagrams can then be CLEARLY labelled (the annotations are beyond minimal given the anatomical features described in the text) so that the features being described can be seen and appreciated by the viewer. The hyposphene and hypantrum (as well as the pre- and postzygapophyses) need to be DISTINGUISHED CLEARLY IN SUCH ILLUSTRATIONS, NOT GIVEN THE SAME ANNOTATION ABBREVIATION.

25. The graphical representations of the cladograms and the data (and the analytic techniques) are - I am forced to observe - a combination of:

- i) "We'll partition things this way, manually" and then ...
- ii) "Let's use a load of stats packages because that will provide us with graphical output that makes this intuitively-driven work look 'scientific'".

Yes ... I realize that this is presented as a parody.

However, I have to observe that the series of objections and clarifications called for by Reviewer 3 points to deeper levels of their frustration about the 'what did you do?' and 'why did you do it?' ... of the results that are presented.

Decision letter (RSOS-190258.R1)

27-Jun-2019

Dear Ms Stefanic:

On behalf of the Editors, I am pleased to inform you that your Manuscript RSOS-190258.R1 entitled "The Evolution of the Hyposphene-hypantrum Articulation in Archosauria and Its Role in the Evolution of Large Body Size" has been accepted for publication in Royal Society Open Science subject to minor revision in accordance with the referee suggestions. Please find the referees' comments at the end of this email.

The reviewers and Subject Editor have recommended publication, but also suggest some minor revisions to your manuscript. Therefore, I invite you to respond to the comments and revise your manuscript.

- Ethics statement

- Data accessibility

<http://datadryad.org/submit?journalID=RSOS&manu=RSOS-190258.R1>

- Competing interests

- Authors' contributions

- Acknowledgements

- Funding statement

Because the schedule for publication is very tight, it is a condition of publication that you submit the revised version of your manuscript before 06-Jul-2019. Please note that the revision deadline will expire at 00.00am on this date. If you do not think you will be able to meet this date please let me know immediately.

Kind regards,

on behalf of Dr Julia Brenda Desojo (Associate Editor) and Kevin Padian (Subject Editor)
openscience@royalsociety.org

Reviewer comments to Author:

Reviewer: 2

All of my concerns have been addressed.

Reviewer: 4

Hyposphene-Hypantrum ms (RSOS-190258.R1)

General observations

What is the general nature of this 'hypothesis-driven' article? It is narrowly focused on their tightly constrained definition of one anatomical configuration (the hyposphene-hypantrum [hypo-hypa]) and its distribution within sub-clades of archosaurian reptiles.

This promotes the idea that the hypo-hypa is a solely size-related construct (i.e. they are analogues randomly distributed across, and restricted to, large-sized archosaurians). The thrust of their argument is that there is no underlying phylogenetic component to their appearance/distribution in individual lineages/clades. Some do, some don't - and if they don't they have alternative mechanisms to solve the 'problems' of rigging and bracing large skeletons.

This 'notion' is backed up by statistical analyses and graphical presentations that are purporting to support this general thesis.

Unfortunately, this general thesis is undermined by their observation that basal (=ancestral) archosaurs inherited a common tendency to develop these structures (page 16:10), which are then expressed, or not expressed, later in their phylogeny. This is not an inspiring juxtaposition of concepts. If hypo-hypa are indeed part of a common inheritance then these features are arguably homologous (as the intricate nature of their similarity across taxa implies) rather than analogous, as the tenor of this article seems to wish to imply.

Figures. Annotation abbreviations in the text would be appropriate if the reader wants to cross-reference anatomy that is being described textually with what has been CLEARLY(?) Illustrated. But, the illustrations are inadequate and confusingly annotated.

As a trivial point of fact, I have a dorsal vertebra of a quite large *Crocodylus niloticus* on my desk (as I type this), which has a structure not unlike a hyposphene between its postzygapophyses! However, it is more of a functional/structural analogue than a feature that fits into their narrow definition of what a hypo is, of course.

First impression: this article is not at all well written and would need to be carefully re-written throughout. I thought it was just the abstract/intro, but this problem is pervasive. The language is occasionally inaccurate, syntactically challenged and quite often confused/confusing to the reader. A non-specialist reading this article would (I fear) be left bemused by this level of literacy. The Introduction is particularly poor in this respect - rather summed up by the last three sentences therein.

In-text commentary

1. Minor point. Abstr and page 2:11. "20-70 tonne" dinosaurs. These are, quite frankly, ridiculously OTT estimates of mass/weight because limb bone strength values will not support

such weights under static loading on land, let alone under dynamic loading – the latter would prompt immediate failure! Such ‘calculations’ seriously over-estimate the density of these large animals, but are used to add ‘drama’ to the story (I suppose).

2. The focus is on hypo-hypa, but there is no comment about interspinous ligaments in theropods and sauropods; these are very prominent and leave complex and rugose bony scars on the leading and trailing edges of dorsal spines. These ligament bands would have acted as ossified tendon equivalents in stabilizing and strengthening the dorsal vertebral column - if we are comparing, as they do, all three dinosaur clades.

3. Page 3:17 “Many orders of magnitude” – “Many”? ... “smallest and largest” (what dimensions are the authors talking about?).

4. Persistent problem. The use of English is not consistent, or particularly good. A few trivial examples: Page 3:9 - the disparity of body sizes - you mean body size! “sizes” seems to be used ubiquitously and incorrectly. Page 4: 19 - you are not testing a “question” you are exploring the implications of an hypothesis; Page 6:2 “posterior neural arch” (anatomically that makes no literal sense) and gives the impression that there is an anterior and a posterior neural arch (which is absurd).

5. Page 6:6. Posterior zygapophyses DO NOT face dorsally – this is fundamental anatomy. Page 6:8 – reference to Figure 1. The figure is woeful, it needs to be a diagram which shows clearly the features in question and has clear annotations – and there should be a distinction between the hypo and the hypa as well as the zygapophyses (which are equally distinct) in such images – the use of the same annotation simply adds to the general ‘air of confusion’ in this ms.

6. Page 8:12-15. Measurements - why should length of the postzygapophysis vs centrum height create a proxy that has anything to do with size-corrected surface area of contact between vertebrae? I simply fail to understand the reasoning behind this. The logical choice would have been to use the surface area of the centrum articular surface scaled against femoral length – that does not require a, quite frankly, bizarre ‘proxy’. The impression given is that ‘dimensional data’ are to some extent contrived.

7. Page 8: 16+ Did the same measurement collection for named crocs (2) and several birds ‘normalize’ for vertebral ‘size’? How does that correlate with what was measured in big dinosaurs? I really don’t understand the why? and what? – is going on here.

8. Page 9: first para. Wording is clumsy. And, why was such precision needed in partial vertebral columns? Again the wording and phraseology on this page are simply clumsy and imprecise. Trivia – but ... Page 9:19 “millimetres” (spelling? US/UK) – a millimeter in a UK journal would be a reference to an instrument that measures “millis”!

9. Page 10:8. What does “... angle of the postzygapophyses in relation to the neural arch” actually mean anatomically and why is it important? This is totally unclear and yet it becomes a “fair assumption”. Do we need a simplified diagram/figure where all these measurements or estimations are indicated and annotated appropriately?

10. Page 10:19. “We plotted only trunk vertebrae (presacral position 10 to the last presacral before the sacrum)” - Q. Is it not clear that this is pedantic repetition? Or, worse still, a matter of literacy?

11. And on the same line ... “position 10” ... again what does that mean? - is it Dorsal 10 ... counted (from where?) established on what basis? ... or is this ‘guesstimation’ based upon comparison drawn from alligator and allosaur vertebral columns? A further question that arises

from this is: why should it be that hyposphene-hypantra only exist in one restricted region of the dorsal column? The biomechanical implications actually seem far more interesting per se than what is being tortured in this article.

12. Page 11 - Statistical analyses - this section (again) lacks literacy. I am at a loss to explain how such garbled English can be generated in a re-read, adjusted, pored over and carefully revised article, such as this?

13. Results - Page 12 onwards. Firstly, the opening paragraph is simply a mangled version of English. Later on, the statements are inconsistent and partly contradictory (again this is made more difficult to follow and understand than it should be because of the extremely poor English usage).

I extract the last paragraph of this section: Pages 14/15:

“Our analysis of maximum length of zygapophyses as a proxy for articular surface area in extant and extinct archosaurs showed that among members of crown Crocodylia and crown Aves, relative articular surface area is roughly the same, when corrected for body size. (Figure 8, Table 2, Table 3) Additionally, we notice that for the extinct archosaur taxa we analyzed that possess the hyposphene-hypantrum (e.g., *Dilophosaurus*, *Desmotosuchus*) (Figure 9, Table 4), the more inclusive measurement of articular length (postzygapophyseal maximum length + hyposphene maximum length / centrum height) plot more closely with the extant data, whereas the less inclusive measurement (orange data points) plot below the extant data. Extinct taxa that do not have hyposphene-hypantrum articulations plot closely with the extant data. However these relationships are merely visual and we do not have enough data to run a statistic test with enough power to be meaningful. We do however believe that this is an interesting pattern worth investigating further.”

Just ignoring the lazy punctuation and typos, the entire section has to be read and re-read as I try to extract meaning and sense. This isn't Kantian philosophy!

15. Discussion. Page 15 onwards. Starts off reasonably, but .. “Outside Archosauria, the hyposphene-hypantrum articulation is almost always absent, regardless of body size. The one clear exception is the presence of the accessory articulation structures in the stem archosaur *Azendohsaurus madagaskarensis* (FL = 205 mm, FMNH PR 2779, [46]). This condition differs from all archosaurs; however, because the hyposphene hypantrum articulation is only present in one vertebra in the anterior trunk and not present throughout the trunk [46].

Q. Anything wrong with this section? One, it is clumsily phrased and ungrammatical. Two, this information could be written far more succinctly. Three, the critical “almost” rather undermines the general thesis – the example described shows the feature that they define so rigidly so ... it is there! This is a significant observation because of the logical implications that flow from it. Facts that don't fit are admitted and then dismissed as irrelevant ...

16. I am sorry, but the poor grammar and syntactical errors just get more and more irritating in the Discussion section. Why was such poor English usage not weeded out a long time ago? It is intolerable in a manuscript that has evidently been through the review process for a prestigious journal ... TWICE! As an independent reviewer of what appears to be a problematic article (judged by previous peer review comments), I find that I keep losing patience and become increasingly (no doubt unfairly) critical.

17. Page 20. The authors' 'discussion' of the structure of intervertebral articulation in titanosaurs shows no intuitive or logical consistency. It simply confuses the reader and leads to a feeling of exasperation. This is simply poor expression, and intellectually inadequate.

18. 20:17. "For example, ornithischians have been cited as possessing ossified tendons in their vertebrae" - this is anatomically incorrect and very misleading. No doubt it is yet another example of inadequate English usage. But it also sows seeds of doubt about the anatomical competence of the authors.

19. Pages 20/21: Pterosaurs - some are actually known to be HUGE! So I am not sure that these authors are being entirely truthful about their reasons for excluding this group. Equally, I do not know whether some of the HUGE pterosaurs have well-preserved dorsal vertebrae that might reveal critical features of their anatomy. But, rather like the example of birds, the stresses encountered by the vertebral column (or indeed the way they are braced) in such animals may be quite atypical and such factors are not really being considered in a biomechanical sense by these authors.

20. Logical issues. 16:10 Authors state that the common ancestor or Archosauria may have had the ability to form "the structure" (hypo-hypa). Followed by a very confused consideration of phytosaurs (if, as seems to be implied, phytosaurs are archosaurians per se).

Then Page 16:23 the presence of this trait is stated as being "ambiguously optimized at the base of Archosauria ..."

Then Page 16 the gain in this articulation is acquired independently (=convergently) in the pseudosuchian and avemetatarsalian lineages of archosaurs.

Pseudos: it is regarded as an ontogenetic phenomenon in 'paracrocodylomorphs' (because of its distribution in known specimens of Mandasuchus). Among aetosaurs it seems to be size-related (effectively the same phenomenon).

Avemetatarsalians: small examples with hypo-hypa include silesaurids and Teleocrater and small dinos: Tawa) but absent in the very small Dromomeron.

Dino clades Therops & Saurops (yes) but Orns (no).

Confounders: Titanosaurs ... no (why?) ... there is some discussion of the presence of possible hypo-hypa in early titanosaurs, and a decision is then made to consider that they don't have the fine details of hypo-hypa morphology as defined by the authors.

21. Page 22. Crocodylian procoely in dorsal vertebrae. Merited lengthy consideration of its role in stabilising intervertebral articulation. I mention this here because it highlights a deficiency in this article. Titanosaurian dinosaurs exhibit opithocoelic dorsal vertebrae - the reverse pattern of what is seen in crocodiles and yet surely a structural conformation that has the same potential benefits? Are the authors not aware of this morphology in titanosaurs?

22. Pages 24/25. Birds. There are large birds that do not have hypo-hypa. But why is that? There is no consideration of the unique thoracic construction seen in birds generally - or the ligament support of the neural spines (which resembles the ossified tendon trellises of ornithischians). Heterocoely and intervertebral bracing in more derived birds is perfectly reasonable (and translates across to the titanosaur observation).

23. Living archosaurs lack hyposphene-hypantral articulations. But both are HIGHLY DERIVED - flight or semi-aquatic (both media subject the vertebral column to entirely different forces, and these forces are resisted in their own unique ways. These are NOT strictly comparable to the more generalised TERRESTRIAL archosaurs of the Mesozoic Era. The authors are comparing apples and oranges (to use a metaphor rather badly). And there is that rather interesting large Nile croc dorsal vertebra on my desk.

24. Figures - a simple DIAGRAMMATIC vertebra (or exemplar vertebrae) should be illustrated rather than these unclear photographic images. The diagrams can then be CLEARLY labelled (the annotations are beyond minimal given the anatomical features described in the text) so that the features being described can be seen and appreciated by the viewer. The hyposphene and hypantrum (as well as the pre- and postzygapophyses) need to be DISTINGUISHED CLEARLY IN SUCH ILLUSTRATIONS, NOT GIVEN THE SAME ANNOTATION ABBREVIATION.

25. The graphical representations of the cladograms and the data (and the analytic techniques) are - I am forced to observe - a combination of:

- i) "We'll partition things this way, manually" and then ...
- ii) "Let's use a load of stats packages because that will provide us with graphical output that makes this intuitively-driven work look 'scientific'".

Yes ... I realize that this is presented as a parody.

However, I have to observe that the series of objections and clarifications called for by Reviewer 3 points to deeper levels of their frustration about the 'what did you do?' and 'why did you do it?' ... of the results that are presented.

Author's Response to Decision Letter for (RSOS-190258.R1)

See Appendix C.

Decision letter (RSOS-190258.R2)

14-Aug-2019

Dear Ms Stefanic:

On behalf of the Editors, I am pleased to inform you that your Manuscript RSOS-190258.R2 entitled "The Evolution and Role of the Hyposphene-hypantrum Articulation in Archosauria: Phylogeny, Size, and/or Mechanics?" has been accepted for publication in Royal Society Open Science subject to minor revision in accordance with the referee suggestions. Please find the referees' comments at the end of this email.

The reviewers and Subject Editor have recommended publication, but also suggest some minor revisions to your manuscript. Therefore, I invite you to respond to the comments and revise your manuscript.

- Ethics statement

If your study uses humans or animals please include details of the ethical approval received, including the name of the committee that granted approval. For human studies please also detail

whether informed consent was obtained. For field studies on animals please include details of all permissions, licences and/or approvals granted to carry out the fieldwork.

- Data accessibility

If you wish to submit your supporting data or code to Dryad (<http://datadryad.org/>), or modify your current submission to dryad, please use the following link:
<http://datadryad.org/submit?journalID=RSOS&manu=RSOS-190258.R2>

- Competing interests

- Authors' contributions

- Acknowledgements

- Funding statement

Because the schedule for publication is very tight, it is a condition of publication that you submit the revised version of your manuscript before 23-Aug-2019. Please note that the revision

deadline will expire at 00.00am on this date. If you do not think you will be able to meet this date please let me know immediately.

on behalf of Dr Julia Brenda Desojo (Associate Editor) and Kevin Padian (Subject Editor)
openscience@royalsociety.org

Author's Response to Decision Letter for (RSOS-190258.R2)

See Appendix D.

Decision letter (RSOS-190258.R3)

22-Aug-2019

Dear Ms Stefanic,

I am pleased to inform you that your manuscript entitled "The Evolution and Role of the Hyposphene-hypantrum Articulation in Archosauria: Phylogeny, Size, and/or Mechanics?" is now accepted for publication in Royal Society Open Science.

on behalf of Dr Julia Brenda Desojo (Associate Editor) and Kevin Padian (Subject Editor)
openscience@royalsociety.org

Appendix A

06-Jul-2018

Dear Ms Stefanic:

Manuscript ID RSOS-180579 entitled "The Evolution of the Hyposphene-hypantrum Articulation in Archosauria and Its Role in the Evolution of Large Body Size" which you submitted to Royal Society Open Science, has been reviewed. The comments from reviewers are included at the bottom of this letter.

In view of the criticisms of the reviewers, the manuscript has been rejected in its current form. However, a new manuscript may be submitted which takes into consideration these comments.

Please note that resubmitting your manuscript does not guarantee eventual acceptance, and that your resubmission will be subject to peer review before a decision is made.

Your resubmitted manuscript should be submitted by 03-Jan-2019. If you are unable to submit by this date please contact the Editorial Office.

Please note that Royal Society Open Science will introduce article processing charges for all new submissions received from 1 January 2018. Charges will also apply to papers transferred to Royal Society Open Science from other Royal Society Publishing journals, as well as papers submitted as part of our collaboration with the Royal Society of Chemistry (<http://rsos.royalsocietypublishing.org/chemistry>). If your manuscript is submitted and accepted for publication after 1 Jan 2018, you will be asked to pay the article processing charge, unless you request a waiver and this is approved by Royal Society Publishing. You can find out more about the charges at <http://rsos.royalsocietypublishing.org/page/charges>. Should you have any queries, please contact openscience@royalsociety.org.

Kind regards,
Andrew Dunn
Senior Publishing Editor
Royal Society Open Science
openscience@royalsociety.org

on behalf of Dr Julia Brenda Desojo (Associate Editor) and Kevin Padian (Subject Editor)
openscience@royalsociety.org

Associate Editor Comments to Author (Dr Julia Brenda Desojo):

Comments to the Author:

Dear authors, this is an interesting contribution about the evolution of accessory articulations in archosaurs, but the manuscript should be improve because the conclusions are not justified by the results in the present form. Particularly, the presence of Hyposphene/Hypantrum in larger-bodied taxa contradict the author hypothesis proposed that the acquisition of these accessory articulations was correlated with increases in body size. In order to resolve it, additional taxa, such as large ornithischians, pterosaurs and titanosaurs should be incorporate to the analysis. After that, a reanalysis of the functional explanations for the loss of these articulation should be reevaluate them. The statistical analyses need to be expanded and the clarity and layout of the text need to be significantly improved.

Reviewers' Comments to Author:

Reviewer: 1

Comments to the Author(s)

General comments

Hyposphene/hypantrum articulations are commonly found in extinct archosaurs, but are absent from living representatives of the clade. The authors test the hypothesis that the acquisition of these accessory articulations was correlated with increases in body size (and the converse that they were lost during periods of subsequent miniaturisation) by comparing character optimisations of hyposphene/hypantrum presence/absence with ancestral state reconstructions of a body mass proxy (femoral length). They find a strong correlation between the two. However, although I am generally sympathetic to the author's case, as the presence of these articulations in larger-bodied taxa makes intuitive biomechanical sense, I think that the MS requires some additional work before it can be accepted.

We have been careful in reworking how we are framing this paper based on reviewer feedback, our study is all about showing correlation between the presence/absence of the hyposphene-hypantrum articulation and body size, not about providing an explanation. We removed the entire Biomechanical Implications section (p.) and the sentence on p. 1 "We posit that this articulation increased vertebral bracing and therefore facilitated the evolution of large body sizes in extinct archosaurs."

The analyses presented are incomplete and would benefit from the inclusion of additional taxa (large ornithischians and pterosaurs would be obvious additions). In addition, it's unclear how ancestral body femoral lengths are derived and more clarity on the evenness of taxon sampling with respect to body size would make some of the assumptions implicit in the dataset clearer. In some ways, the taxon sample selected for the analysis looks biased in a way that would provide their preferred result and although the addition of other taxa might not overturn it, they might reduce its statistical significance or explanatory power. In addition, although the authors prefer to infer functional explanations to phylogenetic ones when examining the observed pattern of hyposphene/hypantrum presence/absence, it would be difficult to disentangle these effects as the functional changes optimise at key nodes within archosaur phylogeny (and there is obvious phylogenetic signal in the autapomorphic loss of these features, and their continued absence, in even large bodied ornithischians).

Some functional explanations for the losses of these articulations are not explored, but are likely to be important when considering the evolution of this feature in archosaurs as a whole. As the authors note, there are complex patterns of gain and loss, but by effectively ignoring ornithischians, pterosaurs and titanosaurs, they miss an opportunity to come up with a more holistic explanation. As noted below, the evolution of other vertebral bracing structures (e.g. ossified tendons in ornithischians) might have been key in explaining some of these patterns, but taxa that are not congruent with their main hypothesis are overlooked, whereas these could actually provide some circumstantial support for their thesis.

We agree with the reviewer that it is important not to ignore taxa that are exceptions to our main hypothesis, so we made sure to add in mention of ossified tendons in ornithischians being a secondary bracing mechanism that allowed them to get to large body sizes without the gain of the hyposphene-hypantrum. Additionally, we mention that titanosaur vertebrae have an extremely derived morphology (e.g., extensive pneumaticity) that we also posit that this is a secondary method of growing to large sizes without the addition of the hyposphene-hypantrum.

Detailed comments

p. 1, lines 26–28. All body masses provided should be in metric tonnes, not tons. Please correct throughout the MS. **Fixed.**

p. 2, lines 16–17. Indian elephants generally don't get above 5 tonnes, whereas bull African elephants do get up to 7 tonnes, though this is still relatively rare. I'd suggest changing the species here as it's generally recognised that African bush (savannah) elephants (*Loxodonta africana*) are the largest living terrestrial vertebrates. If you do decide to stick with Indian elephants, the correct spelling is *Elephas maximus*.

Fixed, kept as *Elephas maximus*.

p. 2, lines 26–28. There's been work on the evolution of large body size in sauropods in particular, with Martin Sander's group producing a number of papers on links between sauropod biology and body size, as summarised in Sander et al. (2011) in *Biol. Rev.* and also in Klein et al. (2011) *Biology of the Sauropod Dinosaurs*. **Thanks, added in these citations.**

p. 2, lines 29–31. You're right, but it would be useful to provide some examples of these skeletal changes with body size (e.g., relative robustness, etc.), otherwise this is a straw man. You list some of these changes below, but you could integrate these parts of the intro more closely. **Reworded this sentence to be much more general and to not mention specific changes until where examples are given and cited below.**

p. 2, line 37. I'd suggest 'model system' rather than just 'model'. **Changed to "model system"**

p. 2, line 45. Also in terms of overall robustness, patterns of scaling, limb bone eccentricity, etc. Some of these things change ontogenetically with body size within species too. **Yes, we agree.**

p. 3, line 5. Give a couple of examples of smaller dinosaurs that lack these features.

Done. Added: "(e.g., *Rahonavis*, *Mahakala*, *Microvenator*, *Mononykus*, *Parvicursor*)"

p. 3, lines 9–10. It depends how you define 'small' and 'large'. Teleocrater and Effigia possess hyposphene/hypantra, but are both relatively small by archosaurian standards (the modal mass of a non-avian dinosaur is around 1 tonne and these taxa get nowhere near this size). They're both certainly a lot smaller than many ornithischians that lack these features. Also, zygosphenes and zygantra (which function to more closely integrate vertebrae, just in a slightly different way morphologically) are found in a numerous small-bodied taxa (snakes, lizards). We agree that it is important to have a distinction between what "small" and "large" mean, and we generally refer to small-bodied taxa as being below the thresholds for presence of the hyposphene-hypantrum (i.e. 130 mm femur length for avemetatarsalians, 230 mm femur length for pseudosuchians). In this section we changed the wording to "smaller end of the archosaur spectrum" to make our statement more clear.

p. 3, line 26. To some extent, the absence of hyposphene/hypantrum articulations in large-bodied ornithischians already falsifies this hypothesis, at least in part, without needing to test it quantitatively.

Added: "with a few exceptions (e.g., *Ornithischia*, *Aves*, *Crocodylia*)"

p. 4, lines 40–47 and p. 5, lines 2–13. In addition to surveying taxa that are spread across the tree, thus sampling across phylogeny, did you also survey representative taxa from different size classes? Although the taxa and their femoral lengths are noted in a table, it might be good to include some kind of summary diagram that makes it clearer how even your sampling is across size classes and the tree to show your data are not biased towards particular clades or body sizes. For example, visual inspection of your list indicates that very small bodied taxa (e.g., small ornithischians, pterosaurs: femoral lengths <100 mm) are not as extensively sampled as larger-bodied taxa. I'm also surprised that pterosaurs were excluded: maybe the rationale for this should be mentioned? Finally, within the large-bodied taxon sample there are distinct phylogenetic skews: large-bodied ornithischian taxa are severely undersampled (two taxa), whereas there are 10 large-bodied sauropodomorphs (but no small ones, such as *Saturnalia*). In reality there's much more overlap in the sizes of non-avian dinosaurs that lack hyposphenes/hypantra with those that possess this feature than summarised in Figure 1B.

We agree with the reviewer that this would be important to include. We tried to include as many taxa across the tree in clades that you see changes in. Unfortunately, tiny taxa (femoral length < 100mm) that the reviewer has suggested are not available because 1) it is very difficult to see between the vertebrae and 2) the preservation of these small forms make identifying the absence or presence very difficult. We included the small taxa that we were confident the presence or absence of the structures could be scored.

We added a sentence about why pterosaurs were excluded in the methods (p. 13): "We also chose to effectively exclude pterosaurs as a whole because we could not confirm the presence or absence of the hyposphene-hypantrum in any of the specimens of early pterosaurs. This is because most are preserved as flattened slabs and are so small that even in uCT it is virtually impossible to see between the vertebrae to confidently score presence or absence of the hyposphene-hypantrum articulation."

The addition of more ornithischians would not alter our findings because we focused on parts of the tree where there were changes in the presence/ absence (i.e., gains or losses) of the hyposphene-hypantrum articulation and it is clear that the group only loses the articulation once, at the base of the clade, and no ornithischian has been reported to gain it.

We added *Saturnalia* (small sauropodomorph with h-h articulation) to Table 1.

p. 5, lines 33–35. It's not clear how these ancestral sizes were calculated. Have all of the ancestral values you used been lifted from the literature or have you calculated these ancestral values yourselves? In either case, how were they calculated? Squared-change parsimony or something else? Also, if the ancestral sizes were taken from the literature these references mention only some theropods and pseudosuchians – how were the ancestral sizes calculated for members of the other clades you include? We reworded the sentence this comment is referring to in order to make it more clear that we took ancestral size values from previously published literature (Turner and Nesbitt 2013 and Lee et al. 2014).

p. 6, lines 6–18. You are not quantifying the surface area of the articulation here, but the length of the articulation (depending on the shape of the articular surface there could be wide variation in the actual amount of contact for the same length). Would be better to change your terminology throughout the MS to talk about this measure of postzygapophyseal length as a proxy for articular area and to note the pros and cons of using this measurement rather than surface area per se. Added "...we obtained a measurement that could be used as a proxy for relative surface area of articulation between vertebrae"

p. 6, lines 6–18. You mention that you also gathered data on the hyposphenes, but don't mention how you got these measurements or whether they were also normalised in some way. Or do you include the measurement of the hyposphene within your overall measurement of postzygapophysis length? It's not clear how any the additional articular length from the hypotheses is being captured in your analyses. Clarified how the measurements were taken of postzygapophyses and

hyosphenes

p. 9, line 36. You could refer to Figure 1 here. **Added.**

p. 10, lines 22–26. It would be interesting to see what happened if you added in data from a few more large-bodied ornithischians here. I suspect the R2 would go down and there would definitely be a much, much larger ‘transitional’ range in Fig. 1B for avemetarsalians. **We talk about why we excluded all ornithischians from our statistical analysis on page 12.**

p. 11, lines 15–54. The derived lithostrotian titanosaur *Opisthocoelicaudia* has rather nice dorsals and no trace of hyosphene/hypantrum articulations. I think it’s disingenuous to regard their absence as ‘ambiguous’ in this clade when all of the available evidence suggests that they were genuinely absent. It would be better to regard this a real anomaly on the basis of current data rather than trying to imply that they might have been there, simply as their absence isn’t congruent with the rest of your thesis on the relationship between body size and hyosphene/hypantrum presence/absence. Also, large ornithischians (e.g., multi-tonned hadrosaurs and ceratopsians, many of which are larger than some sauropods and all but the very largest theropods) lack this feature, but simply aren’t included in your analysis. As a result, body size might be part of the problem, but it can’t be the only factor, nor can it be that critical if the very largest archosaurs live without them.

We agree with the reviewer that titanosaurs appear to not have hyosphene-hypantrum articulation. We have changed our wording in the manuscript from “ambiguous” to “absent”. Due to the very derived morphology of titanosaurs (see our explanation on p. 12), we are choosing to exclude them in our statistical analyses of hyosphene-hypantrum presence/absence as it related to body size because the group as a whole is an exception, the same way ornithischians are, and the analysis focuses on groups that gain and lose the structure throughout their phylogenetic history: Pseudosuchia and Avemetatarsalia.

p. 12, lines 3–22. It would be interesting to include titanosaurs in these regressions to see what effect they have on the overall results and to compare the regressions with and without them. At the moment, the analyses are being stacked to make body size look more important by removing large taxa that lack these features (titanosaurs, ornithischians). This isn’t a thorough test of the hypothesis. You are making an a priori judgement about the role of phylogeny here also, rather than testing it, although elsewhere in the paper you say phylogeny doesn’t have a strong effect. Quite a bit of circular reasoning. **Added a sentence about “ossified tendons” (p. 12-13) in Ornithischia and justification for ignoring them in our analyses. Also added further justification for why titanosaurs are excluded (derived morphology).**

p. 12, lines 43 onward. Again, I’m not sure how you quantified hyosphene size here. **Clarified this on p. 6 by adding: “Postzygapophyses were measured from the point where they contact the neural arch to their distal-most aspect. Hyosphenes were measured along the entire aspect of their lateral faces.” And “we obtained a measurement that could be used as a proxy for relative surface area of articulation between vertebrae”**

p. 14, lines 53 onward. Figures 2 and 3 suggest that the presence of hyosphene/hypantra is clearly convergent on the basis of character optimisations, without the need to invoke body size as another factor, and that it was not present in the archosaur common ancestor, contrary to the statement in the text.

We agree with the reviewer that the presence of the hyosphene-hypantrum is convergent across many clades; this is based on a strict reading of the tree. However, we suggest that the reason that you see this convergence is best explained by body size.

p. 16, lines 31–45. Early ornithischians are comparable in size to *Tawa* and *Asilisaurus* in femoral length, but lack hyosphenes/hypantra. Interesting that other dinosaurs gain these features at small body sizes, but ornithischians don’t. This has to be phylogenetic signal, as you suggest in passing elsewhere, but is worthy of more comment.

We have added to the text to make the case that ornithischians do not fit the pattern of gaining the hyosphene-hypantrum when increasing body size across the threshold present in other dinosaurs and relatives.. Either ornithichians as a group never had them or lost them early in evolution at or very near the base of the clade. Importantly, the change only happened once. In fact, heterodontosaurids (*Heterodontosaurus* specifically, we added a citation on this on p. 13 when we mention ossified tendons: Luca et al. 1976) already have ossified tendons. So, it may be no coincidence that the absence of the hyosphene-hypantrum and the presence of ossified tendons shows up at the same place if the ossified tendons are a secondary bracing mechanism that is likely providing the same type of biomechanical support for larger body size like we now mention in this resubmission.

p. 17, lines 3–8. How would this optimisation change if you used an alternative dinosaur phylogeny that included either Ornithoscelida or Phytodinosauria (e.g. Baron et al., 2017; Langer et al. 2017; Parry et al. 2017)? If Ornithoscelida a real entity this suggests that these features are primitive for Dinosauria (and then lost in Ornithischia), not just a saurischian feature.

Based on different phylogenies, this should not matter much because it is about loss and gain (presence/absence) of the hyosphene-hypantrum. The focus of our study was correlating presence/absence with body size in the groups that possess

the feature. No matter which phylogeny we used, the presence of the hyposphene-hypantrum would not necessarily be primitive for Dinosauria; it either could have been a dinosaur synapomorphy, and lost in ornithischians, or it could have been gained independently in saurischians. Either way, our correlation between presence and absence of the feature with large body size would not differ from the results we found no matter which phylogeny we used. We added a line on p.18 that says: "Other phylogenetic hypotheses like that of Baron et al. [66] for early dinosaurs (e.g., ornithischians are more closely related to theropods than either is to sauropodomorphs) would not change this result when you consider the number of state changes (i.e., presence or absence) and femoral length."

pp. 20–22. A number of the same measurements are repeated several times in successive paragraphs, which doesn't seem necessary. **Duplicate measurements have been eliminated.**

pp. 22-24. It's interesting that you don't mention two other obvious bracing systems of archosaurs: osteoderms and ossified tendons. It's been suggested that the osteoderms of crown crocodylians act as a bracing system (and they lack hyposphenes) and also that the ossified tendons present in all ornithischians also provided this function (and they lack hyposphenes). The lack of these features in these taxa could reflect the acquisition of these other bracing systems also. It seems like something that would be important to consider in ornithischians especially, but it's not mentioned here at all. Miniturisation might not have been necessary in the loss of this system at all – it might simply have been replaced by other features at larger body sizes. **We added mention of ossified tendons in ornithischians on p. 1 and p. 12. Osteoderms are mentioned on p. 19 and p. 26.**

Figs 2–5. You include Squamata and Phytosauria in your trees, but there are no data on body size or the presence/absence of hyposphenes in these animals in your data tables. These data should be included. Also, I'd argue that squamates have something very similar to these features, even though the name differs. If you think that the zygantra/zygosphenes of squamates are not homologous with hyposphene/hypantra I'd be tempted to exclude them from your analysis here. **The zygosphene-zygantrum articulation is not homologous with the hyposphene-hypantrum so we do exclude them from our analysis. Squamata is included in our trees solely for context. We added a sentence in the Materials and Methods section under (c) Phylogenetic survey: "We include the group Squamata on our phylogenies for context and to illustrate that within Reptilia no other taxa outside of Archosauria possess the hyposphene-hypantrum articulation. While many reptiles do have an accessory intervertebral articulation, the zygosphene-zygantrum articulation, these structures are not homologous to the hyposphene-hypantrum."**

Figs 2–5. I'm unclear why some of the lineages (e.g. Avemetatarsalia in Fig. 2 or Pseudosuchia in Fig. 3) have different colours at different points along their lengths. If using maximum parsimony surely there should simply be an extrapolation of the ancestral state for that clade along the entire length of the branch from the tip to the node. I understand that this might be to capture some information on the presence/absence of the feature within each clade each time this is done, but as you're really only interested in the condition at the base of each clade as an outgroup in each case, adding this extra nuance in just makes it look confusing (and the nuances within each clade are captured in the more expanded trees you have for Pseudosuchia/Avemetatarsalia in successive figures anyhow).

We are unsure of what the reviewer is requesting with this comment.

No list of institutional abbreviations is provided for the various specimens listed in the MS and the tables. **A section of Institutional Abbreviations has been added after the Introduction and before Materials and Methods.**

Paul M. Barrett

Reviewer: 2

Comments to the Author(s)

This is a well-written, clean manuscript that looks at the evolution of the hyposphene-hypantrum articulations in archosaurs, an important vertebral accessory articulation that was once thought to be restricted to dinosaurs. The authors argue that the presence of this structure is convergent between groups, and mostly tied to large body size; however, its lack in large bodied groups such as the ornithischian dinosaurs, as well as crown group archosaurs suggest that there is some phylogenetic significance as well. This conflict is addressed by the hypothesis that large animals require vertebral bracing and these articulations served this purpose, but that in groups lacking them they accomplished this bracing in other ways. I feel this argument is a little weak, for reasons found in the following comments, but overall the presence in larger individuals does appear to be for the most part supported.

Other comments:

Page 3, line 8: The possibly first recognition of these structures in a pseudosuchian archosaur was by Parker (2003; 2008) for a large specimen of the aetosaur *Desmatosuchus*. **OK, added in Parker 2008 citation here and re-numbered citations because that paper was in the References as #60.**

Page 9, Lines 38-39: Yes, there appears to be a relationship between size and presence of the structure, but only in lineages where it has been determined to be present. I don't recall mention of a species level taxon where the structure is present in larger individuals but not smaller ones. At Petrified Forest there is a newly collected giant *Typhothorax* that clearly has the complex; however, it is also present in a much smaller specimen in the same collections. These specimens should

be added to the database and the structures marked as present in *Tyothorax*. To be fair the very large specimen was not available at the time of the study, and the smaller specimen was confusing until the large specimen made it clear what is present. Because the large specimen was not available at the time of our data collection and we will not have a chance to see it in person before resubmission of this manuscript, we did not add it to our database for this study. Additionally, in order to add it to our logistic regression (Figure 4) it would need to have an associated femur. We would be interested in taking a look at this specimen though and the fact that a large *Tyothorax* does possess the hyposphene-hypantrum would support our hypothesis presented in this study (assuming the specimen falls above our pseudosuchian body size threshold) and would mean that there is an additional independent gain of the structure in aetosaurs.

Page 9, line 45: This sentence reads awkwardly. Reworded the sentence.

Page 9, line 44-47: the association in femur length is hard to test in some specimens, such as MNA V9300, where the femur is unknown. Added: "It is important to note that some specimens could not be included in this analysis because no femora are known for the taxon (e.g., *Desmatosuchus spurensis*, MNA V9300)."

Page 12, line 15: missing a word in: may be a phylogenetic constraint. Fixed.

Page 16, line 4: The structure is also present in *Tyothorax*. *Aetobarbakinoides* has a very similar structure to what is seen in *Scutarx deltatylus* and described and discussed by Parker (2016). So either what is in those taxa is not a hyposphene-hypantrum, or the complex is in some smaller sized animals. Although the structures are similar, based on our definition put forth in Stefanic and Nesbitt (2018), the structures in *Aetobarbakinoides* and *Tyothorax* are not true hyposphene-hypantrum articulations

Page 22, line 34: Some pseudosuchians (*Desmatosuchus*, shuvosaurids) and early diverging theropods also ankylose additional vertebrae. How does this affect your hypothesis? We don't think this changes our hypothesis because only one or possibly two vertebrae are incorporated into the sacrum from the trunk series. In this case, only a small percentage (>10%) of vertebrae with the hyposphene-hypantrum are affected. Sacral fusion does not correlate closely with size (Nesbitt 2011), although generally, larger animals have fusion in the sacrum.

Page 24, line 42: awkward phrasing. Reworded.

Page 25, line 25: some large aetosaurs (*Desmatosuchus*, *Longosuchus*) fuse their cervical and anterior trunk armor, paramedian and lateral. That is an interesting observation.

Page 41: line 8: aetosaur phylogeny from Parker (2016). Added citation to figure caption.

Figure 2: structure present in *Tyothorax* and *Aetobarbakinoides*. If what is in *Aetobarbakinoides* is not a hyposphene/hypantrum, then it is also not present in *Scutarx*. We looked closely at all of these specimens and determined that the structure present in *Aetobarbakinoides* and *Tyothorax* do not satisfy our definition of the hyposphene-hypantrum articulation (see Stefanic and Nesbitt 2018), but *Scutarx* does.

Figure 3: How would the hypothesis that silesaurids are early ornithischians affect your hypothesis? This should be mentioned. That would mean that our data suggest that the common ancestor of Dinosauria possessed a hyposphene-hypantrum articulation and that it was lost in 'classic/core' Ornithischia instead of gained in Saurischia.

Figure 4: How was femur length determined for aetosaurs, most of which lack femora? *Desmatosuchus smalli* has a femur as does *Longosuchus* and *Tyothorax* (AMNH). Ancestral femoral length for Aetosauria was taken from Turner and Nesbitt (2013).

Figure 7: Please better explain your method for determining position of the trunk vertebrae based on measurements of the diapophyses and parapophyses. That would be very useful and also needs to be testable. We looked at complete columns and estimated where the fossil vertebra would have fallen. See on p. 7: "The fossil vertebral columns were not complete, but location could be approximated within one position in either direction in the column for each vertebra based on the location of the para- and diapophyses on the centrum using an extant and complete vertebral column from *Alligator mississippiensis* (TMM M-12606) and nearly complete fossil vertebral column from *Allosaurus fragilis* [38] as models."

Table 1: *Postosuchus kirkpatrickorum* is the correct name. It's been wrong for decades. Fixed

How was the femur length for *Desmatosuchus* determined? Did you use *D. smalli*? Yes we used *D. smalli*

Desojo et al, 2012 describe the structure as present in *Aetobarbakinoides*. We disagree.

Paratyothorax also has a similar structure (TTUP-09416). However, only *Tyothorax*, *Desmatosuchus*, and *Longosuchus* have a split hypantrum that is not connected along its entire anteroposterior length by a bony bar. We say that none of these count as a true hyposphene based on the definition put forth by Stefanic and Nesbitt (2018) and that *Longosuchus* is ambiguous in terms of presence/absence.

Reviewer: 3

Comments to the Author(s)

This paper examines the evolution of vertebral accessory joints in Archosaurs, and specifically tests the hypothesis that development of the hyposphene-hypantrum may relate to biomechanical stabilization as a consequence of large size. While I like the concept of this paper, and the hypothesis is interesting, I think there could be significant improvements in terms of both its communication and analyses implemented.

Firstly, the methods need to be better explained and could be improved in several regards. Key issues are a. the exclusion of certain groups without sufficient justification, b. the need for phylogenetic correction of the logistic regression analysis, and c. statistical tests for the joint size analysis (see more below).

Another major issue with this paper is that the methods, results and discussion are not presented in a logical sequence, making it quite difficult for the reader to follow. For example, the results figures are cited extensively in the methods, before the results have been stated. Also, several methods justifications are included within the results. Finally, detailed descriptions of the main results data of the paper (articulation presence) doesn't come until the discussion. I suggest the text be reorganized.

Specific comments:

Methods

- Body size correlation – Figure 1 is cited here in the methods. This is clearly a results figures and so shouldn't be cited here. **The reviewer is correct. We took all results figure citations out of the methods section and renumbered all of our figures.**

- You analyzed pseudosuchia and avemetatarsalia separately. In the intro or here it would be good to define these groups and state why you separate them instead of looking across archosaurs (or if you did so, put these results in supp). **We only analyzed them separately when we discovered that they have different body size thresholds for presence of the hyposphene-hypantrum articulation. We added a sentence to the third paragraph of the Introduction, "We focused solely on members of Archosauria because no other vertebrate outside of the group has been reported to possess these articulations or any structures homologous to the articulation."**

- Phylogenetic survey – again, don't cite results here. **We fixed this and renumbered all of our figures.**

Measurements

- It is stated that articular surface area is measured, however based on the description in the text it is actually articular surface width? Please clarify this.

We modified this section to be clearer about the measurements we took: "... measure the articular processes (the postzygapophyses in all taxa and hyposphene structures when applicable) on the posterior aspects of trunk vertebrae to track surface length involved in intervertebral articulation. Postzygapophyses were measured from the point where they contact the neural arch to their distal-most aspect. Hyposphenes were measured along the entire aspect of their lateral faces. To obtain a unit-less metric independent of body size, we also measured centrum height for each vertebra. By dividing centrum height by the length (long-axis) of one postzygapophysis for each vertebra observed, we obtained a measurement that could be used as a proxy for relative surface area of articulation between vertebrae."

- The standardization procedure also seems unconventional. If I understand correctly, the centrum height is divided by the articular width. Usually, the measurement of interest is divided by the standardizing measure? Please explain.

While unconventional, our standardization procedure produced the unitless metric that made sense to us. As long as there is standardization across all of our data the method for obtaining the metric should not matter that we divided height by width.

- Centrum height is used for standardization to "obtain a unit-less metric independent of body size". However, this assumes that centrum height scales isometrically to body size. Is there any evidence supporting this? Why was centrum height chosen over other measures? Please include discussion of these effects

We are not getting full body size, just relative body-size, and centrum height was available for all of our specimens.

- The method of 'correcting' the photographic measurements is also confusing. I think it would help to state your motivation at the beginning of the paragraph. I assume this correction is to take into account out-of-plane distances? How significant do you think this effect is? Added justification is required. The labelling on Figure 6 is also confusing as the length on the 2D image is labelled as A but is described as B in the text.

Reworded the section explaining the conversion factor: "To make sure that the photographic measurements were comparable with our in-person measurements, because the measurements in-person were in a slightly different three dimensional plane than those taken from a photograph in Photoshop, we took both measurements "A" (in-person) and "B" (in Photoshop) (Figure 2) on the same vertebrae to determine a standardized conversion factor of how different those measurements were from each other. That way, we could multiply one measure by that factor to convert it to a number about equal to what the other measure would be."

We fixed the labeling on Figure 6, we did have A and B accidentally swapped in the figure.

- Statistical analyses – The study rightfully emphasizes placing the data into their phylogenetic context, but this is not reflected in the statistical analyses. Phylogenetic logistic regression is available through the r package phylolm, and would be appropriate here.

The test illustrated in Figure 4 is specifically meant to exclude phylogenetic context once within either Pseudosuchia or Avemetatarsalia and show that within these groups, how well does body size reflect whether or not a species will exhibit the hyposphene-hypantrum articulation.

Results:

- Transition ranges and thresholds are raised here for the first time. Please define how these are calculated in the methods.

Added to the end of the methods under (e) Statistical Analyses section: "Our results will yield threshold ranges for body size that can be interpreted as above the range you expect the hyposphene-hypantrum to be present and below the range you expect it to be absent. The range itself represents a transitional body size where the articulation may or may not be seen in taxa of that size."

- There is some conflation of results and discussion here. For example, the exploration of potential size variation in *Mandasuchus* seems like a discussion point to me. Similarly, the detailed descriptions of absence and presence in different taxa from the discussion belongs in the results I think. These are the primary data collected by the study, so I think it would be clearer if they were summarized in the results section. Similarly, the discussion of titanosaurs seems out of place here.

Moved, "among the three known individuals of *Mandasuchus tanyauchen*, the one from which a complete femur is known (NHMUK PV R6792) is not the largest reported of the taxon [40], so it is clear that *Mandasuchus tanyauchen* grew larger than the holotype. This hints at a relationship between ontogeny and the presence or absence of the hyposphene-hypantrum in that the adult or near maximum body size may be important in determining the presence or absence of the structure. Unfortunately, few pseudosuchians are represented by a growth series and this cannot be explored further presently." to the discussion section.

Also moved, "A confounding scenario in the distribution of the hyposphene-hypantrum in archosaurs is the reported loss of those structures in titanosaurs, taxa well above the body size threshold (Figure 6) [18]. These are the largest of all of the dinosaurs [3], and the hyposphene-hypantrum articulation is lost near the base of the clade (Figure 6). The earliest diverging members of Titanosauria, *Andesaurus delgadoi* [43] and *Phuwiangosaurus sirindhornae* (SM K11-0038, [44]), both clearly possess the articulation, but [18] demonstrated that later diverging lineages of titanosaurs lack a hyposphene-hypantrum. We have observed one titanosaur vertebra in-person, *Alamosaurus sanjuanensis* (TMM 41891-1) and agree that it does not have the hyposphene-hypantrum articulation; however, one vertebra out of the trunk series of an individual may not be indicative of the presence or absence for that genus. The authors of [18] cited the late diverging titanosaurs *Argentinosaurus huinculensis* and *Epachthosaurus sciuttoi* as possessing "hyposphenal bars" but not hyposphene-hypantrum articulations based on the definition we have provided. However, the original publication of *Argentinosaurus huinculensis* [45] figures a trunk vertebra with a clearly defined and labeled hyposphene that does fit our definition. Furthermore, although the structure in most titanosaurs does not fit our definition, the condition in some titanosaurs might be a highly derived modification of the hyposphene-hypantrum. For this study, we consider the hyposphene-hypantrum to be absent in titanosaurs and because of their large size we consider this absence noteworthy.

Because of their extremely derived vertebral morphology and lack of a hyposphene-hypantrum articulation in the gigantic titanosaurs, we eliminated these taxa from our body size logistic regression for avemetatarsalians. We excluded the pseudosuchian *Sillosuchus longicervix* from our logistic regression as well due to its ambiguity in hyposphene-hypantrum presence or absence. We also exclude all ornithischian dinosaurs from our analyses because they are a major dinosaur clade that never possesses the hyposphene-hypantrum articulation and therefore the absence of the articulation in later diverging, large ornithischians may be attributed to phylogenetic constraint. Ornithischians have also been cited as possessing ossified tendons in their vertebrae (Luca, Organ, Zhou), which may be a secondary mechanism for vertebral bracing. The distribution of ossified tendons in Ornithischia and its relationship to larger body size should be investigated further; however, we choose to exclude the clade in this study. Although we include several ornithischian dinosaurs in Figure 6 and Table 1, for completeness, we did not include these taxa in our logistic regression analysis.

We also chose to exclude pterosaurs as a whole for this study because we could not confirm the presence or absence of the hyposphene-hypantrum in any of the specimens of early pterosaurs. This is because most are preserved as flattened slabs and are so small that even in uCT it is virtually impossible to see between the vertebrae to confidently score presence or absence of the hyposphene-hypantrum articulation.” to the discussion section

- Along the same lines, the fact that taxa (titanosaurs and ornithischians) were excluded from the statistical analyses is raised for the first time in the results. This should be discussed and fully justified in the methods, as they currently seem very ad hoc. The authors argue that ornithischians do not develop joints due to phylogenetic constraints, yet one of the main findings of the study is that size overcomes phylogenetic constraints. I think this requires more discussion and justification.

Added a sentence justifying these exclusions to Methods section (e) Statistical analyses: “We excluded several groups of archosaurs that lack the hyposphene-hypantrum from our statistical analyses (e.g., titanosaurs, ornithischians, pterosaurs) that we consider exceptions because they lose the articulation at or before the base of their clade and then have secondarily derived vertebral morphology before taxa grow to large body sizes (see Discussion section: (b) Exceptions within Archosauria (extinct lineages)).”

- The authors compare body size to articular surface area/width, suggesting that taxa with accessory articulations have smaller joints, but do not provide statistical tests. This could be tested with an ANCOVA/PGLS, so it is not clear why qualitative comparisons are relied upon. Figure 7 could also be made clearer (see below).

For these data shown in Figure 8a and 8b (in the previous submission they were numbered 7b and 7c) we are showing that specimens that have a hyposphene-hypantrum plot distinctly from specimens that do not have a hyposphene-hypantrum in terms of relative surface area of postzygapophyseal articulation. We show this visually using position along the vertebral column as our x-axis, because only datapoints from the same vertebral position can be compared to each other. We did not run a statistical analysis on these data because the subset of specimens/ taxa that had vertebrae from a majority of the trunk regions of the vertebral column was very low (and of course skewed towards the extant). To run an ANCOVA or PGLS we would need many more taxa than we sampled in this study and would need to run each statistical test on only data points of the same vertebral position. Any test we would have ran with this data would have had too low statistical power to be informative, so ultimately we chose to show these data as a visual relationship only. We will however continue to think about how these data could be tested using statistical methods.

Discussion:

As mentioned above, there is a lot of results-type content in the discussion which distracts from the major points being argued. I suggest condensing the discussion to focus on interpretation of the patterns already described in the results.

We have reorganized the manuscript, particularly the Results and Discussion sections, based on this reviewer's comments.

Biomechanics –

There is some literature on mammals which may be useful to draw on here, with regards adaptations/joints of the vertebral column to increasing size (e.g., Jones, 2015; Chen et al., 2005; Halpert and Jenkins, 1987). Also Jones and Holbrook (2016) test a similar hypothesis in horses.

We have removed the Biomechanical section from the manuscript .

Figures:

- I think there should be a figure showing the anatomy of the hyposphene/hypantrum in relation to other vertebral joints for readers unfamiliar with this anatomy. Created a new figure (cited as Figure 1 in the text) showing vertebrae with (aetosaur and prosauropod) and without (crocodilian and bird) the hyposphene hypantrum in posterior view, labeling the postzygapophyses and hyposphene where applicable. Figure caption: “Examples of articular surfaces on the posterior aspect of several archosaur trunk vertebrae, including extant species that lack the hyposphene-hypantrum, *Alligator mississippiensis* [TMM M-12606] (a) and *Struthio camelus* [NMNH 291160] (b), and extinct taxa that possess the hyposphene-hypantrum articulation, *Desmotosuchus spurensis* [MNA V9300] (c) and *Plateosaurus* [AMNH 2108] (d). pz: postzygapophysis, ho: hyposphene. Scales = 1 cm.”

- The order of the figures seems off (as with the text). Figure 6 relates to methods, and should come before the results figures. Figure 6 is now Figure 1 and all figures have been renumbered.

- Seems like there is some repetition in the figures. Can Fig 2/4 and Fig 3/5 not be combined? Figures 3 and 5 show different taxa and cannot be combined. Figures 2/4 however, have been combined and they are now referenced in the text as Figure 5.

- Figure 7 – mistake in the caption, a doesn't seem to show how measurements were taken. Fixed.

- Color mistake in legend “outside Archosauria” We found no mistake here.

- Figure 7 – These graphs are really difficult to read. Maybe stem and leaf plots by size group would be easier? We think that these graphs are adequate and we are leaving them as they are because no other reviewer commented negatively on them.

- Figure 7a – this seems like a methods figure as it is comparing the different data collection techniques. I think this should be moved earlier or put into the supplement. Referenced this figure in the Methods section and changed its number to Figure 3.

Tables:

- Only 3 table captions listed but (review was cut off here) There are three table captions and three tables included with this manuscript.

Appendix B

Reviewer comments to Author:

Reviewer: 2

Comments to the Author(s)

The authors have addressed my concerns from the initial draft. The definition of what constitutes a hyposphene-hypantrum articulation will continue to be debated and I don't entirely agree with the findings of the authors as to what taxa possess this feature; however, their proposed definition has been clearly stated and their identifications are consistent with that definition so I am fine with publication as is. Future studies will support or contest these conclusions.

Reviewer: 3

Comments to the Author(s)

While the structure of the manuscript is much improved, I still feel that it requires significant work to improve clarity and justify its conclusions. Further, there are several points from the original review which remain unaddressed. To my mind the most significant issues are: a. statistical analyses, b. the way the excluded groups are dealt with, and c. several issues with presentation of data. Before publication the authors should implement appropriate phylogenetic statistical analyses, represent and analyze the complete dataset prior to excluding the outlier groups and ensure that all figures are legible, correct and clearly presented.

See my responses to select author replies below:

- You analyzed pseudosuchia and avemetatarsalia separately. In the intro or here it would be good to define these groups and state why you separate them instead of looking across archosaurs (or if you did so, put these results in supp).

We only analyzed them separately when we discovered that they have different body size thresholds for presence of the hyposphene-hypantrum articulation. We added a sentence to the third paragraph of the Introduction, "We focused solely on members of Archosauria because no other vertebrate outside of the group has been reported to possess these articulations or any structures homologous to the articulation."

This is not what I was driving at here. Why are the groups separated? If they do have different thresholds, how do you interpret that? Suggests interaction between phylogeny and biomechanics.

We added the following sentence to the "statistical analysis" section: "*We analyzed the two clades separately because we noticed that species on each side of the tree gained the hyposphene-hypantrum at different body sizes, and therefore combining them would not give a true signal how related presence of the articulation is related to body size.*"

-

- It is stated that articular surface area is measured, however based on the description in the text it is actually articular surface width? Please clarify this.

We modified this section to be clearer about the measurements we took: "... measure the articular processes (the postzygapophyses in all taxa and hyposphene structures when applicable) on the posterior aspects of trunk vertebrae to track surface length involved in intervertebral articulation. Postzygapophyses were measured from the point where they contact the neural arch to their distal-most aspect. Hyposphenes were measured along the entire aspect of their lateral faces. To obtain a unit-less metric independent of body size, we also measured centrum height for each vertebra. By dividing centrum height by the length (long-axis) of one postzygapophysis for each vertebra observed, we obtained a measurement that could be used as a proxy for relative surface area of articulation between vertebrae."

While I appreciate the enhanced clarity here, it could still be improved. "Distal-most" (distal to what?) would be clearer as "dorsal-most". Also referring to the "long-axis" of the zygapophysis is confusing as long-axis is generally cranio-caudal in the axial skeleton. I would suggest "maximum length of the articular surface". Same on P9, line 17.

Also, this does not justify use of relative surface area throughout the text – this is misleading as it is not what was measured. Alter wording to use 'maximum length' or something similar throughout the paper.

Changed wording to “The postzygapophyses were measured *along the maximum length of their articular surface*— from the point where they contact the neural arch to their distal-most aspect. Hyposphenes were measured along their lateral faces” on p.7. Also added “maximum length of...” on p. 9 and throughout (see teal highlights)

-

The method of ‘correcting’ the photographic measurements is also confusing. I think it would help to state your motivation at the beginning of the paragraph. I assume this correction is to take into account out-of-plane distances? How significant do you think this effect is? Added justification is required. The labelling on Figure 6 is also confusing as the length on the 2D image is labelled as A but is described as B in the text.

Reworded the section explaining the conversion factor: “To make sure that the photographic measurements were comparable with our in-person measurements, because the measurements in-person were in a slightly different three dimensional plane than those taken from a photograph in Photoshop, we took both measurements “A” (in-person) and “B” (in Photoshop) (Figure 2) on the same vertebrae to determine a standardized conversion factor of how different those measurements were from each other. That way, we could multiply one measure by that factor to convert it to a number about equal to what the other measure would be.” We fixed the labeling on Figure 6, we did have A and B accidentally swapped in the figure.

This is better, but there is still a problem with Figure 2. There seems to be a panel missing (oblique view is described in caption but does not appear in figure), so measurement A is not illustrated anywhere in the figure, making the triangle with A and B very confusing. Both measurements A and B need to be illustrated or this figure is uninterpretable.

Changed introduction of the conversion factor paragraph to: “To account for the postzygapophyseal photograph measurements being slightly out-of-plane from the in-person measurements, we computed a conversion factor between the two methods. To do this, we took both types of measurements on the same vertebra to determine how different those measurements were from each other. That way, we could multiply one measure by that factor to convert it to a number about equal to what the other measure would be (Table 4).”

We see the confusion with interpreting our figure now. We changed Figure 6 to include a new view (ventral) of an alligator vertebra with measurement A illustrated.

-

P10, line 10: Make a statement about how variable zygapophysis angle is in archosaurs and if you expect that this could lead to any systematic bias (or not).

Expanded on this sentence to say: “Our assumption in using this conversion factor for all of the fossils surveyed using this method is that the angle of the postzygapophyses in relation to the neural arch is similar across all archosaur taxa, *and based on personal observation we conclude that this is a fair assumption to make here.*”

- Statistical analyses – The study rightfully emphasizes placing the data into their phylogenetic context, but this is not reflected in the statistical analyses. Phylogenetic logistic regression is available through the R package phylolm, and would be appropriate here.

The test illustrated in Figure 4 is specifically meant to exclude phylogenetic context once within either Pseudosuchia or Avemetatarsalia and show that within these groups, how well does body size reflect whether or not a species will exhibit the hyposphene-hypantrum articulation.

It is not possible to “exclude phylogenetic context” here. For example, the patterns shown in the logistic regression in Figure 4 could be achieved by a single evolutionary transition with multiple species sampled on either side – in which case the true sample size would be $n=1$, and there would be no way to determine statistical significance based on such a single occurrence. I’m not saying this is the case here, simply making an extreme example to illustrate my point. Most likely given the observed patterns the effect would still be significant, but it needs to be shown. An uncorrected analysis would be appropriate if the trait were 100% labile with no phylogenetic signal, but this is clearly not the case here.

The authors should therefore use a phylogenetically informed analysis for determining significance. The options are either a phylogenetic logistic regression, or a PGLS. Given that both variables are observed with error, and there isn’t a clear controlled variable here, it’s not important which is dependent vs independent. PGLS would allow you to ask if taxa with joints are larger than those without, and is simple to implement in R. I suggest authors run the analysis both with and without the ornithischians and titanosaurs to assess the effect.

The size variable should be logged both in Figure 4 and in the analysis as its distribution is skewed by the very large taxa. This will also aide comparisons between the data subsets.

We performed a two-tailed heteroscedastic t-test on all of the data from Table 1 that didn’t have a “?” for femoral length or presence/absence of hh and got a value of 0.279, when we took out ornithischians, titanosaurs, crown crocodylians and birds, our value was 0.0001. This is an indication that those taxa are outliers, as we mention in the manuscript as a reason why they were excluded. We could not however get our data to work in a PGLS. Since the trees we used in our figures were created by hand by combining phylogenies from the literature we did not have branch lengths and we did not have the time and capacity over the two weeks since receiving this second round of reviewer comments to figure out how to accurately add them. We regret that we could not fulfill Reviewer 3’s request of running a PGLS, and we hope that our careful consideration of all of their other comments and major revisions to the manuscript can make up for this.

-
Results:

- Transition ranges and thresholds are raised here for the first time. Please define how these are calculated in the methods.

Added to the end of the methods under (e) Statistical Analyses section: "Our results will yield threshold ranges for body size that can be interpreted as above the range you expect the hyposphene-hypantrum to be present and below the range you expect it to be absent. The range itself represents a transitional body size where the articulation may or may not be seen in taxa of that size."

This does not explain how these ranges are calculated. Based on Figure 4 it seems that the range is simply the smallest specimen with a joint to the largest specimen without a joint. Is that correct? Or is it derived from the logistic regression somehow? Clarify.

We re-worded the last few paragraphs of the "statistical analyses" section to be more clear about the threshold ranges: "The plot of this logistic regression will yield visual threshold ranges for body size that can be interpreted as above the range you expect the hyposphene-hypantrum to be present and below the range you expect it to be absent. *These ranges are simply the values of femoral length of the smallest taxon reported to possess the hyposphene-hypantrum and the largest taxon reported to lack the hyposphene-hypantrum. Within the range, we interpret a transitional body size where the articulation may or may not be seen in taxa of that size.*"

-
Along the same lines, the fact that taxa (titanosaurs and ornithischians) were excluded from the statistical analyses is raised for the first time in the results. This should be discussed and fully justified in the methods, as they currently seem very ad hoc. The authors argue that ornithischians do not develop joints due to phylogenetic constraints, yet one of the main findings of the study is that size overcomes phylogenetic constraints. I think this requires more discussion and justification.

Added a sentence justifying these exclusions to Methods section (e) Statistical analyses: "We excluded several groups of archosaurs that lack the hyposphene-hypantrum from our statistical analyses (e.g., titanosaurs, ornithischians, pterosaurs) that we consider exceptions because they lose the articulation at or before the base of their clade and then have secondarily derived vertebral morphology before taxa grow to large body sizes (see Discussion section: (b) Exceptions within Archosauria (extinct lineages))."

The exclusion of these groups still seems ad hoc. It would make more sense to plot all the data and allow the data to identify those groups as outliers, at which point you can discuss why they might be outliers and exclude them and look at patterns without them. The primary data presented by this study are the taxonomic survey data of presence of hh joints (Table 1), so a figure displaying this data in its entirety would be helpful. For example, a scatterplot of presence vs size by phylogenetic subgroup.

We created a new figure (Figure 3) using the data in Table 1 that is a scatterplot of presence vs size with phylogenetic subgroups in different colors and outliers called out with labels.

-

- The authors compare body size to articular surface area/width, suggesting that taxa with accessory articulations have smaller joints, but do not provide statistical tests. This could be tested with an ANCOVA/PGLS, so it is not clear why qualitative comparisons are relied upon. Figure 7 could also be made clearer (see below). For these data shown in Figure 8a and 8b (in the previous submission they were numbered 7b and 7c) we are showing that specimens that have a hyposphene-hypantrum plot distinctly from specimens that do not have a hyposphene-hypantrum in terms of relative surface area of postzygapophyseal articulation. We show this visually using position along the vertebral column as our x-axis, because only datapoints from the same vertebral position can be compared to each other. We did not run a statistical analysis on these data because the subset of specimens/ taxa that had vertebrae from a majority of the trunk regions of the vertebral column was very low (and of course skewed towards the extant). To run an ANCOVA or PGLS we would need many more taxa than we sampled in this study and would need to run each statistical test on only data points of the same vertebral position. Any test we would have ran with this data would have had too low statistical power to be informative, so ultimately we chose to show these data as a visual relationship only. We will however continue to think about how these data could be tested using statistical methods.

Please state the sampling for this analysis in the methods. Also please clarify how the data in figure 8 relate – is each point a single specimen or a mean (see below)? Is it a mix of specimens? Given there seems to be no along-column trend in joint length, one possibility is to calculate mean length for each specimen by averaging across vertebrae. This may give you enough sampling to statistically test between those with and without the joints, both including and excluding the extra articulation surface.

We have taken out all of the articulation surface graphs (Figures 3, 8a, 8b) and created now ones in their place (Figures 8, 9) to be easier to interpret and less busy/ noisy. We hope that these new plots communicate our hypothesis and ideas about articulation length/ surface area better. We still think that our sample size is much too low to run any statistical tests with substantial statistical power.

Figures:

- Figure 7 – mistake in the caption, a doesn't seem to show how measurements were taken

Fixed.

As stated above this figure still doesn't show how measurement A was taken. I think a panel must be missing as it doesn't make sense as is.

Updated the measurement figure to show both measurements A and B.

-

- Color mistake in legend "outside Archosauria"

We found no mistake here.

I'm afraid there is a mistake in the key. I cannot see any points in the plot corresponding to the cream crosses "outside Archosauria <300mm" category in the key, whereas I can see various green crosses.

This figure no longer exists.

-

Also, Figure 3 & 8 – What do these data represent? It is unclear if these data are individual specimens or group means. If individual, which species are represented (list in caption)? If they are group means, please display some measure of spread. Why is there (a) and no (b) in 3. There is not enough information to interpret these figures.

These figures no longer exist.

-

- Figure 7 – These graphs are really difficult to read. Maybe stem and leaf plots by size group would be easier?

We think that these graphs are adequate and we are leaving them as they are because no other reviewer commented negatively on them.

While these plots may be 'adequate', they are far from compelling. There are too many groupings to convey a clear message. I suggest the authors simplify/clarify the figures if they want readers to come away with a clear take-home.

We have taken out all of the articulation surface graphs (Figures 3, 8a, 8b) and created new ones in their place (Figures 8, 9) to be easier to interpret and less busy/ noisy. We hope that these new plots communicate our hypothesis and ideas about articulation length/ surface area better. We still think that our sample size is much too low to run any statistical tests with substantial statistical power.

-

Additional points:

• **Do you have any hypotheses you are testing with this measurement data, as these results aren't currently placed into any specific framework? For example, do you hypothesize that taxa with the joints will have longer articular surfaces, relative to size, than those without? Why might you expect that? This analysis is difficult to interpret because the context is not laid out and questions are not clear.**

We hypothesize that taxa with the joints will have relatively shorter zygapophyseal lengths in relation to size and that when adding in the lengths of the hyposphene, the overall articulation length will be about the same as taxa with only zygapophyseal articulation – because we hypothesize that the hyposphene-hypantrum increases complexity of articulation without increasing lengths of surfaces in contact with each other. We hope this is now clearly expressed through our new figures, Figure 8 and Figure 9 and how we explain them in our "results" section.

- **P14, line 5 onwards: A tentative link between size, hh and joint length is raised in the results but not tested nor its implications discussed (see setting up framework above). Either give this result context and discuss it (even if the conclusion is we don't have enough information to accept or reject the hypothesis right now) or remove it.**

As stated above, we hope that we now adequately address this framework/ idea of hh articulation presence or absence and zygapophyseal articulation length being related to one another.

- **The authors conclude that the hh joint is "closely constrained by body size in extinct archosaurs". However, it seems it is both related to body size and phylogeny – there are certain major phylogenetic groups for which the joint never evolves, for biomechanical or other reasons. Therefore, the evolution of this trait is more accurately described as a mosaic of straight allometry and phylogenetic inertia.**

We changed our wording in the "conclusion" to be very clear about our takeaway message that body size determines acquisition of the trait UNLESS there is another mechanism in place doing the same thing, rendering the hh unnecessary for growth to large body size: "In all living members of Archosauria, the hyposphene-hypantrum articulation is absent. However, the fossil record shows that it was once widespread and closely constrained by body size *and whether or not an alternative vertebral bracing mechanism is present.*"

Appendix C

Dear editors, First of all, we greatly appreciate the intensive effort of Padian during this whole process, especially at the end of this process. We have gone through his comments (both authors) and changes carefully and have put together a much improved manuscript through his guidance. Most of the changes suggested by him address the concerns of previous reviewers and reviewer 4 below. We understand that reviewer 4 was frustrated with this paper, but also it is clear that there was little constructive criticism and many of the comments are difficult to interpret what the reviewer was trying to convey. Therefore, we did our best to satisfy some of the larger concerns and we do agree that we improved the paper based on the suggested changes. In our “tracked changes” manuscript upload, changes we made during this round of peer-review are highlighted in green. Changes made during previous rounds of review are highlighted in yellow and teal. We address some of the larger changes below. Our responses are in bolded font.

Reviewer comments to Author: Reviewer: 2 All of my concerns have been addressed.
We thank Reviewer 2 for their time and input throughout this peer-review process.

Reviewer: 4 Hyposphene-Hypantrum ms (RSOS-190258.R1)
General observations

What is the general nature of this ‘hypothesis-driven’ article? It is narrowly focused on their tightly constrained definition of one anatomical configuration (the hyposphenehypantrum [hypo-hypa]) and its distribution within sub-clades of archosaurian reptiles.

This promotes the idea that the hypo-hypa is a solely size-related construct (i.e. they are analogues randomly distributed across, and restricted to, large-sized archosaurians). The thrust of their argument is that there is no underlying phylogenetic component to their appearance/distribution in individual lineages/clades. Some do, some don’t – and if they don’t they have alternative mechanisms to solve the ‘problems’ of rigging and bracing large skeletons.

This ‘notion’ is backed up by statistical analyses and graphical presentations that are purporting to support this general thesis.

Unfortunately, this general thesis is undermined by their observation that basal (=ancestral) archosaurs inherited a common tendency to develop these structures (page 16:10), which are then expressed, or not expressed, later in their phylogeny. This is not an inspiring juxtaposition of concepts. If hypo-hypa are indeed part of a common inheritance then these features are arguably homologous (as the intricate nature of their similarity across taxa implies) rather than analogous, as the tenor of this article seems to wish to imply.

Figures. Annotation abbreviations in the text would be appropriate if the reader wants to cross-reference anatomy that is being described textually with what has been CLEARLY(?) Illustrated. But, the illustrations are inadequate and confusingly annotated.

As a trivial point of fact, I have a dorsal vertebra of a quite large *Crocodylus niloticus* on my desk (as I type this), which has a structure not unlike a hyposphene between its postzygapophyses! However, it is more of a functional/structural analogue than a feature that fits into their narrow definition of what a hypo is, of course.

First impression: this article is not at all well written and would need to be carefully rewritten throughout. I thought it was just the abstract/intro, but this problem is pervasive. The language is occasionally inaccurate, syntactically challenged and quite often confused/confusing to the reader. A non-specialist reading this article would (I fear) be left bemused by this level of literacy. The Introduction is particularly poor in this respect – rather summed up by the last three sentences therein.

In-text commentary

1. Minor point. Abstr and page 2:11. "20-70 tonne" dinosaurs. These are, quite frankly, ridiculously OTT estimates of mass/weight because limb bone strength values will not support such weights under static loading on land, let alone under dynamic loading – the latter would prompt immediate failure! Such 'calculations' seriously overestimate the density of these large animals, but are used to add 'drama' to the story (I suppose).

2. The focus is on hypo-hypa, but there is no comment about interspinous ligaments in theropods and sauropods; these are very prominent and leave complex and rugose bony scars on the leading and trailing edges of dorsal spines. These ligament bands would have acted as ossified tendon equivalents in stabilizing and strengthening the dorsal vertebral column - if we are comparing, as they do, all three dinosaur clades.

3. Page 3:17 "Many orders of magnitude" – "Many"? ... "smallest and largest" (what dimensions are the authors talking about?).

4. Persistent problem. The use of English is not consistent, or particularly good. A few trivial examples: Page 3:9 - the disparity of body sizes - you mean body size! "sizes" seems to be used ubiquitously and incorrectly. Page 4: 19 - you are not testing a "question" you are exploring the implications of an hypothesis; Page 6:2 "posterior neural arch" (anatomically that makes no literal sense) and gives the impression that there is an anterior and a posterior neural arch (which is absurd).

5. Page 6:6. Posterior zygapophyses DO NOT face dorsally – this is fundamental anatomy. Page 6:8 – reference to Figure 1. The figure is woeful, it needs to be a diagram which shows clearly the features in question and has clear annotations – and there should be a distinction between the hypo and the hypa as well as the zygapophyses (which are equally distinct) in such images – the use of the same annotation simply adds to the general 'air of confusion' in this ms.

6. Page 8:12-15. Measurements - why should length of the postzygapophysis vs centrum height create a proxy that has anything to do with size-corrected surface area of contact between vertebrae? I simply fail to understand the reasoning behind this. The logical choice would have been to use the surface area of the centrum articular surface scaled against femoral length – that does not require a, quite frankly, bizarre 'proxy'. The impression given is that 'dimensional data' are to some extent contrived.

7. Page 8: 16+ Did the same measurement collection for named crocs (2) and several birds 'normalize' for vertebral 'size'? How does that correlate with what was measured in big dinosaurs? I really don't understand the why? and what? – is going on here.

8. Page 9: first para. Wording is clumsy. And, why was such precision needed in partial vertebral columns? Again the wording and phraseology on this page are simply clumsy and imprecise. Trivia – but ... Page 9:19 "millimetres"(spelling? US/UK) – a millimeter in a UK journal would be a reference to an instrument that measures "millis"!

9. Page 10:8. What does "... angle of the postzygapophyses in relation to the neural arch" actually mean anatomically and why is it important? This is totally unclear and yet it becomes a "fair assumption". Do we need a simplified diagram/figure where all these measurements or estimations are indicated and annotated appropriately?

10. Page 10:19. "We plotted only trunk vertebrae (presacral position 10 to the last presacral before the sacrum)" - Q. Is it not clear that this is pedantic repetition? Or, worse still, a matter of literacy?

11. And on the same line ... "position 10" ... again what does that mean? - is it Dorsal 10 ... counted (from where?) established on what basis? ... or is this 'guesstimation' based upon comparison drawn from alligator and allosaur vertebral columns? A further question that arises from this is: why should it be that hyposphene-hypantra only exist in one restricted region of the dorsal column? The biomechanical implications actually seem far more interesting per se than what is being tortured in this article.

12. Page 11 - Statistical analyses - this section (again) lacks literacy. I am at a loss to explain how such garbled English can be generated in a re-read, adjusted, pored over and carefully revised article, such as this?

13. Results - Page 12 onwards. Firstly, the opening paragraph is simply a mangled version of English. Later on, the statements are inconsistent and partly contradictory (again this is made more difficult to follow and understand than it should be because of the extremely poor English usage). I extract the last paragraph of this section: Pages 14/15: "Our analysis of maximum length of zygapophyses as a proxy for articular surface area in extant and extinct archosaurs showed that among members of crown Crocodylia and crown Aves, relative articular surface area is roughly the same, when corrected for body size. (Figure 8, Table 2, Table 3) Additionally, we notice that for the extinct archosaur taxa we analyzed that possess the hyposphene-hypantrum (e.g., *Dilophosaurus*, *Desmatosuchus*) (Figure 9, Table 4), the more inclusive measurement of articular length (postzygapophyseal maximum length + hyposphene maximum length / centrum height) plot more closely with the extant data, whereas the less inclusive measurement (orange data points) plot below the extant data. Extinct taxa that do not have hyposphene-hypantrum articulations plot closely with the extant data. However these relationships are merely visual and we do not have enough data to run a statistical test with enough power to be meaningful. We do however believe that this is an interesting pattern worth investigating further." Just ignoring the lazy punctuation and typos, the entire section has to be read and reread as I try to extract meaning and sense. This isn't Kantian philosophy!

15. Discussion. Page 15 onwards. Starts off reasonably, but .. "Outside Archosauria, the hyposphene-hypantrum articulation is almost always absent, regardless of body size. The one clear exception is the presence of the accessory articulation structures in the stem archosaur *Azendohsaurus madagaskarensis* (FL = 205 mm, FMNH PR 2779, [46]). This condition differs from all archosaurs; however, because the hyposphene hypantrum articulation is only present in one vertebra in the anterior trunk and not present throughout the trunk [46]. Q. Anything wrong with this section? One, it is clumsily phrased and ungrammatical. Two, this information could be written far more succinctly. Three, the critical "almost" rather undermines the general thesis – the example described shows the feature that they define so rigidly so ... it is there! This is a significant observation because of the logical implications that flow from it. Facts that don't fit are admitted and then dismissed as irrelevant ...

16. I am sorry, but the poor grammar and syntactical errors just get more and more irritating in the Discussion section. Why was such poor English usage not weeded out a long time ago? It is intolerable in a manuscript that has evidently been through the review process for a prestigious journal ... TWICE! As an independent reviewer of what appears to be a problematic article (judged by previous peer review comments), I find that I keep losing patience and become increasingly (no doubt unfairly) critical.

17. Page 20. The authors' 'discussion' of the structure of intervertebral articulation in titanosaurs shows no intuitive or logical consistency. It simply confuses the reader and leads to a feeling of exasperation. This is simply poor expression, and intellectually inadequate.

18. 20:17. "For example, ornithischians have been cited as possessing ossified tendons in their vertebrae" - this is anatomically incorrect and very misleading. No doubt it is yet another example of inadequate English usage. But it also sows seeds of doubt about the anatomical competence of the authors.

19. Pages 20/21: Pterosaurs - some are actually known to be HUGE! So I am not sure that these authors are being entirely truthful about their reasons for excluding this group. Equally, I do not know whether some of the HUGE pterosaurs have wellpreserved dorsal vertebrae that might reveal critical features of their anatomy. But, rather like the example of birds, the stresses encountered by the vertebral column (or indeed the way they are braced) in such animals may be quite atypical and such factors are not really being considered in a biomechanical sense by these authors.

20. Logical issues. 16:10 Authors state that the common ancestor of Archosauria may have had the ability to form "the structure" (hypo-hypa). Followed by a very confused consideration of phytosaurs (if, as seems to be implied, phytosaurs are archosaurians per se). Then Page 16:23 the presence of this trait is stated as being "ambiguously optimized at the base of Archosauria ..." Then Page 16 the gain in this articulation is acquired independently (=convergently) in the pseudosuchian and avemetatarsalian lineages of archosaurs. Pseudos: it is regarded as an ontogenetic phenomenon in 'paracrocodylomorphs' (because of its distribution in known specimens of *Mandasuchus*). Among aetosaurs it seems to be size-related (effectively the same phenomenon). Avemetatarsalians: small examples with hypo-hypa include silesaurids and *Teleocrater* and small dinos: *Tawa*) but absent in the very small *Dromomeron*. Dino clades Therops & Saurops (yes) but Orns (no). Confounders: Titanosaurs ... no (why?) ... there is some discussion of the presence of possible hypo-hypa in early titanosaurs, and a decision is then made to consider that they don't have the fine details of hypo-hypa morphology as defined by the authors.

21. Page 22. Crocodylian procoely in dorsal vertebrae. Merited lengthy consideration of its role in stabilising intervertebral articulation. I mention this here because it highlights a deficiency in this article. Titanosaurian dinosaurs exhibit opithocoelic dorsal vertebrae - the reverse pattern of what is seen in crocodiles and yet surely a structural conformation that has the same potential benefits? Are the authors not aware of this morphology in titanosaurs?
22. Pages 24/25. Birds. There are large birds that do not have hypo-hypa. But why is that? There is no consideration of the unique thoracic construction seen in birds generally - or the ligament support of the neural spines (which resembles the ossified tendon trellises of ornithischians). Heterocoely and intervertebral bracing in more derived birds is perfectly reasonable (and translates across to the titanosaur observation).
23. Living archosaurs lack hyposphene-hypantral articulations. But both are HIGHLY DERIVED - flight or semi-aquatic (both media subject the vertebral column to entirely different forces, and these forces are resisted in their own unique ways. These are NOT strictly comparable to the more generalised TERRESTRIAL archosaurs of the Mesozoic Era. The authors are comparing apples and oranges (to use a metaphor rather badly). And there is that rather interesting large Nile croc dorsal vertebra on my desk.
24. Figures - a simple DIAGRAMMATIC vertebra (or exemplar vertebrae) should be illustrated rather than these unclear photographic images. The diagrams can then be CLEARLY labelled (the annotations are beyond minimal given the anatomical features described in the text) so that the features being described can be seen and appreciated by the viewer. The hyposphene and hypantrum (as well as the pre- and postzygapophyses) need to be DISTINGUISHED CLEARLY IN SUCH ILLUSTRATIONS, NOT GIVEN THE SAME ANNOTATION ABBREVIATION.
25. The graphical representations of the cladograms and the data (and the analytic techniques) are - I am forced to observe - a combination of: i) "We'll partition things this way, manually" and then ... ii) "Let's use a load of stats packages because that will provide us with graphical output that makes this intuitively-driven work look 'scientific'". Yes ... I realize that this is presented as a parody. However, I have to observe that the series of objections and clarifications called for by Reviewer 3 points to deeper levels of their frustration about the 'what did you do?' and 'why did you do it?' ... of the results that are presented.

Appendix D

Below are the minor revisions suggested by the editors with our responses in **bold**:

1/13: archosaurs evolved

fixed

7/8: because the

fixed

11/22: transitional

fixed

20/10: become large (actually I would frame this paragraph in past tense)

changed the paragraph to past tense

20/12: [add] the "small morph" of Q (Langston 1981, Scientific American, "Pterosaurs"), although posterior dorsals are missing; no vertebrae are preserved in the "large morph"

changed & added Langston (1981) to literature cited

22/10: allowed, not allowed for

fixed